# Physics from Video: Identifiability of Time-Invariant Second-Order ODEs under Minimal Trajectory Conditions

**Yuanyuan Wang** [* 1]  **Wenjie Wang** [* 2]  **Kun Zhang** [1 3]  **Mingming Gong** [1 2]

## Abstract

Bridging the gap between visual realism and physical understanding is a core challenge for video-based world models. We study the structural identifiability of continuous-time physical laws from raw pixels, focusing on whether an encoder-only pipeline can uniquely recover the parameters of second-order linear ODEs. We prove that a level-set slope-coverage condition ensures the learned latent space is locally affine to the true physical state, enabling exact parameter recovery. Our theory provides the first characterization of minimal data requirements across damping regimes, establishing that underdamped systems are identifiable from a single video clip, whereas other regimes require three diverse trajectories. We further introduce a variance-floor regularizer to stabilize the decoder-free objective and prevent latent collapse. Validated on synthetic and real-world data, our approach demonstrates that interpretable physical constants can be reliably estimated from video without the need for compute-intensive pixel reconstruction, ensuring both physical correctness and transparency. Code is available at https://github.com/wenjiewang3/PhysicsFromVideo.

## 1. Introduction

The quest for *world models* has renewed a central debate in AI: do modern generative video models truly internalize physical laws, or do they primarily act as powerful pixel predictors that achieve visual realism without reliable physical grounding? Recent physics-centric evaluations, such as

Physics-IQ (Motamed et al., 2025) (and related benchmarks (Kang et al., 2024)), suggest that while these models produce visually stunning videos, they exhibit a fundamental decoupling between visual fidelity and physical correctness. Specifically, they frequently violate basic physical invariants, revealing that high-fidelity synthesis can mask a systematic lack of grounding in the causal laws of the observed world. This discrepancy highlights a critical gap: achieving visual plausibility does not imply the recovery of the true, interpretable physical laws that govern the observed world.

This gap motivates a complementary goal: recovering *interpretable* physical laws and parameters directly from raw videos. Commodity cameras are increasingly used as scalable, non-contact sensors across infrastructure monitoring (Bai et al., 2023), digital health and biomechanics (Uhlrich et al., 2023; Boswell et al., 2023), and robotics/graphics (Zhu et al., 2025; Zhao et al., 2025). If pixels can be mapped to reliable continuous-time dynamical models, video can enable monitoring, prediction, diagnosis, and design where dedicated instrumentation is unavailable or costly. Such recovered laws can also act as explicit constraints or priors for physics-grounded video prediction and generation, helping bridge the disconnect between visual realism and physical correctness. Yet identification from video is fundamentally harder than classical system identification: the state is latent, supervision is limited, and the compact coordinate system in which the dynamics are simple is unknown. This leads to a foundational, yet unresolved, theoretical question:

*Can we identify the true underlying physical parameters from raw video, and if so, under what formal conditions?*

Supervised regression from video to states or parameters requires high-quality ground truth that is often expensive or infeasible to obtain at scale (Meijering et al., 2016; Varoquaux & Cheplygina, 2022), motivating unsupervised approaches that incorporate known governing equations as inductive bias. Many pipelines nevertheless rely on *reconstruction-driven* objectives, either via autoencoder–decoder designs (Kandukuri et al., 2020; 2022) or analysis-by-synthesis through differentiable rendering and photometric matching (Ma et al., 2022; Stotko et al., 2024; Zhao et al., 2025). Many such methods mitigate appearance ambiguity in prac-

---

*Equal contribution [1]Mohamed bin Zayed University of Artificial Intelligence, Abu Dhabi, UAE [2]University of Melbourne, Melbourne, Australia [3]Carnegie Mellon University, Pittsburgh, USA. Correspondence to: Mingming Gong <mingming.gong@unimelb.edu.au>.

*Proceedings of the 43rd International Conference on Machine Learning*, Seoul, South Korea. PMLR 306, 2026. Copyright 2026 by the author(s).

tice through structured decoders, restricted functional forms, or stronger rendering priors. Their identifiability, however, depends jointly on the dynamics model and the observation-model priors: when the reconstruction loss operates in pixel space and the decoder or renderer is expressive enough, it can in principle compensate for physically incorrect parameters, so the recovered parameters need not be unique without additional structural assumptions. Furthermore, high-fidelity reconstruction is data- and compute-intensive and can dominate training, even when the downstream task does not require pixel-level reconstruction. Recent decoder-free, encoder-only formulations (Garcia et al., 2025) take a bold step by dropping reconstruction and enforcing physics directly in latent space, substantially reducing reliance on pixel synthesis. However, while removing the decoder mitigates appearance confounding, it exposes an even more fundamental theoretical vacuum: without the pixel-level constraint, the latent coordinate system is defined only up to an arbitrary reparameterization. Consequently, a good fit in latent space provides no assurance that the recovered parameters are unique or match the true physical system. Establishing *structural identifiability* is therefore a prerequisite for reliable physics-from-video, especially in ill-conditioned regimes (Gutenkunst et al., 2007).

In this work, we study identifiability in an *encoder-only*, *decoder-free* setting, isolating when a video trajectory itself pins down a physically meaningful coordinate system and yields provable parameter recovery. Since identifying physical parameters from raw video is intrinsically challenging, we start with a simple yet widely used continuous-time model that enables sharp analysis: the homogeneous second-order linear time-invariant (LTI) ODE $z''(t) + \gamma_1 z'(t) + \gamma_0 z(t) = 0$, a canonical abstraction for damped resonance and single-mode transients that underlies ring-down/modal analysis and damping estimation in mechanical vibration and structural monitoring (Ewins, 2009; Li et al., 2022; Yokoyama, 2023; Noh et al., 2025). Because many applications rely on the *interpretability* of $(\gamma_1, \gamma_0)$, understanding when these parameters are uniquely recoverable from raw video is a prerequisite for using decoder-free vision as a reliable diagnostic instrument.

**Contributions.** We (i) provide, to the best of our knowledge, the first *structural identifiability* guarantees for learning continuous-time physical parameters *from raw video alone* in an encoder-only framework; specifically, for the homogeneous second-order linear ODE, we characterize when a single clip suffices and when multiple trajectories are necessary across damping regimes; (ii) derive non-asymptotic finite-sample error bounds that separate measurement noise, finite-difference bias, and design conditioning; (iii) propose a simple variance-floor regularizer that improves conditioning and stabilizes training.

## 2. Related Work

**Reconstruction-driven physics and parameter inference from video.** A large class of methods learns dynamics and physical parameters from raw video by coupling visual representation learning with differentiable time evolution and supervising via pixel-space reconstruction. In *autoencoder–decoder* pipelines, an encoder maps frames to latents, a dynamics module predicts their temporal rollout, and a decoder reconstructs future frames (often with masks, spatial transformers, or object-centric structure to encourage factorized explanations) (Watters et al., 2017; de Avila Belbute-Peres et al., 2018; Asenov et al., 2019; Kandukuri et al., 2020; 2022). A related *analysis-by-synthesis* line treats physical parameters as optimization variables and fits them through differentiable simulation and rendering or photometric alignment (Jaques et al., 2020; Murthy et al., 2020; Ma et al., 2022; Stotko et al., 2024; Zhao et al., 2025). While these approaches can avoid manual labels, parameter estimation is fundamentally mediated by the observation model: nuisance factors such as lighting, texture, background and viewpoint can compensate for incorrect dynamics yet still achieve low pixel loss. As a result, the inverse problem is often not identifiable, leading to shortcut solutions, sensitivity to initialization, and degraded transfer under domain shift.

**Decoder-free latent-space physics from video.** To reduce reliance on rendering, the LPFV approach (Garcia et al., 2025) enforces physical constraints directly in latent space, removing pixel reconstruction. Related single-video formulations can estimate per-scene parameters but still depend on photometric objectives (Hofherr et al., 2023). Relative to LPFV, our method is intentionally close at the architectural level: both are encoder-only and enforce physics directly in latent space. The main contribution of this paper is therefore the *identifiability* analysis: we provide explicit trajectory-level certificates that determine when continuous-time parameters are uniquely recoverable from video. Methodologically, LPFV uses an Euler/one-sided derivative construction, whereas we use centered finite differences for both first and second derivatives, which reduces the discretization bias in the latent ODE residual from first order to second order in $\Delta t$. LPFV also uses a KL-style anti-collapse term, whereas we use a variance-floor regularizer.

**Trajectory-based scientific machine learning and system identification.** When trajectories (states/derivatives) are observed, scientific machine learning provides powerful tools for discovering and fitting dynamics, including sparse equation discovery (Brunton et al., 2016; Champion et al., 2019), physics-informed inverse problem solvers (Raissi et al., 2019; Karniadakis et al., 2021), universal/neural differential equation frameworks (Chen et al., 2018; Rackauckas et al., 2020), and structure-preserving Lagrangian/Hamiltonian formulations (Greydanus et al., 2019; Zhong et al., 2019;

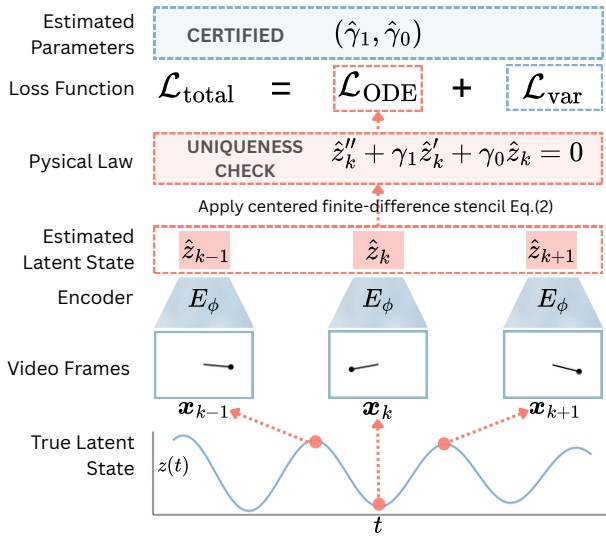

*Figure 1.* **Encoder-only pipeline with identifiability-aware parameter estimation.** A shared encoder $E_\phi$ maps video frames $\boldsymbol{x}_k$ to scalar latents $\hat{z}_k$. Using the centered finite-difference stencil (Eq. (2)), we compute $\hat{z}'_k$ and $\hat{z}''_k$ and enforce the ODE (1) with a residual loss $\mathcal{L}_{\text{ODE}}$. A variance-floor regularizer $\mathcal{L}_{\text{var}}$ prevents latent collapse and improves conditioning, yielding parameter estimates $(\hat{\gamma}_1, \hat{\gamma}_0)$. The *UNIQUENESS CHECK* box denotes the trajectory-level diagnostic from Sec. 4: it tests whether the current clip/window satisfies the coverage condition required for uniqueness. The *CERTIFIED* label means that this diagnostic passes, so the recovered parameters are uniquely determined by the observed video, rather than merely achieving low residual loss.

Lutter et al., 2019; Toth et al., 2019; Cranmer et al., 2020). In parallel, the identifiability of linear ODE systems from discrete samples (including single-trajectory settings) has been studied extensively (Stanhope et al., 2014; Qiu et al., 2022; Wang et al., 2023; 2024a;b). These works typically assume direct access to the state (or a fixed observation model), whereas physics-from-video must infer the state from pixels, introducing latent reparameterization ambiguity. Our coverage-based analysis resolves this ambiguity in an encoder-only setting and yields both exact structural identification and finite-sample guarantees, together with practical single-clip versus multi-trajectory data-collection guidance.

## 3. Problem setting

This section introduces the notation, learning setup, and standing assumptions used throughout.

**Video and encoder.** We observe a video clip $\{\boldsymbol{x}_k\}_{k=0}^{T}$ with frames $\boldsymbol{x}_k \in \mathbb{R}^{w \times h \times c}$ sampled at a known step size $\Delta t > 0$. The absolute time origin $t_0$ is unknown and immaterial for autonomous dynamics; we write $t_k = t_0 + k\Delta t$. We adopt an *encoder-only* pipeline: a deterministic encoder $E_\phi$ maps each frame to a scalar latent $\hat{z}_k := E_\phi(\boldsymbol{x}_k)$, in-

tended to represent the physical state at time $t_k$ (up to an unknown reparameterization). Throughout the paper, the hat marks quantities produced by the encoder or estimated from data, as opposed to the underlying ground truth. The encoder is applied independently to each frame with shared parameters across time, producing a 1D latent $\hat{z}_k$. Our theoretical results do not depend on a particular architecture beyond determinism and sufficient smoothness implied by the state-consistency assumption (Def. 3.1).

**ODE model.** Our analysis considers the homogeneous second-order linear time-invariant (LTI) ODE

$$z''(t) + \gamma_1 z'(t) + \gamma_0 z(t) = 0, \quad (\gamma_1, \gamma_0) \in \mathbb{R}^2. \quad (1)$$

Closed-form solutions across damping regimes are summarized in App. A. A non-homogeneous constant term $g$ in $z'' + \gamma_1 z' + \gamma_0 z + g = 0$ only shifts the equilibrium and does not affect identifiability of $(\gamma_1, \gamma_0)$ for the homogeneous dynamics; we defer this extension to App. E.

**Observation model.** We model each frame as

$$\boldsymbol{x}(t) = \mathcal{X}\big(z(t), \, \sigma(z(t)), \, e(t); \vartheta\big),$$

where $z(t)$ is the physical state, $\sigma(z)$ denotes *state-coupled* nuisances (e.g., shading/visibility varying with state), $e(t)$ collects *exogenous* factors (e.g., background, exposure, illumination), and $\vartheta$ represents time-invariant scene/camera parameters. In our data collection we aim to keep exogenous factors effectively fixed within each clip, so that appearance varies primarily through the state trajectory.

**Definition 3.1** (State consistency). Let $I$ be an observed time interval and define the realized state range $\mathcal{R}_z := z(I)$. We say the video is *state-consistent* on $I$ if there exists an open interval $\mathcal{O} \subset \mathbb{R}$ with $\mathcal{R}_z \subset \mathcal{O}$ and a function $f : \mathcal{O} \to \mathbb{R}$ with $f \in C^2(\mathcal{O})$, i.e., $f$ is twice continuously differentiable on $\mathcal{O}$, such that

$$\hat{z}(t) = f\big(z(t)\big) \quad \text{for all } t \in I,$$

where $\hat{z}(t) := E_\phi(\boldsymbol{x}(t))$.

Def. 3.1 formalizes when the learned coordinate $\hat{z}$ behaves as a smooth reparameterization of the true state on the realized range. All identifiability results reduce to geometric properties of the trajectory under this reparameterization, certified by coverage-type conditions on a state interval $U$ introduced in Sec. 4.

*Remark* 3.2 (Practical guidance). In practice, controlled capture (static camera, fixed exposure/white balance, approximately constant lighting, uniform background) often helps make Def. 3.1 a reasonable modeling assumption.

Def. 3.1 is hard to verify from raw pixels alone, and our theory does not claim arbitrary real video satisfies it. To

probe the practical boundary, we run a synthetic stress test in App. F.4: it keeps the underdamped ODE fixed and perturbs only the observation model with exogenous nuisance (static background noise, moving clutter, camera jitter, dynamic lighting). The finite-sample estimator retains nontrivial robustness to mild-to-moderate violations (e.g., background noise and camera jitter remain accurate across all severities), but stronger time-varying nuisance can break recovery. So the right reading of Def. 3.1 is not all-or-nothing: it is the exact-theorem regime, around which the practical estimator survives modest deviations.

All structural identifiability results assume state consistency (Def. 3.1); our finite-sample analysis additionally allows additive encoder noise around the state-consistent map.

**Encoder-only loss.** Given a clip $\{x_k\}_{k=0}^T$ sampled at step $\Delta t$, the encoder produces latents $\hat{z}_k := E_\phi(x_k)$. We approximate derivatives using a centered finite-difference stencil:

$$\hat{z}_k' = \frac{\hat{z}_{k+1} - \hat{z}_{k-1}}{2\,\Delta t}, \qquad \hat{z}_k'' = \frac{\hat{z}_{k+1} - 2\hat{z}_k + \hat{z}_{k-1}}{\Delta t^2}, \quad (2)$$

for $k = 1, \ldots, T-1$. Given coefficients $(\gamma_1, \gamma_0)$, define the discrete ODE residual

$$r_k(\gamma_1, \gamma_0) := \hat{z}_k'' + \gamma_1\,\hat{z}_k' + \gamma_0\,\hat{z}_k, \qquad (3)$$

and the data-fitting loss

$$\mathcal{L}_{\mathrm{ODE}}(\phi, \gamma_1, \gamma_0) := \frac{1}{T-1} \sum_{k=1}^{T-1} r_k(\gamma_1, \gamma_0)^2. \qquad (4)$$

**Variance floor regularizer.** To prevent latent collapse, we impose a lower bound on the empirical standard deviation of $\{\hat{z}_k\}_{k=0}^T$. Define the empirical mean and variance

$$\widehat{\mu} := \frac{1}{T+1} \sum_{k=0}^T \hat{z}_k, \quad \widehat{\mathrm{Var}}(\hat{z}) := \frac{1}{T+1} \sum_{k=0}^T (\hat{z}_k - \widehat{\mu})^2.$$

Given a target standard deviation $\tau > 0$, we penalize deviation below $\tau$ via

$$\mathcal{L}_{\mathrm{var}}(\hat{z}_{0:T}) := \left( \max\left\{ 0,\ \tau - \sqrt{\widehat{\mathrm{Var}}(\hat{z}) + \varepsilon} \right\} \right)^2, \quad (5)$$

where $\varepsilon > 0$ is a small constant for numerical stability (we use $\varepsilon = 10^{-8}$). The variance floor regularizer serves a dual purpose. Empirically, it prevents latent collapse to the trivial solution $\hat{z}_k \equiv 0$, which attains zero ODE residual under $\mathcal{L}_{\mathrm{ODE}}$. Theoretically, it is essential for our finite-sample analysis: Lem. D.1 shows that a nontrivial lower bound on $\widehat{\mathrm{Var}}(\hat{z})$ yields a conditioning parameter $\psi_{\min} > 0$ for the associated design matrix.

**Total objective.** Our encoder-only training objective is

$$\mathcal{L}_{\mathrm{total}}(\phi, \gamma_1, \gamma_0) := \mathcal{L}_{\mathrm{ODE}}(\phi, \gamma_1, \gamma_0) + \lambda_{\mathrm{var}}\,\mathcal{L}_{\mathrm{var}}(\hat{z}_{0:T}), \quad (6)$$

where $\lambda_{\mathrm{var}} > 0$ controls the variance floor penalty. An overview of our method is shown in Fig. 1.

**Multi-trajectory extension (shared physics).** For $M$ clips $\{x_k^{(m)}\}_{k=0}^{T^{(m)}}$ encoded by a shared $E_\phi$ and sharing parameters $(\gamma_1, \gamma_0)$, define latents $\hat{z}_k^{(m)} := E_\phi(x_k^{(m)})$ and residuals $r_k^{(m)}$ as in (3), with derivatives computed by the centered stencil (2) in each clip. Let $N := \sum_{m=1}^M (T^{(m)} - 1)$ be the total number of interior indices. We minimize

$$\mathcal{L}_{\mathrm{total}}^{\mathrm{multi}}(\phi, \gamma_1, \gamma_0) = \frac{1}{N} \sum_{m=1}^M \sum_{k=1}^{T^{(m)}-1} \left( r_k^{(m)}(\gamma_1, \gamma_0) \right)^2$$
$$+ \frac{\lambda_{\mathrm{var}}}{M} \sum_{m=1}^M \left( \max\left\{ 0,\ \tau - \sqrt{\widehat{\mathrm{Var}}(\hat{z}^{(m)}) + \varepsilon} \right\} \right)^2, \qquad (7)$$

where $\widehat{\mathrm{Var}}(\hat{z}^{(m)})$ is computed over $k = 0, \ldots, T^{(m)}$ in clip $m$. In the multi-clip setting, we assume Def. 3.1 holds for each clip and that the induced reparameterization $f$ is *shared* across clips on the union of their state ranges.

## 4. Main theoretical results

In this section we present our main theoretical results and adopt the following convention to streamline notation.

**Ideal continuous and noiseless regime (structural identifiability).** When studying *structural* properties (Thms. 4.2–4.6), we conceptually work with continuous-time, noiseless latents and write

$$\hat{z}(t) = f\big(z(t)\big), \qquad t \in [t_0, t_0 + L].$$

Here the hat is a placeholder for "the latent produced by an encoder"; in the ideal regime it coincides with the deterministic map $f \circ z$. Under this regime, exact derivatives and ODE equalities are used to derive identifiability results.

**Practical discrete and noisy regime (finite-sample analysis).** In finite-sample analysis (Thms. 4.8 and B.3), we work with the actually observed, discrete sequence

$$\hat{z}_k = f(z_k) + \epsilon_k, \qquad k = 0, 1, \ldots, T,$$

sampled with step $\Delta t$ and differentiated by the centered stencil in (2). Here $\epsilon_k$ captures random fluctuations from nuisances/sensors and is modeled as zero-mean sub-Gaussian (temporal $\beta$-mixing allowed).

When $\epsilon_k \equiv 0$ and $\Delta t \to 0$ under level-set slope coverage, the finite-sample conclusions reduce to the structural ones.

## 4.1. Identifiability with continuous-time observations

### 4.1.1. A GEOMETRIC COVERAGE CONDITION

**Definition 4.1** (Level-set slope coverage). Let $U \subset \mathcal{R}_z$ be a nonempty open interval. We say that a trajectory $z(\cdot)$ has *level-set slope coverage* on $U$ if, for every $u \in U$, there exist at least three times $t_1(u), t_2(u), t_3(u)$ such that $z(t_i(u)) = u$ for $i = 1, 2, 3$, and the corresponding time-slopes are pairwise distinct:

$$z'(t_1(u)), \; z'(t_2(u)), \; z'(t_3(u)) \text{ are all different.}$$

Equivalently, each level $u \in U$ is attained at least three times with three distinct instantaneous velocities.

### 4.1.2. SINGLE-TRAJECTORY IDENTIFIABILITY

**Theorem 4.2** (Single-trajectory identifiability). *Assume state consistency on $I$, i.e., $\hat{z}(t) = f(z(t))$ with $f \in C^2$ on the realized range. Suppose the true state $z$ satisfies the homogeneous second-order linear ODE (1) on $I$. Assume further that the learned latent $\hat{z}$ satisfies a homogeneous second-order linear ODE*

$$\hat{z}''(t) + \eta_1 \hat{z}'(t) + \eta_0 \hat{z}(t) = 0 \quad on \; I. \tag{8}$$

*If level-set slope coverage holds on a nonempty open interval $U \subset z(I)$ and $f' \not\equiv 0$ on $U$, then $f$ is affine on $U$ and the parameters are identified: $(\eta_1, \eta_0) = (\gamma_1, \gamma_0)$.*

A detailed proof of Thm. 4.2 appears in App. C.1. Thm. 4.2 highlights that identifiability is driven by trajectory geometry: once each level $u \in U$ is attained with three distinct time-slopes, the only $C^2$ reparameterizations consistent with both ODEs are affine, forcing the physical parameters to align. This perspective lets us replace hard-to-verify model constraints with verifiable data coverage conditions.

### 4.1.3. COVERAGE HOLD: UNDERDAMPED DYNAMICS

We next give simple dynamical conditions under which level-set slope coverage holds.

**Theorem 4.3** (Underdamped single-trajectory sufficiency). *Consider a nontrivial solution $z$ of ODE (1) on $I$ (i.e., $z$ is not identically zero) and assume the strictly underdamped regime $\gamma_1 > 0$ and $\gamma_1^2 < 4\gamma_0$. Let $\zeta := \gamma_1/2$, $\omega := \sqrt{\gamma_0 - \zeta^2}$, and $P := 2\pi/\omega$. For any time window $[t_a, t_b] \subset I$ of length $L := t_b - t_a \geq 2P$, there exists a nonempty open interval $U$ around the equilibrium such that level-set slope coverage holds on $U$ within $[t_a, t_b]$. Consequently, under the latent assumptions of Thm. 4.2, the parameters $(\gamma_1, \gamma_0)$ are identifiable from this single trajectory.*

A detailed proof of Thm. 4.3 appears in App. C.2.

*Remark* 4.4 (Sufficiency vs. necessity of the window length). The condition $L \geq 2P$ in Thm. 4.3 ensures (independent

of the unknown initial phase) that each attainable level near equilibrium is crossed at least three times within the window, so level-set slope coverage holds on some nonempty open interval $U$. It is *not necessary*: identification follows as soon as there exists a nonempty open interval $U$ such that every $u \in U$ is attained at least three times with pairwise distinct derivatives, which may occur even when $P < L < 2P$ under favorable phase alignment.

### 4.1.4. COVERAGE FAIL: NON-OSCILLATORY AND UNDAMPED REGIMES

Outside the strictly underdamped regime, the required slope diversity is absent, hence single-trajectory coverage fails.

**Theorem 4.5** (Single-trajectory insufficiency in non-oscillatory and undamped regimes). *Consider ODE (1).*

***Critical/overdamped*** ($\gamma_1^2 \geq 4\gamma_0$). *For any solution $z$ of (1) on $I$, either $z$ is constant (equivalently, $z'(t) \equiv 0$), in which case level-set slope coverage fails trivially, or else $z(t)$ has at most one critical point. Consequently, if $z$ is not constant, then for any fixed level $u \in \mathbb{R}$, the equation $z(t) = u$ has at most two solutions in $t$. Therefore level-set slope coverage fails for a single trajectory.*

***Undamped*** ($\gamma_1 = 0$, $\gamma_0 > 0$). *For any nontrivial trajectory and any level $u$ with $|u|$ smaller than the amplitude, the set $\{t : z(t) = u\}$ is infinite, but the corresponding derivatives take only two values, $\pm\omega\sqrt{A^2 - u^2}$ (where $\omega = \sqrt{\gamma_0}$ and $A$ is the amplitude). Thus level-set slope coverage fails for a single trajectory.*

*Consequently, in these regimes the parameters $(\gamma_1, \gamma_0)$ are not identifiable from one trajectory via Thm. 4.2 in general.*

A detailed proof of Thm. 4.5 appears in App. C.3. This lack of identifiability is structural rather than statistical: the required slope diversity is absent by construction. In practice, this motivates combining multiple trajectories so that distinct velocities occur at the same state level.

### 4.1.5. MULTI-TRAJECTORY IDENTIFIABILITY

Replacing "three crossings on one path" with "three slopes across three paths" yields the same algebraic elimination.

**Theorem 4.6** (Three trajectories suffice). *Consider ODE (1) and let $z^{(m)}$ ($m = 1, 2, 3$) be three solutions. Assume there exists a nonempty open interval $U \subset \mathcal{R}_z$ such that for every $u \in U$ there are times $t_m(u)$ with $z^{(m)}(t_m(u)) = u$, and $v_m(u) := z^{(m)'}(t_m(u))$ pairwise distinct for m=1,2,3. Assume a shared encoder $f \in C^2(U)$ and shared latent coefficients $(\eta_1, \eta_0)$ such that, for each $m = 1, 2, 3$ and all $t$ with $z^{(m)}(t) \in U$,*

$$\hat{z}^{(m)}(t) := f(z^{(m)}(t)),$$
$$\hat{z}^{(m)''}(t) + \eta_1 \hat{z}^{(m)'}(t) + \eta_0 \hat{z}^{(m)}(t) = 0.$$

If $f' \not\equiv 0$, then $f$ is affine on $U$ and $(\eta_1, \eta_0) = (\gamma_1, \gamma_0)$.

A detailed proof of Thm. 4.6 appears in App. C.5. Diversity across trajectories substitutes for long observation windows: if each $u \in U$ is realized with three distinct velocities across the dataset, then Thm. 4.6 yields the same identifiability conclusion as Thm. 4.2.

## 4.2. Finite-sample convergence with discrete-time observations

We bound the estimation error incurred when fitting the latent ODE from discrete-time latents using the residuals (3) computed via centered differences (2).

### 4.2.1. SETUP AND PROOF STRATEGY

**A noiseless discrete proxy (bridge only).** To cleanly separate sampling/noise effects from deterministic encoder mismatch, we introduce a noiseless proxy sequence

$$\tilde{z}_k := f\big(z(t_k)\big), \qquad t_k = t_0 + k\,\Delta t,$$

and denote by $\tilde{z}'_k$ and $\tilde{z}''_k$ the *true* time derivatives of the continuous-time proxy $\tilde{z}(t) := f(z(t))$, evaluated at $t = t_k$. We assume $\tilde{z}(t)$ is sufficiently smooth on $I$ so that the centered stencil is second-order accurate.

The observed latents satisfy

$$\hat{z}_k = \tilde{z}_k + \epsilon_k, \qquad \mathbb{E}[\epsilon_k \mid z_k] = 0.$$

We use $\tilde{z}$ only within the proof to separate (i) statistical error from $\{\epsilon_k\}$ and discretization, and (ii) deterministic mismatch captured by $E_{\text{enc}}$. Outside this bridge we write $\hat{z}$ throughout to avoid symbol proliferation.

**Definition 4.7** (Pseudotrue parameter and encoder approximation error). Let $\tilde{X}_k := [\,\tilde{z}'_k,\ \tilde{z}_k\,]$ and $\tilde{y}_k := -\tilde{z}''_k$. Define the (empirical, noiseless) *pseudotrue parameter*

$$\eta^\star \in \arg\min_{\eta \in \mathbb{R}^2}\ \frac{1}{T-1}\sum_{k=1}^{T-1}\big(\tilde{y}_k - \tilde{X}_k\eta\big)^2,$$

and the *encoder-induced approximation error*

$$E_{\text{enc}} := \|\eta^\star - \gamma\|_2\,,$$

where $\gamma = (\gamma_1, \gamma_0)$. If $f$ is affine on the coverage interval $U$, then our structural results imply $\eta^\star = \gamma$ and $E_{\text{enc}} = 0$. In general, $E_{\text{enc}}$ is deterministic model mismatch (App. B.1).

### 4.2.2. SINGLE-CLIP BOUND

**Theorem 4.8** (Finite-sample error). *Let $\hat{z}_k = E_\phi(\boldsymbol{x}_k)$ be the latents of a single clip and define discrete derivatives by (2). Let the ODE residual be $r_k(\eta_1, \eta_0)$ as in (3), and let $\hat{\eta} = (\hat{\eta}_1, \hat{\eta}_0)$ minimize (4) (equivalently, it minimizes $\mathcal{L}_{\text{total}}(\phi, \eta)$*

since $\mathcal{L}_{\text{var}}$ is $\eta$-independent). Assume the chosen window satisfies level-set slope coverage (Def. 4.1). Suppose

$$\hat{z}_k = f(z_k) + \epsilon_k,$$
$$f(u) = au + b + h(u),\ a \neq 0,\ b = O(1),\ h \in C^2(U),$$

*where $\{\epsilon_k\}$ are zero-mean sub-Gaussian (temporal $\beta$-mixing allowed) with noise scale $\sigma_0$. Assume $\Delta t \leq 1$, and let $\sigma = \sigma_{\text{fd}}(\Delta t)$ be the corresponding effective sub-Gaussian noise scale after applying the finite-difference stencils in (2) (see (19)). Let $X_k = [\,\hat{z}'_k,\ \hat{z}_k\,]$ and*

$$\psi_{\min} := \lambda_{\min}\Big(\frac{1}{T-1}\sum_{k=1}^{T-1} X_k^\top X_k\Big) > 0.$$

*Let $E_{\text{enc}}$ be as in Def. 4.7. Then for any $\delta \in (0,1)$, with probability at least $1 - \delta$,*

$$\|\hat{\eta} - \gamma\|_2 \leq \frac{C_1\sigma}{\psi_{\min}}\sqrt{\frac{\log(3/\delta)}{T-1}} + \frac{C_2\sigma^2}{\psi_{\min}} + \frac{C_3\Delta t^{\,2}}{\psi_{\min}} + E_{\text{enc}}.$$
$$(9)$$

*Here $\lambda_{\min}(\cdot)$ denotes smallest eigenvalue. The constants depend only on bounded moments of $\{z_k\}$, bounds on the relevant derivatives of $\tilde{z}(t) = f(z(t))$ over the window, and the fixed stencil coefficients in (2). The noise scale $\sigma = \sigma_{\text{fd}}(\Delta t)$ is an effective (post-stencil) sub-Gaussian scale and may increase as $\Delta t$ decreases due to finite-difference amplification. The term $C_3\Delta t^{\,2}$ is the centered-stencil discretization bias. Under slope coverage and a latent variance floor, $\psi_{\min}$ is guaranteed to be bounded away from zero; App. D (Lem. D.1) provides a practical, checkable lower bound.*

The proof appears in App. C.6. The bound in (9) tightens as (i) the trajectory revisits the same latent levels with more distinct velocities, quantified by a larger $\psi_{\min}$, and (ii) the sample size $T$ increases. Read the bound in App. B.3.

**Multi-clip bound.** Pooling $M$ clips that share the same physics $(\gamma_1, \gamma_0)$ increases the effective sample size to $N = \sum_{m=1}^{M}(T^{(m)} - 1)$ and can improve conditioning $\psi_{\min}$ through cross-trajectory slope diversity. The full multi-clip finite-sample guarantee is stated in App. B.2.

## 5. Experiments

We instantiate the identifiability theory of Sec. 4 using a shared per-frame CNN encoder across all synthetic and real-world experiments. Unless otherwise stated, we set the variance-floor regularization weight to $\lambda_{\text{var}} = 1.0$, use $\tau = 1$, and initialize the ODE parameters as $(\gamma_1, \gamma_0) = (1, 1)$. Synthetic experiments and the wheel-mounted-phone pendulum are reported as mean±std over 5 random seeds. For the clean pendulum benchmark, we follow the 5-video protocol of (Garcia et al., 2025) and report both the original

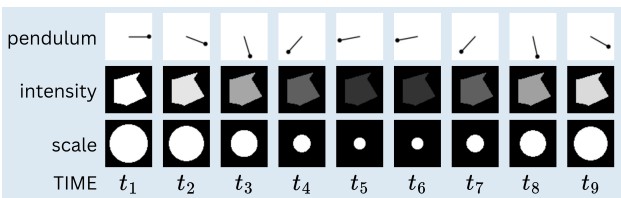

*Figure 2.* Example frames in the underdamped regime.

mean±std length estimates and string-length RMSE. Detailed architecture, learning rates, and benchmark-specific settings appear in App. F.

## 5.1. Simulations

All synthetic systems render video frames from a scalar state $z(t)$ governed by ODE (1) under four damping regimes. Tab. 1 lists the ground-truth used throughout. Example frames are shown in Fig. 2.

*Table 1.* Ground-truth parameters used in simulations. Under = underdamped, Un = undamped, Cri = critical, Over = overdamped.

|  | UNDER | UN | CRI | OVER |
|---|---|---|---|---|
| $\gamma_0$ | 4.0016 | 4 | 4 | 4 |
| $\gamma_1$ | 0.08 | 0 | 4 | 5 |

### 5.1.1. MOTION DYNAMICS: PENDULUM SYSTEMS

**Single trajectory across regimes.** Thms. 4.2, 4.3, and 4.5 imply that single-trajectory identification hinges on whether level-set slope coverage is realized within the clip. Fig. 3 overlays parameter recovery (top) with a numerical continuous-time coverage check on the true state trajectory (bottom). In the underdamped regime, the first coverage-positive window appears at $1.1\pi$ (star), coinciding with the onset of accurate recovery. This does not contradict Thm. 4.3, since the $2P$ condition is sufficient rather than necessary. In the critical and overdamped regimes, no single window is coverage-positive and estimates remain unreliable. In the ideal continuous undamped regime, no single window is coverage-positive, consistent with the structural non-identifiability result of Thm. 4.5. This still does not preclude accurate recovery in the top row, because the actual estimator does not fit the continuous-time problem directly; it fits a centered-difference regression to sampled latents. For an ideal sampled undamped sinusoid, Prop. C.1 shows that this discrete regression has a unique zero-loss target

$$(\gamma_1^\Delta, \gamma_0^\Delta) = (0, \gamma_0^\Delta), \qquad \gamma_0^\Delta = \gamma_0 + O(\Delta t^2),$$

whenever the corresponding discrete design matrix has full column rank. Therefore, when $\Delta t$ is small and the sampled window spans enough phase to make the discrete regression well conditioned, the estimator can stabilize around a parameter pair that is very close to $(0, \gamma_0)$. This explains why long

*Table 2.* Pendulum: estimation with three trajectories in the non-oscillatory regimes. Ground truth is $(\gamma_0, \gamma_1) = (4, 4)$ for CRI and $(4, 5)$ for OVER. TRUE/FALSE indicates whether the selected three trajectories satisfy the cross-trajectory coverage condition.

| REGIME | COVERAGE | $\hat{\gamma}_0$ | $\hat{\gamma}_1$ |
|---|---|---|---|
| CRI | TRUE | $4.0218 \pm 0.0040$ | $4.0352 \pm 0.0017$ |
|  | FALSE | $2.3462 \pm 0.0437$ | $2.7642 \pm 0.0787$ |
| OVER | TRUE | $3.9733 \pm 0.0033$ | $4.9723 \pm 0.0010$ |
|  | FALSE | $2.6314 \pm 0.0080$ | $4.0396 \pm 0.0095$ |

undamped clips can yield accurate estimates in the finite-sample pipeline without contradicting the continuous-time structural result. Additional sweeps over initial conditions are in App. F.5 (Tabs. 7, 8, 9, 10).

**Multi-trajectory in non-oscillatory regimes.** When single-trajectory coverage fails, Thm. 4.6 shows that cross-trajectory diversity can substitute for long windows: under state consistency, three trajectories that reach a common level with three distinct instantaneous velocities guarantee identifiability. We test this in the critical and overdamped regimes by jointly fitting a shared $(\gamma_0, \gamma_1)$ across three clips. Tab. 2 shows accurate recovery when cross-trajectory coverage holds and unstable estimates otherwise; broader sweeps appear in Tabs. 11 and 12.

### 5.1.2. NON-MOTION DYNAMICS: INTENSITY AND SCALE SYSTEMS

To show that our identifiability results are not specific to geometric motion, we also evaluate two *non-motion* video generators that follow the same latent ODE: (i) **intensity**, where the state controls the grayscale intensity of an irregular shape (normalized to $z \in [0.2, 1.0]$); (ii) **scale**, where the state controls the radius of a filled circle (normalized to $z \in [-10, 10]$). Example frames are shown in Fig. 2.

Tab. 3 reports representative parameter estimates across damping regimes for both systems. Whenever the theorem conditions for identifiability are satisfied, the recovered parameters closely match the ground truth, demonstrating that the encoder-only pipeline and the accompanying theory extend beyond motion cues. Additional sweeps over initial conditions are provided in App. F.5 (Tabs. 13 and 14).

### 5.1.3. REGULARIZATION AND ROBUSTNESS

**Avoiding latent collapse.** Because the ODE residual objective alone admits the trivial collapsed solution $\hat{z}_k \equiv 0$, regularization is required in practice. Tab. 4 compares three choices: (i) no regularizer, (ii) a KL-to-$\mathcal{N}(0, 1)$ regularizer as in Garcia et al. (2025), and (iii) our variance-floor regularizer. Without regularization, estimates are consistently poor across regimes. In the oscillatory regimes (un-

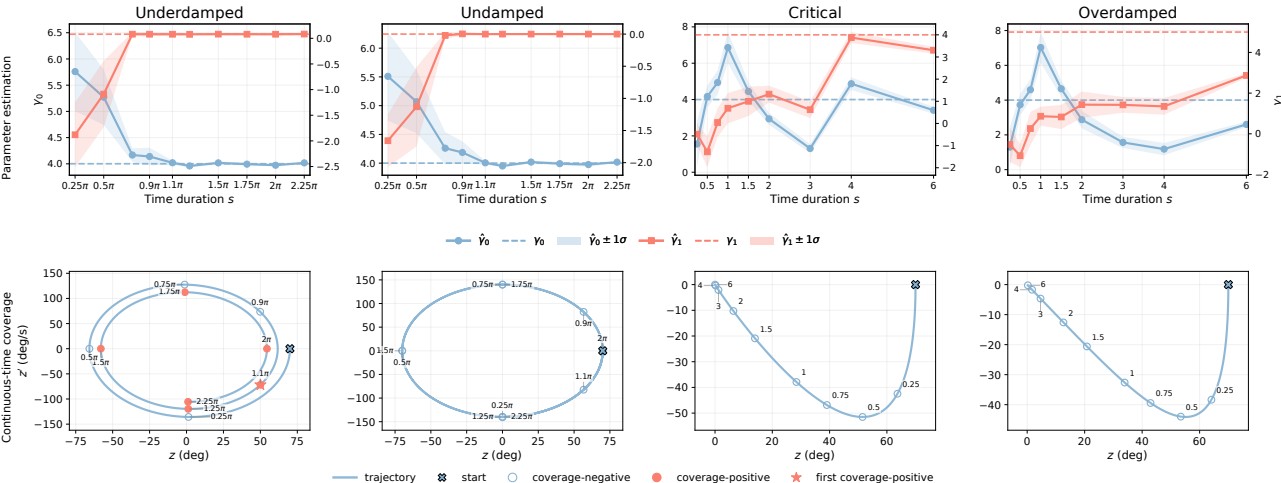

*Figure 3.* Single-trajectory pendulum results across four damping regimes. Top: recovered parameters versus tested video length (mean $\pm 1\sigma$ over 5 seeds; dashed lines show the ground-truth values). The left-side $\gamma_0$ and right-side $\gamma_1$ tick values are shown in all four panels for readability. Bottom: phase portraits of the true continuous-time state trajectory in the $(z, z')$ plane for the same initial condition. The blue curve traces $(z(t), z'(t))$ from $t = 0$. The "x" marker denotes the start of the trajectory. Each labeled endpoint corresponds to one tested video length. Hollow blue circles denote coverage-negative endpoints, filled red circles denote coverage-positive endpoints, and the red star marks the first coverage-positive endpoint.

*Table 3.* Estimated parameters for the intensity and scale systems across damping regimes (with coverage condition). Ground truth by regime is UNDER $(\gamma_0, \gamma_1) = (4.0016, 0.08)$, UN $(4, 0)$, CRI $(4, 4)$, and OVER $(4, 5)$.

|  | REGIME | $\hat{\gamma}_0$ | $\hat{\gamma}_1$ |
|---|---|---|---|
| **INTENSITY** | UNDER | $4.007 \pm 0.007$ | $0.088 \pm 0.004$ |
|  | UN | $3.970 \pm 0.001$ | $0.002 \pm 0.004$ |
|  | CRI | $4.031 \pm 0.022$ | $3.964 \pm 0.012$ |
|  | OVER | $3.962 \pm 0.030$ | $4.968 \pm 0.059$ |
| **SCALE** | UNDER | $4.020 \pm 0.0005$ | $0.074 \pm 0.0003$ |
|  | UN | $3.983 \pm 0.0003$ | $4 \times 10^{-5} \pm 0.0001$ |
|  | CRI | $4.025 \pm 0.0001$ | $4.074 \pm 0.0002$ |
|  | OVER | $4.128 \pm 0.0001$ | $4.918 \pm 0.0001$ |

*Table 4.* Ablation study of loss functions (pendulum). Var-regu denotes the variance-floor regularized loss proposed in this paper. Ground truth by regime is UNDER $(\gamma_0, \gamma_1) = (4.0016, 0.08)$, UN $(4, 0)$, CRI $(4, 4)$, and OVER $(4, 5)$.

| REGIME | LOSS | $\hat{\gamma}_0$ | $\hat{\gamma}_1$ |
|---|---|---|---|
| UNDER | NO-REGU | $-0.0004 \pm 0.0009$ | $0.0425 \pm 0.1194$ |
|  | KL-REGU | $4.0037 \pm 0.0009$ | $0.0798 \pm 0.0003$ |
|  | VAR-REGU | $4.0037 \pm 0.0003$ | $0.0800 \pm 0.0003$ |
| UN | NO-REGU | $0.0582 \pm 0.1627$ | $-0.1144 \pm 0.1764$ |
|  | KL-REGU | $4.0031 \pm 0.0006$ | $0.0005 \pm 0.0006$ |
|  | VAR-REGU | $4.0032 \pm 0.0010$ | $0.0002 \pm 0.0004$ |
| CRI | NO-REGU | $0.0013 \pm 0.0018$ | $2.2204 \pm 0.7833$ |
|  | KL-REGU | $9.2659 \pm 1.0532$ | $6.2857 \pm 0.4848$ |
|  | VAR-REGU | $4.0218 \pm 0.0040$ | $4.0352 \pm 0.0017$ |
| OVER | NO-REGU | $0.0048 \pm 0.0108$ | $2.5409 \pm 0.4765$ |
|  | KL-REGU | $6.8248 \pm 0.1909$ | $7.3831 \pm 0.0906$ |
|  | VAR-REGU | $3.9733 \pm 0.0033$ | $4.9723 \pm 0.0010$ |

derdamped/undamped), both KL and variance-floor regularizers yield accurate recovery. In the critical/overdamped regimes, KL regularization fails while the variance-floor regularizer recovers the correct parameters.

A plausible explanation is that KL-to-$\mathcal{N}(0, 1)$ regularization encourages the latent marginal to match a fixed reference distribution, which can conflict with the strongly non-Gaussian, nonstationary, and often low-variance trajectories encountered in non-oscillatory regimes. This mismatch can distort the learned representation and degrade the effective regression design used to estimate $(\gamma_0, \gamma_1)$. In contrast, the variance-floor regularizer enforces only a minimal scale for $\{\hat{z}_k\}$ while otherwise leaving the trajectory shape unconstrained, directly addressing the collapse degeneracy and aligning with our finite-sample conditioning analysis.

**Sensitivity to the variance-floor target $\tau$.** Varying the target latent standard deviation $\tau$ yields nearly identical parameter estimates across damping regimes, indicating low sensitivity. This is expected: rescaling the latent trajectory does not affect the coefficients of a homogeneous linear ODE, so $\tau$ primarily sets the latent scale rather than identifiability. See App. F.5 (Tab. 15) for the full results.

## 5.2. Real-world experiments

We complement the synthetic studies with two real videos to test the robustness of our decoder-free, single-trajectory identification pipeline as visual nuisance increases.

*Table 5.* Real-video pendulum: string-length recovery. For each ground-truth length, we report the original string-length estimate (mean±std over 5 clips) and the corresponding RMSE (m).

| METHOD | $L^\star = 0.45\,\text{M}$ | | $L^\star = 0.90\,\text{M}$ | | $L^\star = 1.50\,\text{M}$ | |
| | ESTIMATE | RMSE | ESTIMATE | RMSE | ESTIMATE | RMSE |
|---|---|---|---|---|---|---|
| PAIG | $1.01 \pm 0.03$ | 0.561 | $1.01 \pm 0.04$ | 0.116 | $1.01 \pm 0.04$ | 0.491 |
| NIRPI | $0.77 \pm 0.33$ | 0.435 | $0.84 \pm 0.53$ | 0.478 | $0.63 \pm 0.38$ | 0.934 |
| LPFV | $0.51 \pm 0.01$ | 0.061 | $1.07 \pm 0.20$ | 0.247 | $1.30 \pm 0.02$ | 0.201 |
| OURS | $0.50 \pm 0.001$ | **0.050** | $0.97 \pm 0.003$ | **0.070** | $1.60 \pm 0.015$ | **0.101** |

### 5.2.1. CLEAN PENDULUM BENCHMARK

**Setup.** We follow the real-video pendulum benchmark and protocol of (Garcia et al., 2025). Baseline results (PAIG (Jaques et al., 2020), NIRPI (Hofherr et al., 2023), LPFV (Garcia et al., 2025)) are taken from (Garcia et al., 2025) Tab. 2(a); only OURS is computed by our implementation. The dataset contains three string lengths ($L^\star = 0.45/0.90/1.50\,\text{m}$), filmed with a static camera and clean background (App. Fig. 5). For each length, five clips with different initial conditions are provided. To present the results in a more standard error-based form, we additionally report root mean square error (RMSE) in string length. Since the published baselines are available as mean±std string-length estimates over the five benchmark clips, we compute

$$\text{RMSE} = \sqrt{\frac{1}{n} \sum_{i=1}^{n} (\hat{L}_i - L^\star)^2} = \sqrt{(\bar{L} - L^\star)^2 + \frac{n-1}{n} s^2},$$

where $n = 5$, $\bar{L}$ and $s$ are the reported mean and sample standard deviation, respectively.

**Results.** Tab. 5 reports string-length RMSE. OURS achieves the lowest RMSE for all three lengths. PAIG appears competitive at $L^\star = 0.90\,\text{m}$, but Tab. 5 shows that it predicts nearly the same length ($\approx 1.01\,\text{m}$) for all three pendulums, so its middle-length score is largely incidental rather than evidence of reliable recovery. Overall, these results are consistent with our underdamped theory: sufficiently exciting single trajectories can support accurate recovery when the learned latent follows the correct dynamics. Compared with LPFV, the lower RMSE of our method is consistent with the use of a higher-accuracy derivative discretization in the discrete ODE residual (App. F.3.1).

### 5.2.2. WHEEL-MOUNTED PHONE PENDULUM

**Setup.** We next consider a more challenging real capture where a smartphone mounted near a bicycle wheel rim swings in a vertical plane (App. Fig. 6), from Monteiro et al. (2014). Relative to the clean benchmark, this video includes substantial nuisance variation (clutter, partial occlusions, and appearance changes). We therefore run our method on

a cropped/resized version (second row of App. Fig. 6). This preprocessing should be read as a step that moves the clip closer to the modeling regime of Def. 3.1 (see App. F.4 for a synthetic probe of practical tolerance), not as evidence that the theory itself handles arbitrary nuisance. Assuming small oscillations, we estimate the ODE parameters in Eq. (1). We compute a physics-based ground truth for $\gamma_0(= 8.26)$ from geometry and inertia (App. Sec. F.3.2); $\gamma_1$ is not uniquely determined from geometry and is omitted.

**Results.** Our estimated $\hat{\gamma}_0 = 8.998 \pm 0.009$, which is close in magnitude to the physics-based value and stable across 5 trials. The residual gap is expected given approximate small-angle/effective-damping assumptions and pixel-based finite-difference derivatives under heavy nuisance.

## 6. Conclusion

We studied decoder-free physical parameter identification from video in an encoder-only pipeline where a scalar latent is constrained to satisfy a homogeneous second-order linear ODE. Under a state-consistent observation model, we introduced the geometric *level-set slope coverage* condition and showed that it forces the latent reparameterization to be locally affine on an open set, enabling exact recovery of continuous-time parameters from pixels. We further derived finite-sample guarantees for discrete-time training that expose conditioning-driven error amplification, motivating a variance-floor regularizer that prevents latent collapse and improves numerical stability.

A limitation is that our analysis focuses on one-dimensional latents and homogeneous second-order LTI dynamics. While this setting enables sharp identifiability certificates, extending the theory is an important direction. Promising next steps include (i) extending coverage-based certificates to *multi-dimensional* latents and coupled/multi-mode linear systems, and (ii) handling *forced* dynamics (unknown inputs) and mild departures from LTI (time variation or weak nonlinearity). On the deployment side, our results suggest capture- and learning-time protocols that promote coverage, enabling reliable video-based estimation of frequency and damping in applications such as construction monitoring and infrastructure vibration assessment.

## Acknowledgements

KZ would like to acknowledge the support from NSF Award No. 2229881, AI Institute for Societal Decision Making (AI-SDM), the National Institutes of Health (NIH) under Contract R01HL159805, and grants from Quris AI, Florin Court Capital, MBZUAI-WIS Joint Program, and the Al Deira Causal Education project. MG was supported by ARC DP240102088 and WIS-MBZUAI 142571.

## Impact Statement

This paper presents work whose goal is to advance the field of Machine Learning. There are many potential societal consequences of our work, none of which we feel must be specifically highlighted here.

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

## A. Solutions of the Second-Order Homogeneous LTI ODE

This appendix summarizes closed-form solutions of the target ODE used throughout the paper,

$$z''(t) + \gamma_1 z'(t) + \gamma_0 z(t) = 0, \tag{10}$$

under the standing assumption $\gamma_0 > 0$ (oscillator stiffness) and $\gamma_1 \geq 0$ (damping). We use the same notation as in the main text: $(\gamma_0, \gamma_1)$ denote the true continuous-time parameters, and $z(t)$ is the scalar latent/state. We also use the discriminant

$$D := \gamma_1^2 - 4\gamma_0,$$

and define the damping rate $\zeta := \gamma_1/2$.

**Characteristic equation.** Eq. (10) has characteristic polynomial

$$r^2 + \gamma_1 r + \gamma_0 = 0,$$

with roots

$$r_{1,2} = \frac{-\gamma_1 \pm \sqrt{D}}{2}.$$

The qualitative behavior of solutions is determined by the sign of $D$, corresponding to the standard damping regimes below.

### A.1. Underdamped regime ($D < 0$)

When $D < 0$ (equivalently $\gamma_1^2 < 4\gamma_0$), the roots are complex conjugates

$$r_{1,2} = -\zeta \pm i\omega, \qquad \omega := \sqrt{\gamma_0 - \zeta^2}.$$

The general solution is

$$z(t) = e^{-\zeta t}\Big(A\cos(\omega t) + B\sin(\omega t)\Big),$$

for constants $(A, B) \in \mathbb{R}^2$ determined by initial conditions. Given $z(0) = z_0$ and $z'(0) = v_0$, we obtain

$$A = z_0, \qquad B = \frac{v_0 + \zeta z_0}{\omega}.$$

**Period and amplitude decay.** The oscillation period is

$$P = \frac{2\pi}{\omega},$$

and the envelope decays as $e^{-\zeta t}$.

### A.2. Undamped regime ($\gamma_1 = 0$)

The undamped case is the special subcase of §A.1 with $\gamma_1 = 0$, hence $\zeta = 0$ and $\omega = \sqrt{\gamma_0}$. The general solution reduces to a pure sinusoid:

$$z(t) = A\cos(\omega t) + B\sin(\omega t), \qquad \omega = \sqrt{\gamma_0}.$$

Under initial conditions $z(0) = z_0$ and $z'(0) = v_0$, we have

$$A = z_0, \qquad B = \frac{v_0}{\omega}.$$

The period is $P = 2\pi/\omega$, and the amplitude is constant over time.

### A.3. Critically damped regime ($D = 0$)

When $D = 0$ (equivalently $\gamma_1^2 = 4\gamma_0$), the characteristic polynomial has a repeated real root $r = -\zeta = -\gamma_1/2$. The general solution is

$$z(t) = (A + Bt)e^{-\zeta t}, \qquad \zeta = \gamma_1/2.$$

Given $z(0) = z_0$ and $z'(0) = v_0$, we obtain

$$A = z_0, \qquad B = v_0 + \zeta z_0.$$

Solutions do not oscillate and decay to zero at the fastest rate among all non-oscillatory regimes.

### A.4. Overdamped regime ($D > 0$)

When $D > 0$ (equivalently $\gamma_1^2 > 4\gamma_0$), the roots $r_1 > r_2$ in (A) are distinct and real, both negative for $\gamma_0 > 0, \gamma_1 > 0$. The general solution is a sum of two exponentials:

$$z(t) = C_1 e^{r_1 t} + C_2 e^{r_2 t}.$$

Given $z(0) = z_0$ and $z'(0) = v_0$, the coefficients are

$$C_1 = \frac{v_0 - r_2 z_0}{r_1 - r_2}, \qquad C_2 = \frac{r_1 z_0 - v_0}{r_1 - r_2}.$$

As in the critical case, solutions are non-oscillatory and converge to zero.

### A.5. Connection to level-set slope coverage intuition

Our identifiability theory in Sec. 4 is phrased in terms of *level-set slope coverage*: for a given state value $u$, consider the set of times at which $z(t) = u$, and the corresponding instantaneous slopes $\{z'(t) : z(t) = u\}$. Coverage requires that on a nontrivial interval of levels $u$, the same level is attained multiple times with sufficiently diverse slopes.

The closed-form solutions above make the regime-dependent behavior transparent.

**Underdamped ($D < 0$).** From (A.1), $z(t)$ oscillates while its envelope decays as $e^{-\zeta t}$. For levels $u$ away from the extrema, the trajectory typically crosses $u$ repeatedly over time. Moreover, because crossings occur at different phases and amplitudes, the associated slopes $z'(t)$ at $z(t) = u$ can vary across crossings. This repeated-crossing behavior is what enables level-set slope coverage on an interval and underpins single-trajectory identifiability in the underdamped regime under sufficient time span (e.g., the two-period sufficient rule in Sec. 4).

**Undamped ($\gamma_1 = 0$).** In the ideal undamped case (A.2), the motion is purely periodic with constant amplitude. For any level $u$ strictly between the extrema, each period yields two crossings with slopes of opposite sign, and in the continuous noiseless setting this induces at most two possible slopes per level (up to periodic repetition). This limited slope diversity is precisely why the level-set slope coverage certificate does not hold in general for a single trajectory in the undamped regime. In practice, discrete sampling, finite-difference estimation, and modeling/optimization biases may still lead to accurate recovery in simulations, but this does not contradict the structural statement that the coverage-based sufficient condition fails in the idealized setting.

**Critical and overdamped ($D \geq 0$).** For $D = 0$ (critical) and $D > 0$ (overdamped), the solutions (A.3)–(A.4) are non-oscillatory and decay toward zero without repeated cycles. As a result, for a typical initialization the trajectory reaches any given level $u$ at most a small number of times (often once, and at most twice), which sharply limits the number of distinct slopes observable at that level within a single clip. This explains why single-trajectory level-set slope coverage is generically unavailable in these regimes, motivating the multi-trajectory diversity condition in Sec. 4: by combining multiple clips with different initial conditions, one can realize the same level $u$ with multiple distinct velocities, restoring the slope diversity required for identification.

# B. Additional finite-sample results

## B.1. Further explanation of $E_{\text{enc}}$

*Remark* B.1 (Affine-plus-remainder decomposition of $f$). On the coverage interval $U$, we decompose

$$f(u) = a\, u + b + h(u), \qquad h \in C^2(U).$$

Here $au + b$ captures the dominant affine alignment between the latent and the ODE state, while $h$ is the deterministic non-affine remainder (not noise). This notation avoids conflict with the residual $r_k(\eta)$. One may take $(a, b)$ as the best affine approximation of $f$ over the current window and define $h(u) := f(u) - (au + b)$. Under the ideal structural regime, our identifiability results imply $h \equiv 0$ on $U$. In finite-sample practice, $h$ may be small but nonzero; its effect is summarized by $E_{\text{enc}}$ (Def. 4.7).

## B.2. Multi-clip bound (shared physics)

Stacking $M$ clips encoded by a shared $E_\phi$ with common parameters $(\gamma_1, \gamma_0)$ increases the effective sample size to $N = \sum_m (T^{(m)} - 1)$ and typically improves conditioning. Geometrically, multi-clip coverage (e.g., three trajectories reaching the same level with distinct instantaneous velocities) recovers the same continuous-time identifiability conclusion (Thm. 4.6) and empirically increases $\psi_{\min}$.

**A noiseless discrete proxy for multiple clips (bridge only).** For clip $m$ with step $\Delta t^{(m)}$ and times $t_k^{(m)} = t_0^{(m)} + k\,\Delta t^{(m)}$, define

$$\tilde{z}_k^{(m)} := f\big(z^{(m)}(t_k^{(m)})\big),$$

and let $\tilde{z}_k'^{(m)}$ and $\tilde{z}_k''^{(m)}$ denote the corresponding *true* time derivatives of $\tilde{z}^{(m)}(t) = f(z^{(m)}(t))$ evaluated at $t = t_k^{(m)}$. Observed latents decompose as

$$\hat{z}_k^{(m)} = \tilde{z}_k^{(m)} + \epsilon_k^{(m)}, \qquad \mathbb{E}[\epsilon_k^{(m)} \mid z_k^{(m)}] = 0,$$

where $\{\epsilon_k^{(m)}\}$ are zero-mean sub-Gaussian (temporal $\beta$-mixing allowed). This bridge again separates statistical effects (noise/discretization) from deterministic encoder mismatch summarized by $E_{\text{enc}}$.

**Definition B.2** (Multi-clip pseudotrue parameter and encoder approximation error). Let $\tilde{X}_k^{(m)} := [\,\tilde{z}_k'^{(m)},\ \tilde{z}_k^{(m)}\,]$ and $\tilde{y}_k^{(m)} := -\tilde{z}_k''^{(m)}$. With $N := \sum_{m=1}^M (T^{(m)} - 1)$, define the (empirical, noiseless, multi-clip) *pseudotrue parameter*

$$\eta^\star \in \arg\min_{\eta \in \mathbb{R}^2} \ \frac{1}{N} \sum_{m=1}^M \sum_{k=1}^{T^{(m)}-1} \big(\tilde{y}_k^{(m)} - \tilde{X}_k^{(m)}\eta\big)^2,$$

and the encoder-induced approximation error $E_{\text{enc}} := \|\eta^\star - \gamma\|_2$. If $f$ is affine on the joint coverage interval, then $\eta^\star = \gamma$ and $E_{\text{enc}} = 0$. In general, $E_{\text{enc}}$ is deterministic model mismatch and is independent of finite-sample inversion; hence it is not scaled by $1/\psi_{\min}$.

**Theorem B.3** (Multi-trajectory finite-sample error (shared physics)). *Let $\{\boldsymbol{x}_k^{(m)}\}_{k=0}^{T^{(m)}}$ be $M$ clips encoded by the same $E_\phi$ to latents $\hat{z}_k^{(m)}$, sharing parameters $(\gamma_1, \gamma_0)$. For each clip form discrete derivatives and residuals by (2) and (3), and set $N := \sum_{m=1}^M (T^{(m)} - 1)$. Let*

$$\hat{\eta} = \arg\min_\eta \ \frac{1}{N} \sum_{m=1}^M \sum_{k=1}^{T^{(m)}-1} \big(r_k^{(m)}(\eta)\big)^2.$$

*Define $X_k^{(m)} = [\,\hat{z}_k'^{(m)},\ \hat{z}_k^{(m)}\,]$ and*

$$\psi_{\min} := \lambda_{\min}\Big(\frac{1}{N} \sum_{m=1}^M \sum_{k=1}^{T^{(m)}-1} (X_k^{(m)})^\top X_k^{(m)}\Big) > 0.$$

*Assume $\hat{z}_k^{(m)} = f(z_k^{(m)}) + \epsilon_k^{(m)}$, where $\{\epsilon_k^{(m)}\}$ are zero-mean sub-Gaussian (temporal $\beta$-mixing allowed) with base noise scale $\sigma_0$, and assume $\Delta t^{(m)} \leq 1$ for all $m$. Let $\Delta t_{\min} := \min_m \Delta t^{(m)}$ and define $\sigma := \sigma_{\text{fd}}(\Delta t_{\min})$ as the corresponding effective sub-Gaussian noise scale after applying the finite-difference stencils in (2) (see Eq. (23)).*

*Let $E_{\mathrm{enc}}$ be as in Def. B.2. Then for any $\delta \in (0, 1)$, with probability at least $1 - \delta$,*

$$\|\hat{\eta} - \gamma\|_2 \leq \frac{C_1 \sigma}{\psi_{\min}} \sqrt{\frac{\log(3/\delta)}{N}} + \frac{C_2 \sigma^2}{\psi_{\min}} + \frac{C_3 \Delta t_{\max}^2}{\psi_{\min}} + E_{\mathrm{enc}}, \tag{11}$$

*where $\Delta t_{\max} = \max_m \Delta t^{(m)}$.*

The proof appears in Appx. C.7. Aggregating clips increases $N$ and can improve $\psi_{\min}$, yielding a transparent design-to-accuracy prescription: collect clips that jointly realize distinct slopes at shared state levels and maintain nondegenerate variance within each window. Note that the effective noise scale $\sigma = \sigma_{\mathrm{fd}}(\Delta t_{\min})$ depends on the sampling step via the finite-difference stencils, so $\Delta t$ controls a bias–variance tradeoff rather than monotonically tightening the bound.

## B.3. Interpreting the bounds

*Reading the bounds.* Both (9) and (11) have the same structure: each is the sum of four contributions. The first three are amplified by $1/\psi_{\min}$, while $E_{\mathrm{enc}}$ is not.

- **Noise term.** Decays as $1/\sqrt{\mathrm{samples}}$ (use $T-1$ for one clip, $N$ for many). The confidence level $1 - \delta$ appears only through $\log(3/\delta)$. Importantly, the noise prefactor is the *effective post-stencil scale* $\sigma = \sigma_{\mathrm{fd}}(\Delta t)$ (or $\sigma_{\mathrm{fd}}(\Delta t_{\min})$ in the multi-clip case), which may increase as $\Delta t$ decreases due to finite-difference amplification.

- **Error-in-variables term.** The $C_2 \sigma^2/\psi_{\min}$ term is a bias floor induced by noisy finite-difference features $X_k$; it vanishes only as the underlying measurement noise decreases (equivalently, as $\sigma_{\mathrm{fd}}(\Delta t) \to 0$).

- **Discretization term.** Scales as $\Delta t^2$ for one clip, or $\Delta t_{\max}^2$ across multiple clips. Smaller time steps reduce this *truncation bias*, but they can simultaneously increase $\sigma_{\mathrm{fd}}(\Delta t)$ and therefore enlarge the stochastic terms. Overall, $\Delta t$ controls a bias–variance tradeoff.

- **Encoder approximation** ($E_{\mathrm{enc}}$)**.** Captures deterministic mismatch from the encoder; it is not divided by $\psi_{\min}$ and improves as the learned representation better aligns with the assumed ODE model.

## C. Proofs

### C.1. Proof of Theorem 4.2

*Proof.* **Step 1: Chain rule expansion and reduction to a polynomial in $z'$.** Since $\hat{z} = f \circ z$ and $f \in C^2$, along the trajectory we have

$$\hat{z}'(t) = f'(z(t))\, z'(t),$$
$$\hat{z}''(t) = f''(z(t))\, (z'(t))^2 + f'(z(t))\, z''(t).$$

Substitute these into (8) and use the true dynamics (1) to eliminate $z''(t)$:

$$
\begin{aligned}
0 &= \hat{z}'' + \eta_1 \hat{z}' + \eta_0 \hat{z} \\
&= f''(z)\,(z')^2 + f'(z)\, z'' + \eta_1 f'(z)\, z' + \eta_0 f(z) \\
&= f''(z)\,(z')^2 + f'(z)\,(-\gamma_1 z' - \gamma_0 z) + \eta_1 f'(z)\, z' + \eta_0 f(z) \\
&= f''(z)\,(z')^2 + (\eta_1 - \gamma_1)\, f'(z)\, z' + \big(\eta_0 f(z) - \gamma_0 z f'(z)\big).
\end{aligned}
$$

Define, for any scalar state value $u$ and slope $v$,

$$Q(u, v) := f''(u)\, v^2 + (\eta_1 - \gamma_1)\, f'(u)\, v + \big(\eta_0 f(u) - \gamma_0 u f'(u)\big). \tag{12}$$

Then the previous identity says that for all $t \in I$,

$$Q\big(z(t),\, z'(t)\big) = 0. \tag{13}$$

**Step 2: Use level-set slope coverage to force a quadratic to vanish identically.** Fix any $u \in U$. By level-set slope coverage, there exist at least three times $t_i(u)$ with

$$z\big(t_i(u)\big) = u \quad \text{and} \quad z'\big(t_i(u)\big) =: v_i(u) \ \text{pairwise distinct.}$$

Evaluating (13) at these times gives
$$Q\big(u, v_i(u)\big) = 0 \quad (i = 1, 2, 3).$$

But for fixed $u$, $Q(u, \cdot)$ is a quadratic polynomial in $v$ (see (12)). A nonzero quadratic over $\mathbb{R}$ can have at most two distinct real roots; therefore the fact that $Q(u, \cdot)$ vanishes at three pairwise distinct velocities forces

$$Q(u, \cdot) \equiv 0 \quad \text{as a polynomial in } v.$$

Hence, for this $u$, all three coefficients must be zero:

$$
\begin{aligned}
f''(u) &= 0, \\
(\eta_1 - \gamma_1)\, f'(u) &= 0, \\
\eta_0 f(u) - \gamma_0 u f'(u) &= 0.
\end{aligned}
\tag{14}
$$

Because $u$ was arbitrary in the open set $U$, the equalities in (14) hold for every $u \in U$.

**Step 3: Conclude that $f$ is affine on $U$.** From $f''(u) = 0$ for all $u \in U$, we obtain that $f$ is affine on $U$:

$$f(u) = a\, u + b \quad \text{for some constants } a, b.$$

Because $f' \not\equiv 0$ on $U$ by assumption, we must have $a \neq 0$.

**Step 4: Identify $\eta_1$ and $\eta_0$.** Plug $f'(u) = a$ into the second equality of (14):

$$(\eta_1 - \gamma_1)\, a = 0 \ \Rightarrow\ \eta_1 = \gamma_1 \quad (\text{since } a \neq 0).$$

Now substitute $f(u) = au + b$ and $f'(u) = a$ into the third equality of (14):

$$\eta_0\,(au + b) - \gamma_0\,u\,a \;\equiv\; 0 \quad \text{for all } u \in U.$$

Equating the coefficients of the polynomial in $u$ yields

$$\text{coefficient of } u: \quad a(\eta_0 - \gamma_0) = 0 \quad \Rightarrow \quad \eta_0 = \gamma_0,$$
$$\text{constant term}: \quad \eta_0\,b = 0.$$

Combining the two, we have $(\eta_1, \eta_0) = (\gamma_1, \gamma_0)$, and additionally $\gamma_0\,b = 0$. Thus if $\gamma_0 \neq 0$ then $b = 0$ (the affine intercept vanishes).

**Step 5: Summary.** We have shown that $f$ is affine on $U$ and that $\eta_1 = \gamma_1$, $\eta_0 = \gamma_0$; furthermore, if $\gamma_0 \neq 0$, then $f(u) = a\,u$ on $U$. This completes the proof. $\qquad\square$

### C.2. Proof of Theorem 4.3

*Proof.* **Step 1 (amplitude–phase form and monotone envelope).** In the strictly underdamped case, the solution can be written as

$$z(t) = R(t)\cos\theta(t),$$
$$R(t) = A\,e^{-\zeta t} \;(A > 0),$$
$$\theta(t) = \omega t + \psi.$$

The envelope $R(t)$ is strictly decreasing and continuous. On $[t_a, t_b]$ set

$$R_{\min} := \min_{t \in [t_a, t_b]} R(t) = R(t_b) > 0.$$

**Step 2 (open interval of admissible levels).** Define

$$U := (-u_\star, u_\star) \quad \text{with} \quad u_\star := R_{\min}.$$

Since $u \in (-R_{\min}, R_{\min})$, we have $|u| < R_{\min}$. Moreover, by definition of $R_{\min} = \min_{t \in [t_a, t_b]} R(t)$ we have $R(t) \geq R_{\min}$ for all $t \in [t_a, t_b]$. Therefore $|u| < R(t)$ for every $t \in [t_a, t_b]$. That is, the level $z = u$ is reachable throughout the window.

**Step 3 (at least three level crossings for each $u \in U$).** Fix $u \in U$. The equation $z(t) = u$ is equivalent to $\cos\theta(t) = u/R(t)$. Let $s := \theta(t)$; since $\theta'(t) = \omega > 0$, $s$ runs over an interval $[s_a, s_b]$ of length $s_b - s_a = \omega L \geq 4\pi$. Because the set of extrema of $\cos s$ is $\{m\pi : m \in \mathbb{Z}\}$ with spacing $\pi$, any interval of length $\geq 4\pi$ contains at least four consecutive points $m\pi$. Thus there exists $m \in \mathbb{Z}$ with

$$[m\pi, (m+3)\pi] \subseteq [s_a, s_b].$$

Define

$$C(s) := u/R(\theta^{-1}(s)) \in (-1, 1) \quad \text{(continuous)}$$

and

$$H(s) := \cos s - C(s).$$

At the four extrema we have

$$H(m\pi) = \pm 1 - C(m\pi),$$
$$H((m+1)\pi) = \mp 1 - C((m+1)\pi),$$
$$H((m+2)\pi) = \pm 1 - C((m+2)\pi),$$
$$H((m+3)\pi) = \mp 1 - C((m+3)\pi),$$

where the signs alternate. Since $C(\cdot) \in (-1, 1)$, these values alternate in sign:

$$H(m\pi) > 0,$$
$$H((m+1)\pi) < 0,$$
$$H((m+2)\pi) > 0,$$
$$H((m+3)\pi) < 0,$$

or the opposite if $m$ changes parity. By the intermediate value theorem, $H$ vanishes at least once in each of the three disjoint intervals $[m\pi, (m+1)\pi]$, $[(m+1)\pi, (m+2)\pi]$, and $[(m+2)\pi, (m+3)\pi]$. Pulling back via $t = \theta^{-1}(s)$ gives three distinct times $t_1 < t_2 < t_3$ in $[t_a, t_b]$ with $z(t_i) = u$.

**Step 4 (first-order derivatives at crossings are pairwise distinct).** Differentiating $z(t) = R \cos \theta$ gives

$$z'(t) = R'(t) \cos \theta(t) - R(t)\theta'(t) \sin \theta(t)$$
$$= -\zeta R(t) \cos \theta(t) - \omega R(t) \sin \theta(t).$$

At a crossing $z(t) = u$,

$$z'(t) = -\zeta u - \omega R(t) \sin \theta(t)$$
$$= -\zeta u \mp \omega h(R(t)),$$

where

$$h(R) := R\sqrt{1 - (u/R)^2} = \sqrt{R^2 - u^2},$$
$$h'(R) = \frac{R}{\sqrt{R^2 - u^2}} > 0 \quad (R > |u|).$$

Let $t_1 < t_2 < t_3$ be three consecutive crossings from Step 3. As time increases across them, $\sin \theta$ alternates sign and $R(t)$ strictly decreases, hence

$$z'(t_1) = -\zeta u - \omega h(R(t_1)),$$
$$z'(t_2) = -\zeta u + \omega h(R(t_2)),$$
$$z'(t_3) = -\zeta u - \omega h(R(t_3)).$$

Since $h$ is strictly increasing in $R$ and $R(t_1) > R(t_2) > R(t_3)$, we have $z'(t_1) \neq z'(t_3)$, and $z'(t_2) > -\zeta u$ while $z'(t_1), z'(t_3) < -\zeta u$. Thus the three derivative values are pairwise distinct.

**Step 5 (apply Theorem 4.2).** Coverage holds on the nonempty open interval $U$, so Theorem 4.2 yields that $f$ is affine on $U$ and $(\eta_1, \eta_0) = (\gamma_1, \gamma_0)$. $\square$

### C.3. Proof of Theorem 4.5

*Proof.* **(i) Critical/overdamped.** When $\gamma_1^2 \geq 4\gamma_0$, the characteristic roots are real: $r_1, r_2 \in \mathbb{R}$.

*1. Case $r_1 \neq r_2$.*

The general solution is

$$z(t) = C_1 e^{r_1 t} + C_2 e^{r_2 t},$$

so

$$z'(t) = r_1 C_1 e^{r_1 t} + r_2 C_2 e^{r_2 t}.$$

If $C_1 = 0$ or $C_2 = 0$, then $z(t)$ is monotonic and each level $u$ is attained at most once. If $C_1 \neq 0$, solving $z'(t) = 0$ gives

$$e^{(r_1 - r_2)t} = -\frac{r_2 C_2}{r_1 C_1},$$

which has *at most one* real solution because the left-hand side is strictly monotone in $t$. Hence $z$ has at most one critical point (one extremum). A $C^1$ function with at most one extremum is monotone on $(-\infty, t^\star]$ and on $[t^\star, \infty)$ (for some $t^\star$ or with one part absent), so every horizontal line $z = u$ intersects its graph in *at most two* points.

2. *Case $r_1 = r_2 =: r$.*

The general solution is

$$z(t) = (C_1 + C_2 t)e^{rt}$$

and

$$z'(t) = (C_2 + r(C_1 + C_2 t))e^{rt}.$$

The factor in parentheses is affine in $t$, so $z'(t) = 0$ has at most one real solution; again $z$ has at most one critical point and intersects any level $u$ in at most two points.

Therefore, in all critical/overdamped cases a single trajectory yields at most two occurrences of the same level $u$, and level-set slope coverage fails (it requires three occurrences with pairwise distinct first-order derivatives at that level).

**(ii) Undamped.** When $\gamma_1 = 0$ and $\gamma_0 > 0$, write $\omega := \sqrt{\gamma_0}$. Solutions are harmonic:

$$z(t) = A\cos(\omega t) + B\sin(\omega t) = \tilde{A}\cos(\omega t + \psi),$$

with amplitude $\tilde{A} = \sqrt{A^2 + B^2} > 0$. Fix any level $u$ with $|u| < \tilde{A}$. The level set $z(t) = u$ is attained infinitely many times:

$$\omega t + \psi = \pm \arccos(u/\tilde{A}) + 2\pi k, \quad k \in \mathbb{Z}.$$

At these times,

$$z'(t) = -\omega \tilde{A}\sin(\omega t + \psi) = \pm \omega \sqrt{\tilde{A}^2 - u^2},$$

so the first-order derivatives at level $u$ take exactly two values (opposite signs), never three pairwise distinct values. Thus level-set slope coverage fails for every single trajectory.

Combining (i) and (ii) completes the proof. □

### C.4. Why the discrete estimator can still recover accurately in the undamped case

**Proposition C.1** (Discrete pseudotrue coefficients for an ideal sampled undamped trajectory)**.** *Consider the ideal undamped solution*

$$z(t) = \tilde{A}\cos(\omega t + \psi), \qquad \omega^2 = \gamma_0, \qquad \gamma_1 = 0,$$

*sampled at times*

$$t_k = t_0 + k\Delta t, \qquad k = 0, 1, \ldots, T.$$

*Let*

$$z_k := z(t_k), \qquad d_{1,k} := \frac{z_{k+1} - z_{k-1}}{2\Delta t}, \qquad d_{2,k} := \frac{z_{k+1} - 2z_k + z_{k-1}}{\Delta t^2}, \qquad k = 1, \ldots, T-1.$$

*For fixed sampled latents $\{z_k\}$, define the discrete ODE loss*

$$L_{\text{ODE}}^{\text{disc}}(\eta_1, \eta_0) := \frac{1}{T-1}\sum_{k=1}^{T-1}\left(d_{2,k} + \eta_1 d_{1,k} + \eta_0 z_k\right)^2.$$

*Then the following hold.*

1. *The sampled trajectory satisfies the exact centered-difference relation*

$$d_{2,k} + \gamma_0^\Delta z_k = 0, \qquad \gamma_0^\Delta := \frac{4\sin^2(\omega\Delta t/2)}{\Delta t^2}, \qquad \gamma_1^\Delta := 0,$$

*for every $k = 1, \ldots, T-1$. Hence*

$$L_{\text{ODE}}^{\text{disc}}(\gamma_1^\Delta, \gamma_0^\Delta) = 0.$$

2. *Let*

$$X := \begin{bmatrix} d_{1,1} & z_1 \\ \vdots & \vdots \\ d_{1,T-1} & z_{T-1} \end{bmatrix}, \qquad y := \begin{bmatrix} -d_{2,1} \\ \vdots \\ -d_{2,T-1} \end{bmatrix}.$$

*If* $\mathrm{rank}(X) = 2$, *then*

$$(\eta_1^\star, \eta_0^\star) = (\gamma_1^\Delta, \gamma_0^\Delta) = (0, \gamma_0^\Delta)$$

*is the unique minimizer of* $L_{\mathrm{ODE}}^{\mathrm{disc}}$.

3. *As* $\Delta t \to 0$,

$$\gamma_1^\Delta = 0, \qquad \gamma_0^\Delta = \frac{4\sin^2(\omega \Delta t/2)}{\Delta t^2} = \omega^2 - \frac{\omega^4}{12}\Delta t^2 + O(\Delta t^4) = \gamma_0 + O(\Delta t^2).$$

*Proof.* Write

$$\theta_k := \omega t_k + \psi, \qquad z_k = \tilde{A}\cos\theta_k.$$

Using the trigonometric identity

$$\cos(\theta + \alpha) - 2\cos\theta + \cos(\theta - \alpha) = -4\sin^2(\alpha/2)\cos\theta,$$

we obtain

$$z_{k+1} - 2z_k + z_{k-1} = \tilde{A}\big(\cos(\theta_k + \omega\Delta t) - 2\cos\theta_k + \cos(\theta_k - \omega\Delta t)\big) = -4\sin^2(\omega\Delta t/2)\, z_k.$$

Dividing by $\Delta t^2$ gives

$$d_{2,k} = -\frac{4\sin^2(\omega\Delta t/2)}{\Delta t^2}\, z_k = -\gamma_0^\Delta z_k.$$

Hence

$$d_{2,k} + \gamma_1^\Delta d_{1,k} + \gamma_0^\Delta z_k = d_{2,k} + 0 \cdot d_{1,k} + \gamma_0^\Delta z_k = 0,$$

for every $k = 1, \ldots, T - 1$, which proves part (1).

For part (2), note that

$$L_{\mathrm{ODE}}^{\mathrm{disc}}(\eta_1, \eta_0) = \frac{1}{T-1}\|y - X\eta\|_2^2, \qquad \eta := \begin{bmatrix} \eta_1 \\ \eta_0 \end{bmatrix}.$$

By part (1),

$$y = X\begin{bmatrix} 0 \\ \gamma_0^\Delta \end{bmatrix},$$

so $\eta^\star = (0, \gamma_0^\Delta)$ achieves zero loss. If another parameter pair $\eta$ also minimizes the loss, then necessarily

$$0 = \|y - X\eta\|_2^2 = \left\| X\begin{bmatrix} 0 \\ \gamma_0^\Delta \end{bmatrix} - X\eta \right\|_2^2 = \|X(\eta^\star - \eta)\|_2^2.$$

Thus $X(\eta^\star - \eta) = 0$. If $\mathrm{rank}(X) = 2$, then $\ker(X) = \{0\}$, so $\eta = \eta^\star$. Therefore $(0, \gamma_0^\Delta)$ is the unique minimizer.

For part (3), let $x = \omega\Delta t/2$. Using

$$\sin x = x - \frac{x^3}{6} + O(x^5),$$

we have

$$\sin^2 x = x^2 - \frac{x^4}{3} + O(x^6).$$

Substituting $x = \omega\Delta t/2$ gives

$$\gamma_0^\Delta = \frac{4\sin^2(\omega\Delta t/2)}{\Delta t^2} = \frac{4}{\Delta t^2}\left[\left(\frac{\omega\Delta t}{2}\right)^2 - \frac{1}{3}\left(\frac{\omega\Delta t}{2}\right)^4 + O(\Delta t^6)\right] = \omega^2 - \frac{\omega^4}{12}\Delta t^2 + O(\Delta t^4).$$

Since $\omega^2 = \gamma_0$, the claimed expansion follows. $\qquad\square$

**Interpretation.** Prop. C.1 is a statement about the discrete regression problem actually fit by the centered-difference pipeline when the sampled latent trajectory is an ideal undamped sinusoid. It does *not* restore continuous-time single-trajectory identifiability. Thm. 4.5 remains valid: in continuous time, fixing any level $u$ on an undamped trajectory yields only two possible derivatives,

$$z'(t) = \pm\omega\sqrt{\tilde{A}^2 - u^2},$$

so exact level-set slope coverage fails for a single trajectory. What the proposition shows instead is that, after sampling, the centered-difference least-squares fit has a natural discrete pseudotrue target $(\gamma_1^\Delta, \gamma_0^\Delta) = (0, \gamma_0^\Delta)$, and this target is unique whenever the discrete design matrix $X = [d_1 \ z]$ has full column rank. This also clarifies why long undamped clips can become stable in practice. For short windows, the two columns of $X$ can be nearly collinear, so the regression is poorly conditioned even if the minimizer is formally unique. As the sampled window spans more phase, the conditioning improves, and the estimator stabilizes around $(\gamma_1^\Delta, \gamma_0^\Delta)$. Since

$$\gamma_1^\Delta = 0, \qquad \gamma_0^\Delta = \gamma_0 + O(\Delta t^2),$$

accurate undamped estimates on long sampled clips are compatible with, and do not contradict, the continuous-time structural non-identifiability result.

### C.5. Proof of Theorem 4.6

*Proof.* **Step 1 (chain rule identity).** For any trajectory $z(t)$ with image in $U$, set $\hat{z}(t) = f(z(t))$. By the chain rule,

$$\hat{z}'(t) = f'(z(t))\, z'(t),$$
$$\hat{z}''(t) = f''(z(t))\, (z'(t))^2 + f'(z(t))\, z''(t).$$

Using the true dynamics $z'' = -\gamma_1 z' - \gamma_0 z$ and the latent ODE $\hat{z}'' + \eta_1 \hat{z}' + \eta_0 \hat{z} = 0$, we obtain along the trajectory

$$f''(z)\,(z')^2 + (\eta_1 - \gamma_1)\, f'(z)\, z' + \big(\eta_0 f(z) - \gamma_0 z\, f'(z)\big) = 0.$$

For notational clarity, define for $u \in U$ and $v \in \mathbb{R}$ the quadratic polynomial

$$Q(u, v) := f''(u)\, v^2 + (\eta_1 - \gamma_1)\, f'(u)\, v \tag{15}$$
$$+ \big(\eta_0 f(u) - \gamma_0 u\, f'(u)\big).$$

Then for any solution $z$ with image in $U$,

$$Q\big(z(t),\, z'(t)\big) = 0 \quad \text{whenever } z(t) \in U. \tag{16}$$

**Step 2 (use three trajectories to annihilate the quadratic).** Fix an arbitrary $u \in U$. By hypothesis there exist times $t_m(u)$ on the three trajectories with $z^{(m)}(t_m(u)) = u$ and pairwise distinct *first-order derivatives (velocities)* $v_m(u) = z^{(m)'}(t_m(u))$. Applying (16) to each $m$ yields

$$Q\big(u, v_m(u)\big) = 0, \qquad m = 1, 2, 3.$$

For this fixed $u$, $Q(u, \cdot)$ is a real quadratic in $v$. A nonzero real quadratic has at most two distinct real roots; since it vanishes at three pairwise distinct $v_m(u)$, it must be the zero polynomial:

$$Q(u, \cdot) \equiv 0.$$

Hence all three coefficients in (15) vanish at $u$:

$$f''(u) = 0,$$
$$(\eta_1 - \gamma_1)\, f'(u) = 0, \tag{17}$$
$$\eta_0 f(u) - \gamma_0 u\, f'(u) = 0.$$

Because $u \in U$ was arbitrary, (17) holds for every $u \in U$.

**Step 3 (affinity of $f$ on $U$).** From $f''(u) = 0$ for all $u \in U$ we conclude that $f$ is affine on $U$:

$$f(u) = a\,u + b \quad \text{on each connected component of } U.$$

By the nondegeneracy assumption $f' \not\equiv 0$ on $U$, we have $a \neq 0$.

**Step 4 (identify the parameters).** Using $f'(u) = a$ in $(\eta_1 - \gamma_1)f'(u) = 0$ yields

$$\eta_1 = \gamma_1.$$

Substituting $f(u) = au + b$ and $f'(u) = a$ into $\eta_0 f(u) - \gamma_0 u f'(u) = 0$ gives, for all $u \in U$,

$$\eta_0\,(au + b) - \gamma_0\,u\,a \equiv 0.$$

Comparing coefficients in this affine polynomial in $u$,

$$a(\eta_0 - \gamma_0) = 0 \;\Rightarrow\; \eta_0 = \gamma_0, \qquad \eta_0\,b = 0.$$

Thus $(\eta_1, \eta_0) = (\gamma_1, \gamma_0)$, and if $\gamma_0 \neq 0$ then $b = 0$ so $f(u) = a\,u$ on $U$.

**Step 5 (conclusion).** We have shown $f$ is affine on $U$ and $(\eta_1, \eta_0) = (\gamma_1, \gamma_0)$ as claimed; the intercept vanishes when $\gamma_0 \neq 0$. $\qquad\square$

### C.6. Proof of Theorem 4.8

*Proof.* **Step 1 (bias–variance split via Def. 4.7).** By the triangle inequality and Def. 4.7,

$$\|\hat{\eta} - \gamma\|_2 \;\leq\; \|\hat{\eta} - \eta^\star\|_2 + E_{\mathrm{enc}}.$$

We now upper bound the *statistical part* $\|\hat{\eta} - \eta^\star\|_2$; the deterministic approximation bias remains as $E_{\mathrm{enc}}$ in (9).

**Step 2 (normal equations around $\eta^\star$).** Let $X_k := [\,\hat{z}'_k,\ \hat{z}_k\,] \in \mathbb{R}^{1 \times 2}$ and stack $X \in \mathbb{R}^{(T-1) \times 2}$; let $y_k := -\hat{z}''_k$ and $y \in \mathbb{R}^{T-1}$. The minimizer $\hat{\eta}$ of $\frac{1}{T-1}\|y - X\eta\|_2^2$ satisfies

$$\frac{1}{T-1} X^\top (y - X\hat{\eta}) = 0,$$

hence

$$\frac{1}{T-1} X^\top (y - X\eta^\star) = \frac{1}{T-1} X^\top X\,(\hat{\eta} - \eta^\star).$$

Writing $G := \frac{1}{T-1} X^\top X$ and $\psi_{\min} := \lambda_{\min}(G)$ yields the master inequality

$$\|\hat{\eta} - \eta^\star\|_2 \;\leq\; \frac{1}{\psi_{\min}} \left\| \frac{1}{T-1} X^\top r(\eta^\star) \right\|_2, \tag{18}$$

$$r_k(\eta) = \hat{z}''_k + \eta_1 \hat{z}'_k + \eta_0 \hat{z}_k.$$

**Step 3 (residual decomposition at $\eta^\star$).** Use the proxy $\hat{z}_k = \tilde{z}_k + \epsilon_k$ and the centered stencil (2):

$$\hat{z}'_k = \tilde{z}'_k + \xi_k^{(1)} + \delta_k^{(1)}, \qquad \hat{z}''_k = \tilde{z}''_k + \xi_k^{(2)} + \delta_k^{(2)},$$

$$\delta_k^{(1)} := \frac{\tilde{z}_{k+1} - \tilde{z}_{k-1}}{2\Delta t} - \tilde{z}'_k, \qquad \delta_k^{(2)} := \frac{\tilde{z}_{k+1} - 2\tilde{z}_k + \tilde{z}_{k-1}}{\Delta t^2} - \tilde{z}''_k,$$

where $(\xi_k^{(1)}, \xi_k^{(2)})$ are linear transforms of $(\epsilon_{k-1}, \epsilon_k, \epsilon_{k+1})$ (hence sub-Gaussian, with an explicit $\Delta t$-dependent effective scale). Concretely, define the stencil-induced noise terms

$$\xi_k^{(1)} := \frac{\epsilon_{k+1} - \epsilon_{k-1}}{2\Delta t}, \qquad \xi_k^{(2)} := \frac{\epsilon_{k+1} - 2\epsilon_k + \epsilon_{k-1}}{\Delta t^2}.$$

Let $\sigma_0$ be the (base) sub-Gaussian noise scale of $\epsilon_k$ as assumed in Thm. 4.8. By standard stability of sub-Gaussianity under fixed linear combinations, there exist absolute constants $c_0, c_1, c_2 > 0$ (depending only on the fixed stencil and, if applicable, the $\beta$-mixing constants) such that $\epsilon_k$, $\xi_k^{(1)}$, and $\xi_k^{(2)}$ are all sub-Gaussian with scales bounded by

$$c_0 \sigma_0, \qquad c_1 \frac{\sigma_0}{\Delta t}, \qquad c_2 \frac{\sigma_0}{\Delta t^2},$$

respectively. We therefore define the effective (post-stencil) noise level

$$\sigma := \sigma_{\text{fd}}(\Delta t) := \max\left\{ c_0 \sigma_0, \ c_1 \frac{\sigma_0}{\Delta t}, \ c_2 \frac{\sigma_0}{\Delta t^2} \right\}, \tag{19}$$

so that $\epsilon_k$, $\xi_k^{(1)}$, and $\xi_k^{(2)}$ may all be treated as sub-Gaussian with a common scale parameter $\sigma$.

Intuitively, finite differences amplify measurement noise by factors $1/\Delta t$ and $1/\Delta t^2$ for first and second derivatives.

Moreover, $|\delta_k^{(i)}| \le C_\Delta \Delta t^2$ (second-order truncation). Thus

$$r_k(\eta^\star) = \underbrace{\tilde{r}_k(\eta^\star)}_{\text{noiseless residual}} + \underbrace{\xi_k^{(2)} + \eta_1^\star \xi_k^{(1)} + \eta_0^\star \epsilon_k}_{=:u_k \ (\text{stochastic})} + \underbrace{\delta_k^{(2)} + \eta_1^\star \delta_k^{(1)}}_{=:v_k \ (\text{truncation})},$$

where $\tilde{r}_k(\eta) := \tilde{z}_k'' + \eta_1 \tilde{z}_k' + \eta_0 \tilde{z}_k$ and, by normal equations for the *noiseless* problem, $\frac{1}{T-1}\tilde{X}^\top \tilde{r}(\eta^\star) = 0$ with $\tilde{X}_k = [\tilde{z}_k', \tilde{z}_k]$.

**Step 4 (stochastic term).** Let $\Delta X := X - \tilde{X}$. Premultiplying (C.6) by $\frac{1}{T-1}X^\top(\cdot)$ and using $\frac{1}{T-1}\tilde{X}^\top \tilde{r}(\eta^\star) = 0$ (normal equations for the noiseless proxy) yields

$$\frac{1}{T-1}X^\top \tilde{r}(\eta^\star) = \frac{1}{T-1}\Delta X^\top \tilde{r}(\eta^\star).$$

Therefore,

$$\frac{1}{T-1}X^\top u + \frac{1}{T-1}(X - \tilde{X})^\top \tilde{r}(\eta^\star) = \underbrace{\frac{1}{T-1}\tilde{X}^\top u}_{(A)} + \underbrace{\frac{1}{T-1}\Delta X^\top \tilde{r}(\eta^\star)}_{(B)} + \underbrace{\frac{1}{T-1}\Delta X^\top u}_{(C)}.$$

*Terms (A) and (B).* Conditional on $\{z_k\}$, the sequence $\{u_k\}$ is mean-zero sub-Gaussian (with $\beta$-mixing allowed), while $\{\Delta X_k\}$ is also sub-Gaussian since it is a fixed linear transform of $(\epsilon_{k-1}, \epsilon_k, \epsilon_{k+1})$. By (19), the primitive noises $\epsilon_k$ and the stencil noises $\xi_k^{(1)}, \xi_k^{(2)}$ are all sub-Gaussian with common scale $\sigma = \sigma_{\text{fd}}(\Delta t)$. Since $u_k = \xi_k^{(2)} + \eta_1^\star \xi_k^{(1)} + \eta_0^\star \epsilon_k$ is a fixed linear combination, $\{u_k\}$ is sub-Gaussian with scale at most $C_u \sigma$, where $C_u$ depends only on bounds on $\eta^\star$ over the window. Likewise, $\Delta X_k = [\xi_k^{(1)} + \delta_k^{(1)}, \ \epsilon_k]$ is sub-Gaussian with scale $O(\sigma)$ (the deterministic shift $\delta_k^{(1)}$ does not affect tails). Absorb these constants into $C_{1a}$. Moreover, $\tilde{X}$ and $\tilde{r}(\eta^\star)$ depend only on $\{z_k\}$ and the stencil. Hence, Bernstein/Freedman-type concentration for $\beta$-mixing sub-Gaussian sequences implies that, for any $\delta \in (0,1)$, with probability at least $1 - \delta/3$,

$$\|(A) + (B)\|_2 \le C_{1a} \sigma \sqrt{\frac{\log(3/\delta)}{T-1}}.$$

*Term (C): error-in-variables interaction.* Each summand in (C) is a product of sub-Gaussian random variables and is therefore sub-exponential. We decompose it into a (possibly nonzero) conditional mean plus a centered fluctuation:

$$(C) = \underbrace{\mathbb{E}[(C) \mid \{z_k\}]}_{=:b_{\text{eiv}}} + ((C) - b_{\text{eiv}}).$$

By Cauchy–Schwarz and sub-Gaussian moment bounds, $\|b_{\mathrm{eiv}}\|_2 \leq C_{1b}\sigma^2$. Moreover, Bernstein/Freedman-type concentration for $\beta$-mixing sub-exponential sequences yields that, with probability at least $1 - \delta/3$,

$$\|(\mathrm{C}) - b_{\mathrm{eiv}}\|_2 \;\leq\; C_{1c}\,\sigma^2\,\sqrt{\frac{\log(3/\delta)}{T-1}}.$$

Combining these bounds and applying a union bound over the two events above gives, with probability at least $1 - 2\delta/3$,

$$\left\|\frac{1}{T-1}X^\top u \;+\; \frac{1}{T-1}(X-\tilde{X})^\top \tilde{r}(\eta^\star)\right\|_2 \;\leq\; C_1\,\sigma\,\sqrt{\frac{\log(3/\delta)}{T-1}} \;+\; C_2\,\sigma^2, \tag{20}$$

after adjusting constants. Here we absorbed the lower-order fluctuation term $C_{1c}\sigma^2\sqrt{\log(3/\delta)/(T-1)}$ into $C_2\sigma^2$ by adjusting constants (e.g., since $\sqrt{\log(3/\delta)/(T-1)} \leq 1$ when $T - 1 \geq \log(3/\delta)$).

**Step 5 (truncation term control).** Write $X = \tilde{X} + \Delta X$. Then

$$\left\|\frac{1}{T-1}X^\top v\right\|_2 \leq \left\|\frac{1}{T-1}\tilde{X}^\top v\right\|_2 + \left\|\frac{1}{T-1}\Delta X^\top v\right\|_2.$$

Since $\|v\|_\infty \leq C'_\Delta \Delta t^2$ and $\tilde{X}_k = [\,\tilde{z}'_k, \tilde{z}_k\,]$ has bounded moments on the window,

$$\left\|\frac{1}{T-1}\tilde{X}^\top v\right\|_2 \leq C_3\,\Delta t^2.$$

Moreover, $\Delta X_k$ is sub-Gaussian with scale $O(\sigma)$ by (19), and each summand of $\frac{1}{T-1}\Delta X^\top v$ is multiplied by $v_k = O(\Delta t^2)$. Hence, Bernstein/Freedman-type concentration for $\beta$-mixing sub-Gaussian sequences implies that, with probability at least $1 - \delta/3$,

$$\left\|\frac{1}{T-1}\Delta X^\top v\right\|_2 \leq C_{3a}\,\sigma\,\Delta t^2\sqrt{\frac{\log(3/\delta)}{T-1}}.$$

$$\left\|\frac{1}{T-1}X^\top v\right\|_2 \;\leq\; C_3\,\Delta t^2 \;+\; C_{3a}\,\sigma\,\Delta t^2\sqrt{\frac{\log(3/\delta)}{T-1}}. \tag{21}$$

Using $\Delta t \leq 1$, this term is of the same order as the leading stochastic term in Step 4 and can be absorbed into $C_1\sigma\sqrt{\log(3/\delta)/(T-1)}$ after adjusting constants.

**Step 6 (combine).** Plugging (20) (which holds with probability at least $1 - 2\delta/3$) and (21) (which holds with probability at least $1 - \delta/3$) into (18) and taking a union bound yields, with probability at least $1 - \delta$, and after adjusting constants,

$$\|\hat{\eta} - \eta^\star\|_2 \;\leq\; \frac{C_1\sigma}{\psi_{\min}}\sqrt{\frac{\log(3/\delta)}{T-1}} \;+\; \frac{C_2\sigma^2}{\psi_{\min}} \;+\; \frac{C_3\Delta t^2}{\psi_{\min}}.$$

Finally, adding the approximation bias $E_{\mathrm{enc}} = \|\eta^\star - \gamma\|_2$ from Step 1 yields (9). $\qquad\square$

### C.7. Proof of Theorem B.3

*Proof.* **Step 1 (bias–variance split across clips).** By Def. B.2 and the triangle inequality,

$$\|\hat{\eta} - \gamma\|_2 \;\leq\; \|\hat{\eta} - \eta^\star\|_2 \;+\; E_{\mathrm{enc}}.$$

We now bound the *statistical part* $\|\hat{\eta} - \eta^\star\|_2$; the deterministic approximation bias contributes the final $E_{\mathrm{enc}}$ term in (11).

**Step 2 (stacked normal equations around $\eta^\star$).** For clip $m$, set $X_k^{(m)} = [\,\hat{z}_k'^{(m)}, \hat{z}_k^{(m)}\,] \in \mathbb{R}^{1\times 2}$ and $y_k^{(m)} := -\hat{z}_k''^{(m)}$. Stack all clips:

$$X \in \mathbb{R}^{N\times 2}, \quad y \in \mathbb{R}^N, \quad N = \sum_{m=1}^{M}\left(T^{(m)} - 1\right).$$

The minimizer $\hat{\eta}$ of $\frac{1}{N}\|y - X\eta\|_2^2$ satisfies

$$\frac{1}{N}X^\top(y - X\hat{\eta}) = 0,$$

hence

$$\frac{1}{N}X^\top(y - X\eta^\star) = \frac{1}{N}X^\top X(\hat{\eta} - \eta^\star).$$

Let $G := \frac{1}{N}X^\top X$ and $\psi_{\min} := \lambda_{\min}(G)$. Then

$$\|\hat{\eta} - \eta^\star\|_2 \leq \frac{1}{\psi_{\min}}\left\|\frac{1}{N}X^\top r(\eta^\star)\right\|_2, \tag{22}$$
$$r_k^{(m)}(\eta) := \hat{z}_k''^{(m)} + \eta_1\hat{z}_k'^{(m)} + \eta_0\hat{z}_k^{(m)}.$$

**Step 3 (per-clip residual decomposition at $\eta^\star$).** Using the proxy $\hat{z}_k^{(m)} = \tilde{z}_k^{(m)} + \epsilon_k^{(m)}$ and the centered stencil (2),

$$\hat{z}_k'^{(m)} = \tilde{z}_k'^{(m)} + \xi_k^{(1,m)} + \delta_k^{(1,m)},$$
$$\hat{z}_k''^{(m)} = \tilde{z}_k''^{(m)} + \xi_k^{(2,m)} + \delta_k^{(2,m)},$$

where $(\xi_k^{(1,m)}, \xi_k^{(2,m)})$ are linear transforms of $(\epsilon_{k-1}^{(m)}, \epsilon_k^{(m)}, \epsilon_{k+1}^{(m)})$ (hence sub-Gaussian with an explicit $\Delta t^{(m)}$-dependent effective scale). Define

$$\xi_k^{(1,m)} := \frac{\epsilon_{k+1}^{(m)} - \epsilon_{k-1}^{(m)}}{2\Delta t^{(m)}}, \qquad \xi_k^{(2,m)} := \frac{\epsilon_{k+1}^{(m)} - 2\epsilon_k^{(m)} + \epsilon_{k-1}^{(m)}}{(\Delta t^{(m)})^2}.$$

Let $\sigma_0$ be the base sub-Gaussian noise scale of $\epsilon_k^{(m)}$. As in the single-clip case, there exist absolute constants $c_0, c_1, c_2 > 0$ such that $\epsilon_k^{(m)}, \xi_k^{(1,m)}, \xi_k^{(2,m)}$ are sub-Gaussian with scales bounded by

$$c_0\sigma_0, \qquad c_1\frac{\sigma_0}{\Delta t^{(m)}}, \qquad c_2\frac{\sigma_0}{(\Delta t^{(m)})^2}.$$

Define

$$\sigma_{\mathrm{fd}}(\Delta t^{(m)}) := \max\left\{c_0\sigma_0,\ c_1\frac{\sigma_0}{\Delta t^{(m)}},\ c_2\frac{\sigma_0}{(\Delta t^{(m)})^2}\right\}$$

as the corresponding effective scale and set

$$\sigma := \max_m \sigma_{\mathrm{fd}}(\Delta t^{(m)}) = \sigma_{\mathrm{fd}}(\Delta t_{\min}). \tag{23}$$

Moreover, $|\delta_k^{(i,m)}| \leq C_\Delta(\Delta t^{(m)})^2$. Thus

$$r_k^{(m)}(\eta^\star) = \underbrace{\tilde{r}_k^{(m)}(\eta^\star)}_{\text{noiseless residual}} + \underbrace{\xi_k^{(2,m)} + \eta_1^\star\xi_k^{(1,m)} + \eta_0^\star\epsilon_k^{(m)}}_{=:u_k^{(m)}\ \text{(stochastic)}} + \underbrace{\delta_k^{(2,m)} + \eta_1^\star\delta_k^{(1,m)}}_{=:v_k^{(m)}\ \text{(truncation)}}, \tag{24}$$

where $\tilde{r}_k^{(m)}(\eta) := \tilde{z}_k''^{(m)} + \eta_1\tilde{z}_k'^{(m)} + \eta_0\tilde{z}_k^{(m)}$ and, by the noiseless normal equations, $\frac{1}{N}\sum_{m,k}(\tilde{X}_k^{(m)})^\top\tilde{r}_k^{(m)}(\eta^\star) = 0$ with $\tilde{X}_k^{(m)} = [\tilde{z}_k'^{(m)}, \tilde{z}_k^{(m)}]$.

**Step 4 (stochastic term; independent clips).** Let $\Delta X := X - \tilde{X}$. Premultiplying (24) by $\frac{1}{N}X^\top(\cdot)$ and using the noiseless normal equations $\frac{1}{N}\tilde{X}^\top\tilde{r}(\eta^\star) = 0$ yields

$$\frac{1}{N}X^\top\tilde{r}(\eta^\star) = \frac{1}{N}\Delta X^\top\tilde{r}(\eta^\star).$$

Therefore,

$$\frac{1}{N}X^\top u \ + \ \frac{1}{N}(X - \tilde{X})^\top \tilde{r}(\eta^\star) = \underbrace{\frac{1}{N}\tilde{X}^\top u}_{(A)} + \underbrace{\frac{1}{N}\Delta X^\top \tilde{r}(\eta^\star)}_{(B)} + \underbrace{\frac{1}{N}\Delta X^\top u}_{(C)}.$$

*Terms (A) and (B).* Conditional on $\{z_k^{(m)}\}$, the sequence $\{u_k^{(m)}\}$ is mean-zero sub-Gaussian (with within-clip $\beta$-mixing allowed). Moreover, by construction, $\Delta X_k^{(m)}$ is sub-Gaussian as a fixed linear transform of $(\epsilon_{k-1}^{(m)}, \epsilon_k^{(m)}, \epsilon_{k+1}^{(m)})$. Under the assumption that clips are independent, the collections $\{u^{(m)}\}_m$ and $\{\Delta X^{(m)}\}_m$ are independent across $m$. Since $\tilde{X}$ and $\tilde{r}(\eta^\star)$ depend only on the noiseless trajectories and the stencil, Bernstein/Freedman-type concentration (applied within each clip and aggregated over independent clips) implies that, for any $\delta \in (0, 1)$, with probability at least $1 - \delta/3$,

$$\|(A) + (B)\|_2 \ \le \ C_{1a}\, \sigma\, \sqrt{\frac{\log(3/\delta)}{N}}.$$

*Term (C): error-in-variables interaction.* Each summand in (C) is a product of sub-Gaussian random variables and is therefore sub-exponential. Write

$$(C) = \underbrace{\mathbb{E}\Big[(C)\, \Big|\, \{z_k^{(m)}\}\Big]}_{=:b_{\mathrm{eiv}}} + \big((C) - b_{\mathrm{eiv}}\big).$$

By Cauchy–Schwarz and sub-Gaussian moment bounds,

$$\|b_{\mathrm{eiv}}\|_2 \le C_{1b}\, \sigma^2.$$

Moreover, Bernstein/Freedman-type concentration for within-clip $\beta$-mixing and across-clip independence gives that, with probability at least $1 - \delta/3$,

$$\|(C) - b_{\mathrm{eiv}}\|_2 \ \le \ C_{1c}\, \sigma^2\, \sqrt{\frac{\log(3/\delta)}{N}}.$$

Combining these bounds and applying a union bound over the two events above gives, with probability at least $1 - 2\delta/3$,

$$\left\|\frac{1}{N}X^\top u \ + \ \frac{1}{N}(X - \tilde{X})^\top \tilde{r}(\eta^\star)\right\|_2 \ \le \ C_1\, \sigma\, \sqrt{\frac{\log(3/\delta)}{N}} \ + \ C_2\, \sigma^2, \tag{25}$$

after adjusting constants (absorbing the lower-order deviation term into $C_2\sigma^2$ as before).

**Step 5 (truncation term control).** Write $X = \tilde{X} + \Delta X$ and stack $v$ across clips. Then

$$\left\|\frac{1}{N}X^\top v\right\|_2 \le \left\|\frac{1}{N}\tilde{X}^\top v\right\|_2 + \left\|\frac{1}{N}\Delta X^\top v\right\|_2.$$

Since $\|v_k^{(m)}\|_\infty \le C_\Delta'(\Delta t^{(m)})^2 \le C_\Delta'\, \Delta t_{\max}^2$ and $\tilde{X}_k^{(m)} = [\, \tilde{z}_k'^{(m)}, \tilde{z}_k^{(m)}\,]$ has bounded moments on each window,

$$\left\|\frac{1}{N}\tilde{X}^\top v\right\|_2 \le C_3\, \Delta t_{\max}^2.$$

Moreover, $\Delta X_k^{(m)}$ is sub-Gaussian with uniform scale $O(\sigma)$ (by the definition of $\sigma = \sigma_{\mathrm{fd}}(\Delta t_{\min})$), and each summand in $\frac{1}{N}\Delta X^\top v$ is multiplied by $v_k^{(m)} = O(\Delta t_{\max}^2)$. Under within-clip $\beta$-mixing and independence across clips, a Bernstein/Freedman-type concentration bound implies that, for any $\delta \in (0, 1)$, with probability at least $1 - \delta/3$,

$$\left\|\frac{1}{N}\Delta X^\top v\right\|_2 \le C_{3a}\, \sigma\, \Delta t_{\max}^2 \sqrt{\frac{\log(3/\delta)}{N}}.$$

Consequently, on this event,

$$\left\|\frac{1}{N}X^\top v\right\|_2 \le C_3\, \Delta t_{\max}^2 + C_{3a}\, \sigma\, \Delta t_{\max}^2 \sqrt{\frac{\log(3/\delta)}{N}}. \tag{26}$$

Assuming $\Delta t^{(m)} \leq 1$ for all $m$ (hence $\Delta t_{\max} \leq 1$), the second term above is of the same order as the leading stochastic term in Step 4 and can be absorbed into $C_1 \sigma \sqrt{\log(3/\delta)/N}$ after adjusting constants.

**Step 6 (combine).** Insert (25) (which holds with probability at least $1 - 2\delta/3$ from Step 4) and (26) (which holds with probability at least $1 - \delta/3$ from Step 5) into (22). By a union bound, the following holds with probability at least $1 - \delta$:

$$\|\hat{\eta} - \eta^\star\|_2 \ \leq\ \frac{C_1\,\sigma}{\psi_{\min}} \sqrt{\frac{\log(3/\delta)}{N}} + \frac{C_2\,\sigma^2}{\psi_{\min}} + \frac{C_3\,\Delta t_{\max}^2}{\psi_{\min}},$$

after adjusting constants (absorbing the $\Delta t_{\max}^2$-weighted stochastic remainder from (26) into the leading term when $\Delta t_{\max} \leq 1$).

Finally, adding the approximation bias $E_{\mathrm{enc}} = \|\eta^\star - \gamma\|_2$ from Step 1 yields (11).

$\square$

## D. Design conditioning

**Lemma D.1** (Design conditioning via coverage and boundary control). *Let $X_k = [\,\hat{z}'_k, \; \hat{z}_k\,]$ for $k = 1, \ldots, T-1$ and*

$$G \;=\; \frac{1}{T-1} \sum_{k=1}^{T-1} X_k^\top X_k \;=\; \begin{bmatrix} \overline{(\hat{z}')^2} & \overline{\hat{z}'\hat{z}} \\ \overline{\hat{z}'\hat{z}} & \overline{\hat{z}^2} \end{bmatrix},$$

$$\overline{(\cdot)} := \frac{1}{T-1} \sum_{k=1}^{T-1} (\cdot).$$

*Assume:*

1. *level-set slope coverage holds on a nonempty open interval $U$, and let $\mathcal{K} := \{k : \hat{z}_k \in U\}$, $\rho := |\mathcal{K}|/(T-1) \in (0, 1]$;*

2. *there exists a mean squared slope floor $\upsilon_\star > 0$ such that $\frac{1}{|\mathcal{K}|} \sum_{k \in \mathcal{K}} (\hat{z}'_k)^2 \geq \upsilon_\star$;*

3. *the variance-floor regularizer (5) enforces $\sqrt{\widehat{\mathrm{Var}}(\hat{z})} \geq \tau$ at optimum; and*

4. *centered differences (2) are used with fixed $\Delta t$.*

*Let $L := (T-1)\Delta t$ be the window length and define the boundary constant*

$$C_{\mathrm{bd}} \;:=\; \tfrac{1}{2}\big(|\hat{z}_{T-1}\hat{z}_T - \hat{z}_1\hat{z}_0|\big).$$

*Then*

$$\lambda_{\min}(G) \;\geq\; \min\{\,\rho\,\upsilon_\star, \; \overline{\hat{z}^2}\,\} \;-\; \frac{C_{\mathrm{bd}}}{L}. \tag{27}$$

*Define*

$$b_1 \;:=\; \frac{T+1}{T-1} \;-\; \frac{\hat{z}_0^2 + \hat{z}_T^2}{(T-1)\tau^2}, \qquad c_1 \;:=\; \max\{0, \, b_1\}.$$

*Then the variance floor implies the explicit bound*

$$\overline{\hat{z}^2} \;\geq\; \frac{T+1}{T-1}\,\widehat{\mathrm{Var}}(\hat{z}) \;-\; \frac{\hat{z}_0^2 + \hat{z}_T^2}{T-1} \;\geq\; b_1\,\tau^2. \tag{28}$$

*Since $\overline{\hat{z}^2} \geq 0$ always, we may strengthen this to*

$$\overline{\hat{z}^2} \;\geq\; \max\{0, \, b_1\tau^2\} \;=\; c_1\,\tau^2. \tag{29}$$

*So that, combining (27) and (29),*

$$\lambda_{\min}(G) \;\geq\; \min\{\,\rho\,\upsilon_\star, \; c_1\tau^2\,\} \;-\; \frac{C_{\mathrm{bd}}}{L}. \tag{30}$$

*Proof.* Write the entries of $G$ as $G_{11} = \overline{(\hat{z}')^2}$, $G_{22} = \overline{\hat{z}^2}$, and $G_{12} = G_{21} = \overline{\hat{z}'\hat{z}}$. By Gershgorin (or the $2 \times 2$ eigenvalue formula),

$$\lambda_{\min}(G) \;\geq\; \min\{\,G_{11}, G_{22}\,\} \;-\; |G_{12}|. \tag{31}$$

**Step 1 (velocity energy via coverage).** Since $(\hat{z}'_k)^2 \geq 0$ for all $k$ and $\mathcal{K} \subset \{1, \ldots, T-1\}$,

$$G_{11} = \frac{1}{T-1} \sum_{k=1}^{T-1} (\hat{z}'_k)^2 \;\geq\; \frac{|\mathcal{K}|}{T-1} \cdot \frac{1}{|\mathcal{K}|} \sum_{k \in \mathcal{K}} (\hat{z}'_k)^2 \;\geq\; \rho\,\upsilon_\star.$$

**Step 2 (cross term is a boundary term under centered differences).** Using the centered stencil (2),

$$\hat{z}'_k = \frac{\hat{z}_{k+1} - \hat{z}_{k-1}}{2\Delta t},$$

hence the cross term telescopes:

$$\sum_{k=1}^{T-1} \hat{z}_k \hat{z}'_k = \frac{1}{2\Delta t} \sum_{k=1}^{T-1} \left( \hat{z}_k \hat{z}_{k+1} - \hat{z}_k \hat{z}_{k-1} \right)$$

$$= \frac{1}{2\Delta t} \left( \hat{z}_{T-1} \hat{z}_T - \hat{z}_1 \hat{z}_0 \right).$$

Therefore

$$|G_{12}| = \left| \overline{\hat{z}'\hat{z}} \right| = \frac{1}{T-1} \left| \sum_{k=1}^{T-1} \hat{z}_k \hat{z}'_k \right| = \frac{1}{2(T-1)\Delta t} \left( |\hat{z}_{T-1}\hat{z}_T - \hat{z}_1\hat{z}_0| \right) = \frac{C_{\mathrm{bd}}}{L}. \tag{32}$$

**Step 3 (position energy and variance floor).** By definition,

$$\overline{\hat{z}^2} = \frac{1}{T-1} \sum_{k=1}^{T-1} \hat{z}_k^2 = \frac{T+1}{T-1} \cdot \frac{1}{T+1} \sum_{k=0}^{T} \hat{z}_k^2 - \frac{\hat{z}_0^2 + \hat{z}_T^2}{T-1}.$$

Since $\frac{1}{T+1} \sum_{k=0}^{T} \hat{z}_k^2 = \widehat{\mathrm{Var}}(\hat{z}) + \hat{\mu}^2 \geq \widehat{\mathrm{Var}}(\hat{z}) \geq \tau^2$, we obtain (28).

**Step 4 (combine).** Combining (31), Step 1, (32), and Step 3 yields (27). Combining (27) with (29) yields (30). $\square$

*Remark* D.2 (Positivity of the lower bound and sufficient window length). With the centered stencil, the design cross term telescopes to a boundary quantity:

$$G_{12} = \frac{1}{T-1} \sum_{k=1}^{T-1} \hat{z}_k \hat{z}'_k = \frac{1}{2L} \left( \hat{z}_{T-1}\hat{z}_T - \hat{z}_1\hat{z}_0 \right), \qquad L := (T-1)\Delta t.$$

Hence, with $C_{\mathrm{bd}} := \frac{1}{2} |\hat{z}_{T-1}\hat{z}_T - \hat{z}_1\hat{z}_0|$, we have the *exact* identity $|G_{12}| = C_{\mathrm{bd}}/L$. Consequently, the explicit lower bound in (30) can become vacuous when the effective window $L$ is short or when the *boundary products are mismatched*, i.e., $|\hat{z}_{T-1}\hat{z}_T - \hat{z}_1\hat{z}_0|$ is large, even though $\lambda_{\min}(G) \geq 0$ always holds.

Let $m := \min\{\rho v_\star, c_1 \tau^2\} > 0$. For any target margin fraction $\alpha \in (0,1)$, the condition

$$L \geq \frac{C_{\mathrm{bd}}}{(1-\alpha)m} \quad \implies \quad \lambda_{\min}(G) \geq \alpha m > 0$$

follows immediately from $\lambda_{\min}(G) \geq m - C_{\mathrm{bd}}/L$. In particular, choosing $\alpha = \frac{1}{2}$ yields the simple sufficient condition

$$L \geq L_0 := \frac{2 C_{\mathrm{bd}}}{m} \quad \implies \quad \lambda_{\min}(G) \geq \tfrac{1}{2} m > 0.$$

Thus, sufficiently long windows ensure a strictly positive conditioning margin whenever $m > 0$. In the strictly underdamped regime, the phase-agnostic choice $L \geq 2P$ in Theorem 4.3 guarantees level-set slope coverage (i.e., $\rho v_\star > 0$ for a suitable interval $U$), and increasing $L$ further shrinks the boundary correction $C_{\mathrm{bd}}/L$. In practice, one can additionally tighten the bound by choosing/cropping windows that reduce $C_{\mathrm{bd}}$.

**Practical strategies for improving conditioning.** Lemma D.1 bounds the conditioning of the design Gram matrix $G = \frac{1}{T-1} \sum_{k=1}^{T-1} X_k^\top X_k$ in terms of two *in-window energy* quantities and one *boundary* quantity:

$$\lambda_{\min}(G) \gtrsim m - \frac{C_{\mathrm{bd}}}{L}, \qquad m := \min\{\rho v_\star, c_1 \tau^2\}, \qquad C_{\mathrm{bd}} := \tfrac{1}{2} |\hat{z}_{T-1}\hat{z}_T - \hat{z}_1\hat{z}_0|, \quad L := (T-1)\Delta t.$$

With the centered stencil, $|G_{12}| = C_{\mathrm{bd}}/L$, so conditioning can be improved by (i) reducing the boundary mismatch $C_{\mathrm{bd}}$, (ii) increasing the slope-energy coverage term $\rho\, v_\star$, and/or (iii) increasing the position/variance term $c_1\tau^2$. Below are simple, objective-preserving data-arrangement and training strategies that target each term.

**(I) Reduce $C_{\mathrm{bd}}$ (boundary mismatch / cross term).**

- **Boundary-matched window selection / cropping.** If longer clips are available, form training windows by scanning candidate subwindows of the desired length $L$ and selecting those with small boundary mismatch $|\hat{z}_{T-1}\hat{z}_T - \hat{z}_1\hat{z}_0|$ (equivalently, small $C_{\mathrm{bd}}$). In oscillatory regimes, a practical heuristic is to choose endpoints near the same reference phase (e.g., two equilibrium crossings with the same velocity sign), or simply minimize the above score over a small set of candidate offsets.

- **Time-shift (multi-window) augmentation from long trajectories.** Because the ODE is translation-invariant in time, one may extract multiple subwindows (random offsets) from each long clip and treat them as separate training samples. When stacking multiple shifted windows, the aggregated cross term is the average of the per-window boundary mismatches, which often partially cancels across offsets, while the diagonal terms $G_{11}, G_{22}$ average nonnegative energies.

**(II) Increase $\rho\, v_\star$ (coverage fraction and slope energy on the covered set).**

- **Prefer windows with repeated crossings of the coverage interval $U$.** Since $\rho$ is the fraction of indices with $\hat{z}_k \in U$, choose $U$ in regions the trajectory visits frequently (e.g., near equilibrium for underdamped oscillations), and prefer longer windows when feasible. In the strictly underdamped regime, the phase-agnostic rule $L \geq 2P$ (Theorem 4.3) guarantees such coverage for a suitable interval $U$.

- **Avoid near-stationary windows (boost $v_\star$).** Windows dominated by turning points have $\hat{z}' \approx 0$, yielding small $v_\star$ even if $\rho$ is large. As a simple filter, discard or downweight candidate windows whose mean squared slope $\frac{1}{T-1}\sum_{k=1}^{T-1}(\hat{z}'_k)^2$ falls below a threshold.

- **Exploit multi-trajectory diversity.** If single trajectories provide poor slope separation at shared state levels, aggregate multiple trajectories (or multiple initial conditions) so that the covered set $U$ is realized with a richer range of slopes, increasing the effective $\rho\, v_\star$.

**(III) Increase $c_1\tau^2$ (prevent collapse via a variance floor).**

- **Maintain a nontrivial variance floor in the latent.** The term $c_1\tau^2$ arises from the lower bound on $\overline{\hat{z}^2}$ induced by the variance-floor regularizer. In practice, choose $\tau > 0$ and $\lambda_{\mathrm{var}}$ large enough to prevent latent collapse (especially in non-oscillatory regimes), while noting that overall parameter estimates are typically insensitive to the absolute scale due to homogeneity of the governing ODE.

## E. Constant term $g$ (equilibrium and discrete-time intercept)

This appendix serves two purposes. First, it shows that a constant forcing term $g$ can be removed by centering the state around its equilibrium when $\gamma_0 \neq 0$, so the structural identifiability results for $(\gamma_1, \gamma_0)$ remain unchanged. Second, it gives the corresponding discrete-time centered-stencil AR(2) form with an intercept, which is useful if one chooses to estimate $g$ directly from sampled latents.

We emphasize that, unlike $(\gamma_1, \gamma_0)$, the constant $g$ is not identifiable from raw video up to arbitrary affine latent reparameterization, because under an affine reparameterization $\hat{z} = az + b$ $(a \neq 0)$, the dynamical coefficients $(\gamma_1, \gamma_0)$ are invariant while the constant transforms as $\hat{g} = ag - \gamma_0 b$. The inverse map below should therefore be read as recovering $g$ in a fixed latent coordinate, not as a claim that $g$ itself is identifiable; recovering the physical $g$ requires additional calibration fixing the scale and origin of the state.

**Equilibrium shift (centering).** If $\gamma_0 \neq 0$, define $z^\star := -g/\gamma_0$ and set $y := z - z^\star$. Then

$$y'' + \gamma_1 y' + \gamma_0 y = 0,$$

so all homogeneous identifiability statements apply to $y$ unchanged.

**Centered-stencil AR(2) with intercept and inverse map.** With the centered stencil (2), setting the residual $r_k = 0$ yields

$$\left(1 + \frac{\gamma_1 \Delta t}{2}\right) \hat{z}_{k+1}$$
$$= \left(2 - \gamma_0 \Delta t^2\right) \hat{z}_k - \left(1 - \frac{\gamma_1 \Delta t}{2}\right) \hat{z}_{k-1} - \Delta t^2 \, g.$$

Dividing by $\left(1 + \frac{\gamma_1 \Delta t}{2}\right)$ we obtain an AR(2) with intercept

$$\hat{z}_{k+1} = \alpha \, \hat{z}_k + \beta \, \hat{z}_{k-1} + c,$$

where

$$\alpha = \left(2 - \gamma_0 \Delta t^2\right) / \left(1 + \frac{\gamma_1 \Delta t}{2}\right),$$
$$\beta = -\left(1 - \frac{\gamma_1 \Delta t}{2}\right) / \left(1 + \frac{\gamma_1 \Delta t}{2}\right), \tag{33}$$
$$c = -\Delta t^2 g / \left(1 + \frac{\gamma_1 \Delta t}{2}\right).$$

Conversely, letting $s := \gamma_1 \Delta t / 2$ and assuming $1 + s \neq 0$, we have

$$s = \frac{1 + \beta}{1 - \beta},$$
$$\gamma_1 = \frac{2}{\Delta t} \frac{1 + \beta}{1 - \beta},$$
$$\gamma_0 = \frac{2 - \alpha(1 + s)}{\Delta t^2},$$
$$g = -\frac{c(1 + s)}{\Delta t^2}.$$

*Time-shift invariance.* The coefficients $(\alpha, \beta, c)$ in (33) depend on $\Delta t$ and $(\gamma_1, \gamma_0, g)$ but not on the unknown origin $t_0$.

**Notes on the corner case $\gamma_0 = 0$.** If $\gamma_0 = 0$ there is no equilibrium; the homogeneous reduction by centering is unavailable, but the multi-trajectory and three-trajectory conclusions still apply with the obvious modifications.

# F. Experimental settings and additional results

## F.1. Experimental settings

### F.1.1. MODEL ARCHITECTURE

**Encoder (shared across all experiments).** We use the same *per-frame* CNN encoder for both synthetic and real-world videos. Given a grayscale clip $x \in \mathbb{R}^{B \times T \times H \times W}$, the encoder processes each frame independently and outputs a scalar latent trajectory $z \in \mathbb{R}^{B \times T \times 1}$. The network consists of $K=3$ convolutional blocks with base width 32; each block has two $3 \times 3$ convolutions followed by BatchNorm and GELU activations. We downsample using stride 2 in the first $K-1$ blocks and stride 1 in the last block. Finally, we apply global average pooling and a one-hidden-layer MLP (width 128) to map features to a scalar.

### F.1.2. OPTIMIZATION AND HYPERPARAMETERS

**Initialization.** ODE parameters are initialized as $(\gamma_1, \gamma_0) = (1, 1)$.

**Learning rates.** We use separate learning rates for the encoder and the ODE-residual/physics terms:

- Encoder learning rate: $1 \times 10^{-3}$.

- ODE residual learning rate: $1 \times 10^{-2}$.

**Regularization and time scale.** We set the variance-floor regularization weight $\lambda_{\mathrm{var}} = 1.0$. We use $\tau = 1$ in all default experiments; when studying the effect of $\tau$, we sweep across different orders of magnitude and report results in Tab. 15.

## F.2. Simulation

### F.2.1. LATENT DYNAMICS AND RENDERING

All synthetic systems render video frames from a *scalar* latent state $z(t)$ governed by the homogeneous second-order LTI ODE in Eq. (1), under four damping regimes. We consider three video generators:

1. **Pendulum** (motion): the observed pendulum angle follows Eq. (1).

2. **Intensity** (non-motion): the latent state controls the grayscale intensity of an irregular shape, with a normalized range $z \in [0.2, 1.0]$.

3. **Scale** (non-motion): the latent state controls the radius of a filled circle, with a normalized range $z \in [-10, 10]$.

**Video resolution and frame rate.** All videos are rendered at spatial resolution $H \times W = 64 \times 64$. The pendulum videos are sampled at fps $= 20$, while the intensity and scale videos are sampled at fps $= 10$.

**Ground truth parameters, initial conditions, and clip duration.** Ground-truth ODE parameters for each damping regime are listed in Tab. 1, and representative frames are shown in Figs. 2, 4. For each experimental setting, we report the initial conditions $(z_0, z_0')$ and video duration in the corresponding appendix result tables. Unless otherwise stated, all synthetic results are reported as mean±std over 5 random seeds.

## F.3. Real-world experiments

**Clean pendulum benchmark (Garcia et al. (2025))** For the clean pendulum example, we downsample the original frames to $192 \times 108$ and use fps $= 30$. For each string length, we estimate parameters on 5 different videos and report the mean±std across these videos.

**Wheel-mounted phone pendulum (Monteiro et al. (2014))** For the mounted pendulum example, we downsample the original frames to $120 \times 120$ and use fps $= 10$. Results are reported as mean±std over 5 random seeds.

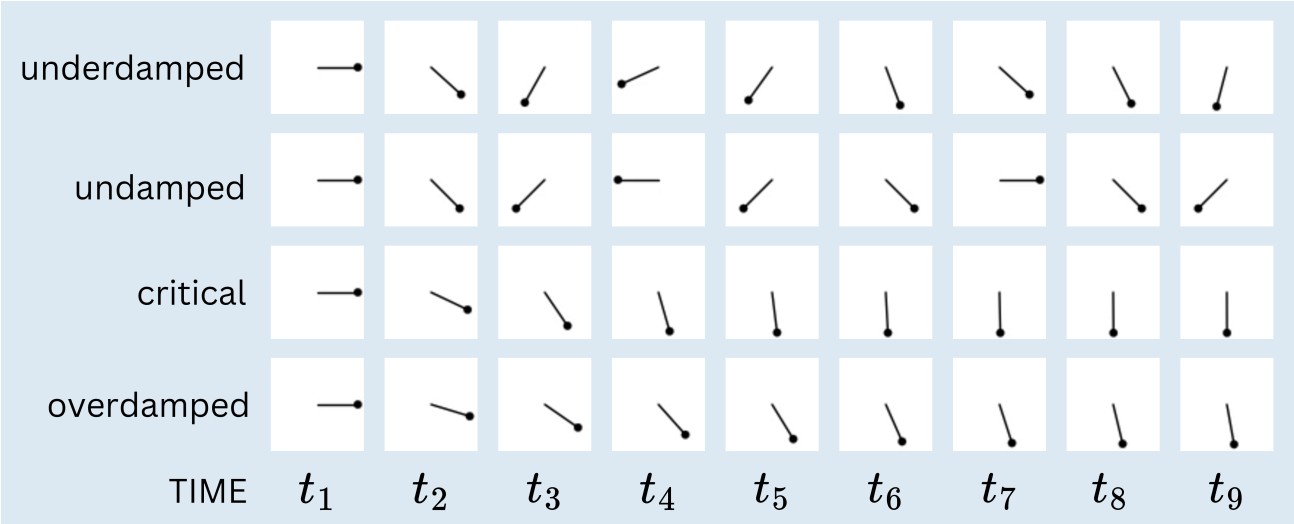

*Figure 4.* **Pendulum motion under four damping regimes (sample frames).** Read each row from left to right: the columns are equally spaced times $t_1, \ldots, t_9$. The black line is the pendulum rod and the dot is the bob. (The vertical downward position is the equilibrium.) **Underdamped:** the pendulum swings back and forth (it *oscillates*); because energy is lost to damping, the swing gets smaller over time, so each new "highest point" is lower than the previous one. **Undamped:** it also swings back and forth, but without damping the swing size stays the same from cycle to cycle. **Critical damping:** the pendulum does *not* swing past the vertical line; instead it returns to the resting position smoothly and as fast as possible without oscillating. **Overdamped:** it also returns without swinging back, but more slowly than the critically damped case.

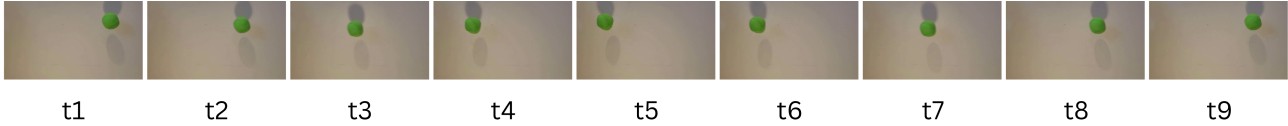

| t1 | t2 | t3 | t4 | t5 | t6 | t7 | t8 | t9 |

*Figure 5.* **LPFV (Garcia et al., 2025) real-video pendulum: sampled frames.** We show sequential frames sampled at equally spaced time steps from a representative clip in the $L = 0.45$ m setting (left to right, increasing time). The benchmark uses a static camera and a clean background to reduce exogenous variation, consistent with the state-consistency assumption discussed in the main text.

### F.3.1. CLEAN PENDULUM BENCHMARK (GARCIA ET AL. (2025))

**Pendulum dynamics and ground-truth parameters**   For small swing angles, the pendulum motion can be approximated by the linear ODE in Eq. (1). Under this approximation, the stiffness parameter satisfies

$$\gamma_0^\star = \frac{g}{L}.$$

Using $g = 9.81$ m/s$^2$, this yields

$$\gamma_0^\star \approx 21.8 \text{ s}^{-2} \text{ for } L = 0.45 \text{ m},$$
$$\gamma_0^\star \approx 10.9 \text{ s}^{-2} \text{ for } L = 0.90 \text{ m},$$
$$\gamma_0^\star \approx 6.54 \text{ s}^{-2} \text{ for } L = 1.50 \text{ m}.$$

We do not report a physical ground truth for $\gamma_1$ because it aggregates multiple dissipation sources (e.g., pivot friction and air drag) and typically requires additional calibration; in our evaluation $\gamma_1$ is treated as an effective parameter estimated from video.

**Additional discussion: Why do we outperform LPFV?**   Both LPFV (Garcia et al., 2025) and our method are encoder-only and constrain the latent trajectory by physical laws, aligning with the setting emphasized by our identifiability analysis in oscillatory (underdamped) regimes. Here we refine the comparison by focusing on (i) the discrete-time derivative construction used inside the ODE residual and (ii) the regularization used to prevent latent collapse.

**(i) Discrete derivatives: Euler / one-sided stencils vs. centered stencils.** Let $\Delta t$ be the frame interval and let $\{\hat{z}_k\}_{k=0}^T$ denote the scalar latent predicted by the encoder at times $t_k = k\Delta t$. LPFV constructs the physics constraint using an Euler-style update and one-sided (consecutive-frame) differences. In particular, it uses the first derivative approximation

$$\hat{z}_k' \approx \frac{\hat{z}_{k+1} - \hat{z}_k}{\Delta t} \qquad \text{or} \qquad \hat{z}_k' \approx \frac{\hat{z}_k - \hat{z}_{k-1}}{\Delta t},$$

which is first-order accurate:

$$\hat{z}_k' = z'(t_k) + \mathcal{O}(\Delta t).$$

A consistent second-derivative approximation induced by one-sided differences is, for example,

$$\hat{z}_k'' \approx \frac{\hat{z}_{k+1} - 2\hat{z}_k + \hat{z}_{k-1}}{\Delta t^2},$$

but when combined with an Euler / one-step formulation the effective residual is still governed by a first-order discretization in $\Delta t$.

In contrast, our objective explicitly uses centered finite differences for both first and second derivatives:

$$\hat{z}_k' = \frac{\hat{z}_{k+1} - \hat{z}_{k-1}}{2\Delta t}, \qquad \hat{z}_k'' = \frac{\hat{z}_{k+1} - 2\hat{z}_k + \hat{z}_{k-1}}{\Delta t^2}.$$

These centered stencils are second-order accurate:

$$\hat{z}_k' = z'(t_k) + \mathcal{O}(\Delta t^2), \qquad \hat{z}_k'' = z''(t_k) + \mathcal{O}(\Delta t^2).$$

Since both methods ultimately estimate parameters by minimizing a discrete ODE residual, higher-order derivative estimates can reduce discretization bias and mitigate noise amplification from differencing, which is particularly relevant in real videos where $\hat{z}_k$ is noisy. This is a direct, plausible mechanism for our consistently lower error compared with LPFV under the same benchmark protocol.

**(ii) Regularization to prevent collapse: variance-floor vs. KL prior.** Both approaches include explicit regularization to avoid degenerate latents. LPFV uses a KL divergence term encouraging a Gaussian prior, typically $\mathcal{N}(0, 1)$, while we use a variance-floor penalty that enforces a minimum within-clip latent variance. Importantly, our synthetic ablation (Tab. 4) shows that in the underdamped case *both* regularizers yield strong recovery, suggesting that the performance gap on the real-video pendulum benchmark is unlikely to be driven solely by the choice of anti-collapse regularizer. Instead, the discretization/derivative construction in (i) provides a more direct explanation consistent with the empirical trend.

*Remark* F.1. We emphasize that the above factors are meant to capture the most salient and theoretically grounded differences. Other implementation details (e.g., preprocessing and optimization hyperparameters) may further contribute, but the discretization order and the resulting numerical properties of the ODE residual are the most immediate mechanisms affecting single-trajectory regression from video.

F.3.2. WHEEL-MOUNTED PHONE PENDULUM (MONTEIRO ET AL. (2014))

**Model and mapping to identification parameters.** In this setup, a smartphone of mass $m$ is attached near the rim of a bicycle wheel, producing a physical pendulum about the wheel axis. Let $\theta(t)$ denote the angular displacement (radians), and let $\omega(t) = \dot{\theta}(t)$. A rigid-body model can be written as

$$I\dot{\omega}(t) = -R\,mg\sin\theta(t) + \tau_{\text{diss}}(\omega, t),$$

where $R$ is the offset from the wheel center to the phone center of mass, $I$ is the total rotational inertia of the wheel-plus-phone system, and $\tau_{\text{diss}}$ summarizes non-conservative torques (e.g., rolling resistance / bearing friction, and potentially air drag). In the notation used in our apparatus notes, the dissipation term may be represented as $Mr$; here we keep the more explicit functional form $\tau_{\text{diss}}(\omega, t)$ to emphasize that it can depend on velocity and time.

For small oscillations, we use the standard approximation

$$\sin\theta(t) \approx \theta(t),$$

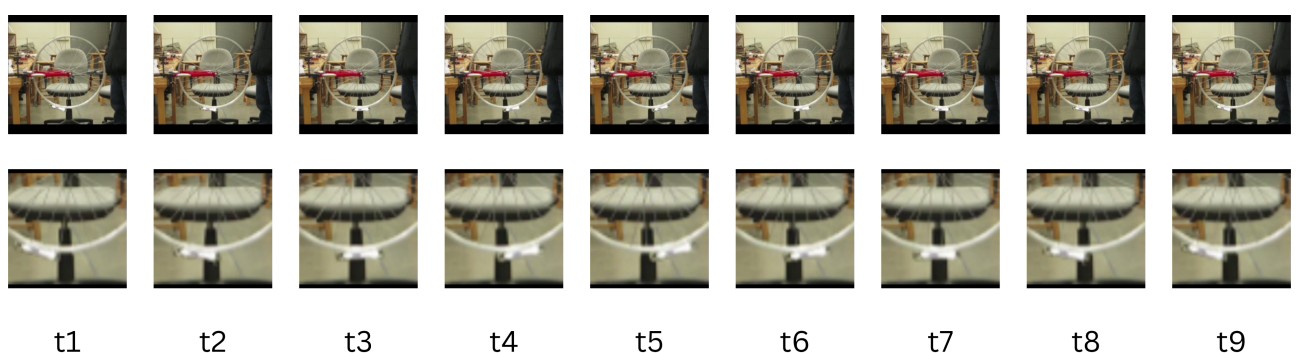

*Figure 6.* **Sampled frames: wheel-mounted phone pendulum (noisy background).** A smartphone attached near the rim of a bicycle wheel swings in a vertical plane, forming a physical pendulum. We show sequential frames sampled at equally spaced time steps from a clip (left to right, increasing time). The first row shows the raw capture; the second row shows the cropped/resized version used in our experiments.

which yields

$$I\,\theta''(t) + R\,mg\,\theta(t) = \tau_{\text{diss}}(\theta'(t), t).$$

To connect this physical model to our identification ODE in Eq. (1), we introduce an *effective linear damping* approximation over the clip duration:

$$\tau_{\text{diss}}(\theta'(t), t) \approx -c\,\theta'(t),$$

where $c \geq 0$ is an unknown effective damping coefficient that aggregates multiple dissipation sources. Substituting this approximation gives the linear ODE

$$I\,\theta''(t) + c\,\theta'(t) + R\,mg\,\theta(t) = 0.$$

Dividing by $I$ yields

$$\theta''(t) + \frac{c}{I}\,\theta'(t) + \frac{mgR}{I}\,\theta(t) = 0.$$

Matching the linearized dynamics to our identification model in Eq. (1),

$$z''(t) + \gamma_1 z'(t) + \gamma_0 z(t) = 0,$$

we obtain

$$\gamma_0^\star = \frac{mgR}{I}, \qquad \gamma_1 \approx \frac{c}{I}.$$

We emphasize that $\gamma_1$ here is an *effective* damping parameter: it depends on the validity of the approximation $\tau_{\text{diss}}(\theta', t) \approx -c\,\theta'$ over the chosen clip duration.

**Ground truth and interpretation.** We compute the total inertia as

$$I = I_w + mR^2,$$

where $I_w$ is the wheel inertia. Using the measured setup parameters

$$R = 0.300\,\text{m}, \qquad m = 0.146\,\text{kg}, \qquad I_w = 0.0388\,\text{kg} \cdot \text{m}^2,$$

we obtain

$$I = 0.0388 + 0.146 \times (0.300)^2 \approx 0.0520\,\text{kg} \cdot \text{m}^2.$$

Therefore,

$$\gamma_0^\star = \frac{mgR}{I} = \frac{0.146 \times 9.81 \times 0.300}{0.0520} \approx 8.26\,\text{s}^{-2}.$$

Unlike $\gamma_0^\star$, which is determined by geometry and inertia, the damping parameter depends on non-conservative effects (bearing friction, rolling resistance, air drag) that are not uniquely specified by the apparatus geometry and may vary across time scales. Moreover, these effects are not necessarily well-described by a constant viscous torque $-c\,\theta'$ without additional instrumentation and calibration. Accordingly, we do not report a physical ground truth for $\gamma_1$ and treat it as an effective parameter estimated from video.

**Interpreting estimation error.** Our measured estimate $\hat{\gamma}_0$ can differ from $\gamma_0^\star$ due to (i) the small-angle approximation $\sin\theta \approx \theta$, (ii) deviations of $\tau_{\mathrm{diss}}$ from a constant linear damping model over the clip duration, and (iii) discretization and measurement noise because derivatives are estimated from pixel observations.

## F.4. State consistency stress test

The goal of this section is to characterize the practical boundary of the state-consistency assumption (Def. 3.1). Since this assumption cannot be directly verified from raw pixels, we instead probe empirically how far the finite-sample estimator tolerates deviations from the exact-theorem regime. We keep the underdamped pendulum ODE fixed and perturb only the observation model with four categories of exogenous nuisance—Background Noise, Moving Clutter, Camera Jitter, and Brightness—each at three severity levels (L1–L3). Sample frames from the perturbed videos are shown in Fig. 7; the corresponding per-seed parameter estimates are reported in Tab. 6; per-category sample frames for all five seeds are shown in Figs. 8–11 (each row shows 10 equally spaced frames from $t = 0$ to $t = 1.5\pi$ for one seed).

*Table 6.* Detailed per-seed results for the state consistency stress test (underdamped pendulum). Ground truth: $(\gamma_0, \gamma_1) = (4.0016, 0.08)$. The baseline (no nuisance) estimate from clean videos is $\hat{\gamma}_0 = 4.0081 \pm 0.0003$ and $\hat{\gamma}_1 = 0.0788 \pm 0.0005$. Entries in red indicate individual runs that deviated noticeably from the ground truth. The rightmost columns report the mean $\pm$ std across the 5 seeds. For Brightness Level 2, one of the five seeds converged to a poor local optimum; excluding it, the remaining four seeds yield $\hat{\gamma}_0 = 4.0070 \pm 0.0114$ and $\hat{\gamma}_1 = 0.0680 \pm 0.0111$. See Fig. 7 for sample frames (seed 1 shown).

| | | SEED 1 | | SEED 2 | | SEED 3 | | SEED 4 | | SEED 5 | | MEAN $\pm$ STD | |
|---|---|---|---|---|---|---|---|---|---|---|---|---|---|
| CATEGORY | LEVEL | $\hat{\gamma}_0$ | $\hat{\gamma}_1$ | $\hat{\gamma}_0$ | $\hat{\gamma}_1$ | $\hat{\gamma}_0$ | $\hat{\gamma}_1$ | $\hat{\gamma}_0$ | $\hat{\gamma}_1$ | $\hat{\gamma}_0$ | $\hat{\gamma}_1$ | $\hat{\gamma}_0$ | $\hat{\gamma}_1$ |
| | L1 | 4.0090 | 0.0785 | 4.0084 | 0.0815 | 4.0033 | 0.0796 | 4.0022 | 0.0801 | 4.0026 | 0.0804 | $4.0051 \pm 0.0033$ | $0.0800 \pm 0.0011$ |
| BG-NOISE | L2 | 4.0025 | 0.0807 | 4.0029 | 0.0811 | 4.0081 | 0.0811 | 4.0077 | 0.0792 | 4.0029 | 0.0805 | $4.0048 \pm 0.0028$ | $0.0805 \pm 0.0008$ |
| | L3 | 4.0039 | 0.0813 | 4.0039 | 0.0806 | 4.0028 | 0.0813 | 4.0049 | 0.0810 | 4.0042 | 0.0817 | $4.0039 \pm 0.0008$ | $0.0812 \pm 0.0004$ |
| | L1 | 4.0072 | 0.0798 | 4.0112 | 0.0769 | 4.0061 | 0.0793 | 4.0070 | 0.0796 | 4.0132 | 0.0790 | $4.0089 \pm 0.0031$ | $0.0789 \pm 0.0012$ |
| MV-CLUTTER | L2 | 4.0082 | 0.0812 | 3.9988 | 0.0801 | 3.9926 | 0.0871 | 4.0136 | 0.0768 | 4.0137 | 0.0696 | $4.0054 \pm 0.0094$ | $0.0790 \pm 0.0064$ |
| | L3 | 4.0011 | 0.0559 | 2.4322 | −2.6736 | 3.8276 | −0.1967 | 3.9565 | −0.2040 | 2.9499 | 0.1525 | $3.4335 \pm 0.7049$ | $-0.5732 \pm 1.1845$ |
| | L1 | 4.0025 | 0.0801 | 4.0024 | 0.0807 | 4.0030 | 0.0796 | 4.0019 | 0.0794 | 4.0021 | 0.0796 | $4.0024 \pm 0.0004$ | $0.0799 \pm 0.0005$ |
| CAM-JITTER | L2 | 4.0034 | 0.0803 | 4.0022 | 0.0799 | 4.0026 | 0.0794 | 4.0053 | 0.0798 | 4.0020 | 0.0798 | $4.0031 \pm 0.0013$ | $0.0798 \pm 0.0003$ |
| | L3 | 4.0021 | 0.0805 | 4.0031 | 0.0799 | 4.0020 | 0.0795 | 4.0031 | 0.0804 | 4.0025 | 0.0783 | $4.0026 \pm 0.0005$ | $0.0797 \pm 0.0009$ |
| | L1 | 4.0116 | 0.0776 | 4.0107 | 0.0788 | 4.0108 | 0.0779 | 4.0088 | 0.0782 | 4.0090 | 0.0773 | $4.0102 \pm 0.0012$ | $0.0780 \pm 0.0006$ |
| BRIGHTNESS | L2 | 3.4615 | −3.0378 | 4.0093 | 0.0521 | 3.9902 | 0.0758 | 4.0140 | 0.0684 | 4.0145 | 0.0755 | $3.8979 \pm 0.2442$ | $-0.5532 \pm 1.3890$ |
| | L3 | 4.0228 | 0.0632 | 3.9969 | 0.0404 | 4.0256 | 0.0441 | 4.0050 | 0.0444 | 4.0020 | 0.0011 | $4.0105 \pm 0.0129$ | $0.0386 \pm 0.0228$ |

**Findings.** Background noise and camera jitter remain robust across all three severities: estimates stay close to the no-nuisance baseline. Moving clutter is tolerated at weak and moderate levels (L1, L2) but breaks at L3, with most seeds diverging. Brightness is mostly tolerable at L1 but unstable at higher severities: at L2, one of the five seeds converges to a poor local optimum (excluding it, the remaining four yield $\hat{\gamma}_0 = 4.0070 \pm 0.0114$ and $\hat{\gamma}_1 = 0.0680 \pm 0.0111$); at L3, $\hat{\gamma}_1$ is consistently biased downward, although $\hat{\gamma}_0$ remains accurate. Overall, the picture is not all-or-nothing: the finite-sample estimator retains nontrivial robustness to modest violations of Def. 3.1, while stronger time-varying nuisance does break recovery.

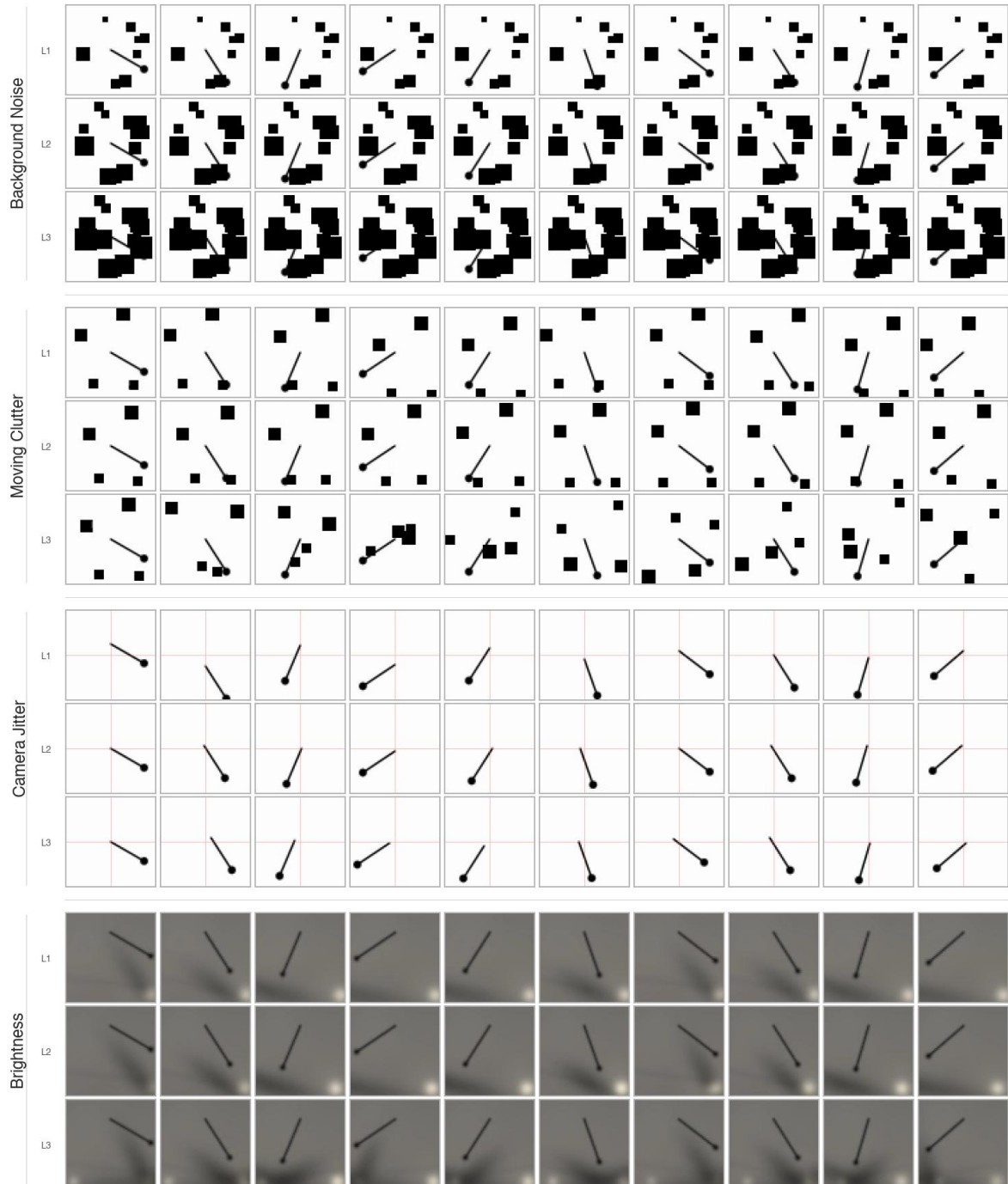

*Figure 7.* State consistency stress test: sample frames from the ablation videos (10 equally spaced frames from $t = 0$ to $t = 1.5\pi$; seed 1 shown). Each row corresponds to one severity level (L1–L3) of a visual nuisance category. The red crosshairs overlaid on the Camera Jitter rows are visualization-only reference lines (not present in the actual training videos) that mark the original frame center, making it easier to see that the pendulum is no longer centered after the simulated camera shake. **Background Noise:** static black squares of random sizes are overlaid on the background. L1: 8 squares (4–10 px); L2: 12 squares (6–14 px); L3: 16 squares (7–16 px). **Moving Clutter:** 4 black squares (4–10 px) move as distractors. L1: all squares follow a coordinated sinusoidal trajectory ($\pm 10$ px horizontal, $\pm 6$ px vertical); L2: each square moves independently along its own smooth sinusoidal path ($\pm 3$ px horizontal, $\pm 1.6$ px vertical); L3: each square bounces independently in random directions at 1.6 px/frame. **Camera Jitter:** the entire frame is shifted to simulate camera shake (exposed regions filled white). L1: smooth vertical oscillation ($\pm 8$ px, 7 full cycles); L2: 2D random walk (max $\pm 3/ \pm 2$ px, 72% step probability, 45% directional momentum); L3: aggressive 2D random walk (max $\pm 5/ \pm 4$ px, 88% step probability) with 18% chance of sudden 2–4 px burst jumps. **Brightness:** the scene is rendered with pseudo-3D lighting. L1: 1 light source with fixed position and fixed intensity; L2: 1 light source whose position and intensity both vary smoothly over time; L3: 2 independent light sources, each with smoothly varying position and intensity.

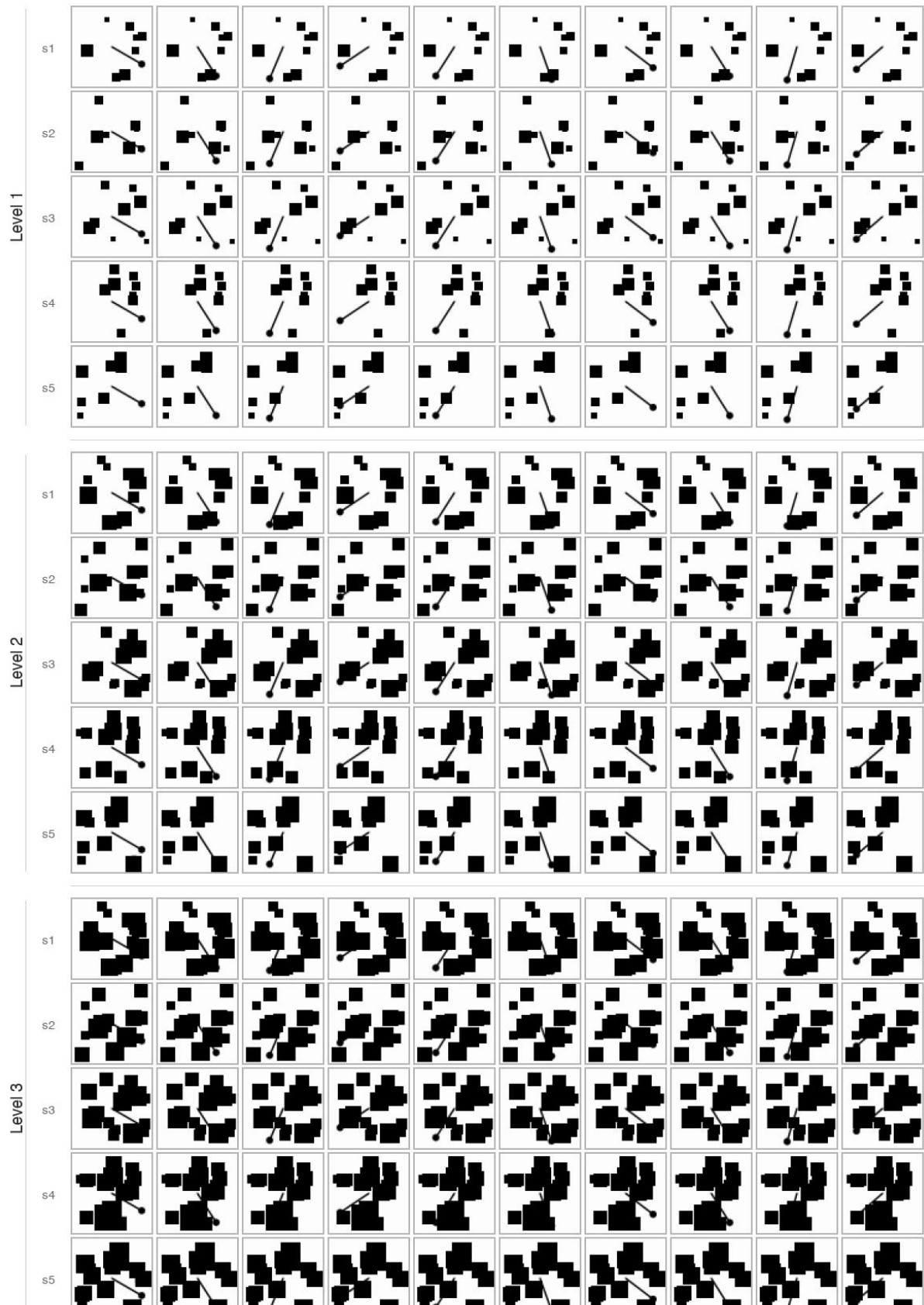

*Figure 8.* Background Noise ablation: all seeds (s1–s5) across Levels 1–3. See Fig. 7 for level definitions.

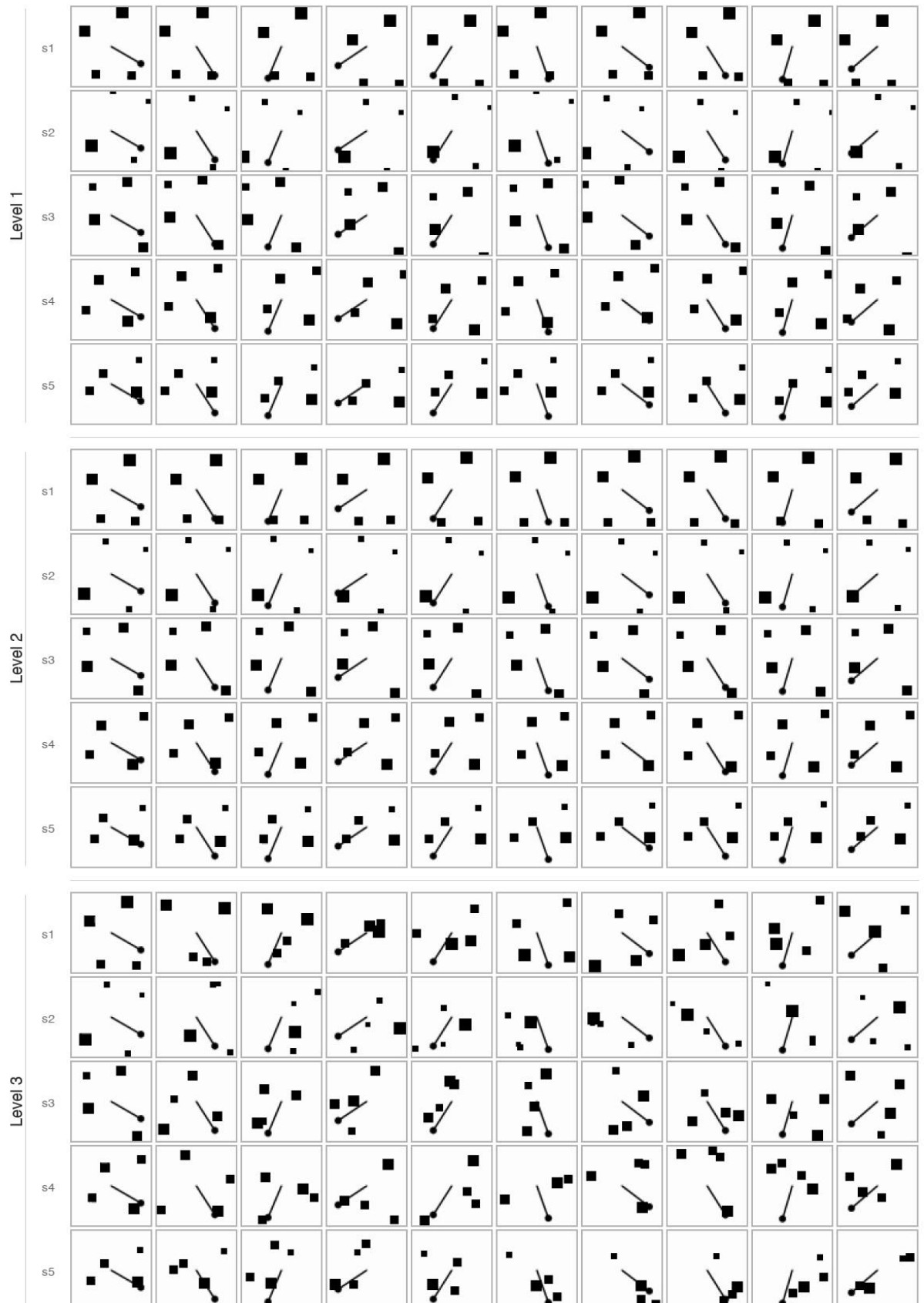

*Figure 9.* Moving Clutter ablation: all seeds (s1–s5) across Levels 1–3. See Fig. 7 for level definitions.

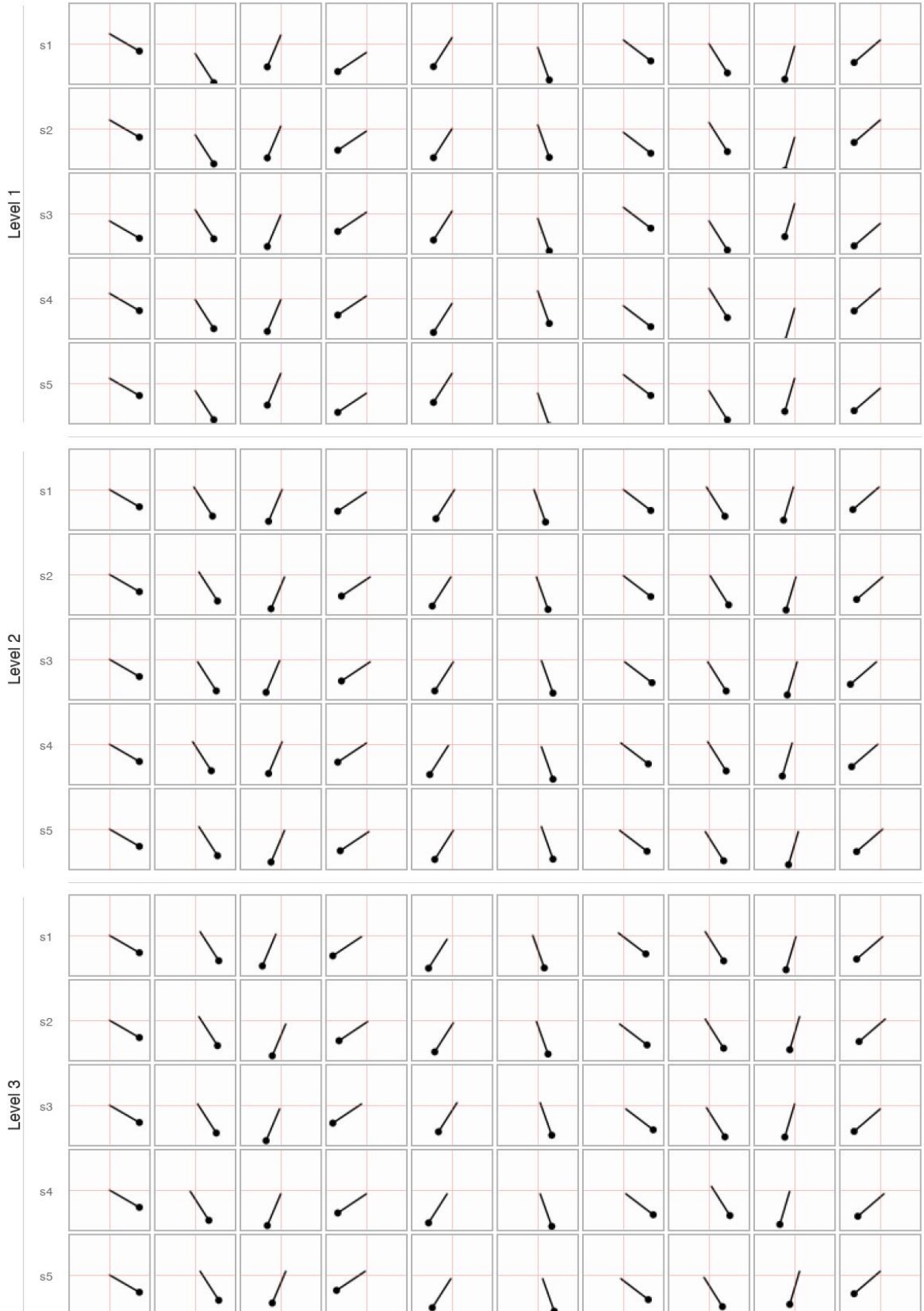

*Figure 10.* Camera Jitter ablation: all seeds (s1–s5) across Levels 1–3. See Fig. 7 for level definitions.

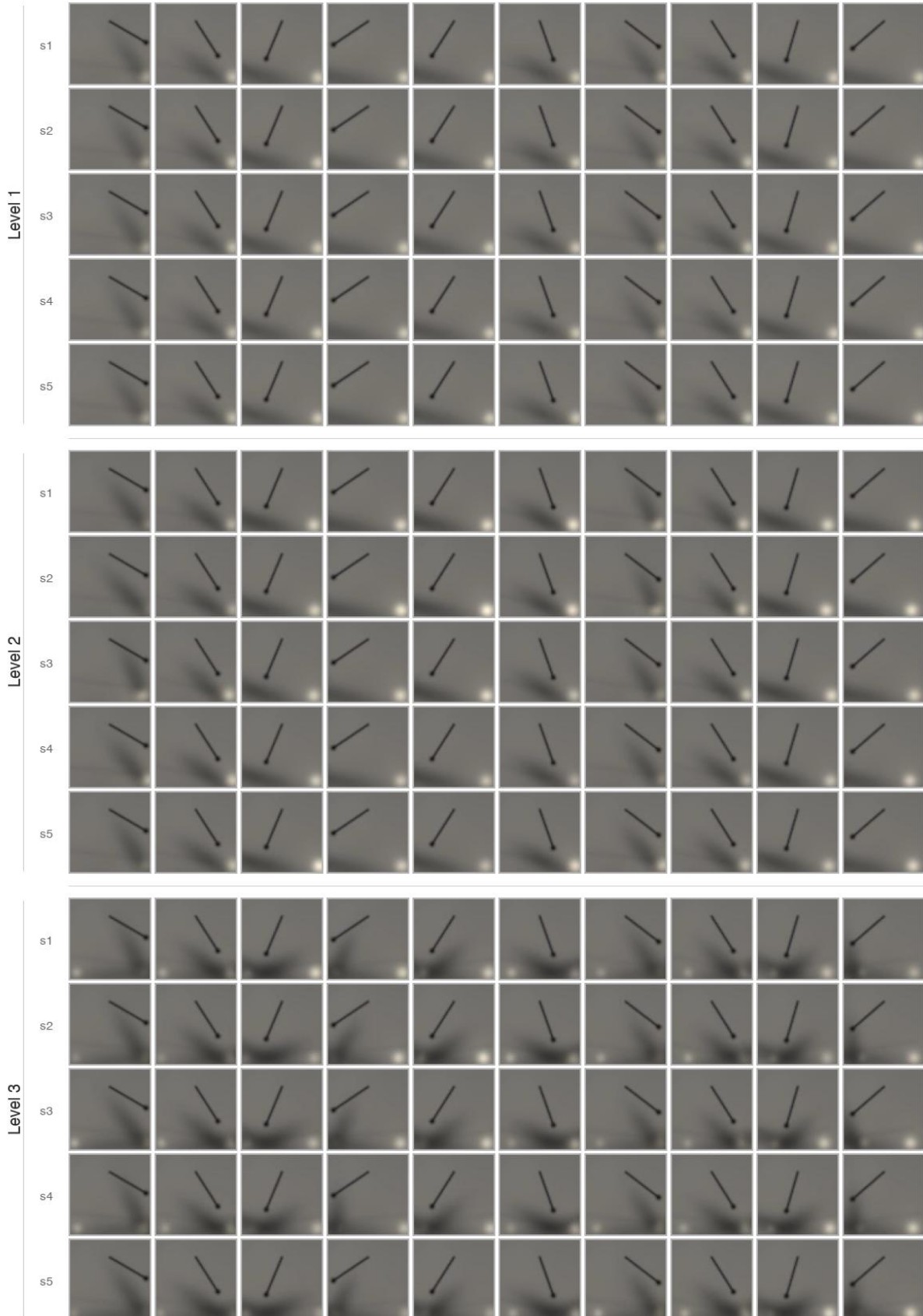

*Figure 11.* Brightness ablation: all seeds (s1–s5) across Levels 1–3. See Fig. 7 for level definitions.

## F.5. Additional simulation results

In this section, we report additional simulation results.

*Table 7.* **Pendulum underdamped case**: results of estimated parameters from **one** single trajectory. For $T = 0.5\pi$, all settings are unidentifiable; for $T = 1.5\pi$ and $T = 2.5\pi$, all settings are identifiable.

| INITIAL VALUE | TIME (S) | $\hat{\gamma}_0$ | $\hat{\gamma}_1$ |
|---|---|---|---|
| (50, 0) | $0.5\pi$ | $4.6187 \pm 0.1080$ | $-0.7503 \pm 0.4279$ |
| | $1.5\pi$ | $4.0011 \pm 0.0003$ | $0.0755 \pm 0.0005$ |
| | $2.5\pi$ | $3.9957 \pm 0.0003$ | $0.0822 \pm 0.0003$ |
| (60, 0) | $0.5\pi$ | $4.5260 \pm 0.0902$ | $-0.3445 \pm 0.2945$ |
| | $1.5\pi$ | $4.0017 \pm 0.0009$ | $0.0847 \pm 0.0004$ |
| | $2.5\pi$ | $4.0081 \pm 0.0003$ | $0.0788 \pm 0.0005$ |
| (70, 0) | $0.5\pi$ | $5.0615 \pm 0.2923$ | $-0.9726 \pm 0.4321$ |
| | $1.5\pi$ | $4.0183 \pm 0.0011$ | $0.0797 \pm 0.0003$ |
| | $2.5\pi$ | $4.0037 \pm 0.0003$ | $0.0800 \pm 0.0003$ |
| GROUND TRUTH (ALL) | | 4.0016 | 0.08 |

*Table 8.* **Pendulum undamped case**: results of estimated parameters from **one** single trajectory. For $T = 0.5\pi$, all settings are unidentifiable; for $T = 1.5\pi$ and $T = 2.5\pi$, all settings are identifiable.

| INITIAL VALUE | TIME (S) | $\hat{\gamma}_0$ | $\hat{\gamma}_1$ |
|---|---|---|---|
| (50, 0) | $0.5\pi$ | $4.7613 \pm 0.2621$ | $-0.9288 \pm 0.5538$ |
| | $1.5\pi$ | $4.0162 \pm 0.0006$ | $-0.0001 \pm 0.0001$ |
| | $2.5\pi$ | $4.0043 \pm 0.0009$ | $0.0003 \pm 0.0005$ |
| (60, 0) | $0.5\pi$ | $4.5404 \pm 0.1089$ | $-0.5516 \pm 0.3015$ |
| | $1.5\pi$ | $4.0249 \pm 0.0020$ | $-0.0012 \pm 0.0004$ |
| | $2.5\pi$ | $4.0024 \pm 0.0008$ | $0.0017 \pm 0.0004$ |
| (70, 0) | $0.5\pi$ | $5.2246 \pm 0.3527$ | $-1.0348 \pm 0.6159$ |
| | $1.5\pi$ | $4.0191 \pm 0.0007$ | $0.0003 \pm 0.0002$ |
| | $2.5\pi$ | $4.0032 \pm 0.0010$ | $0.0002 \pm 0.0004$ |
| GROUND TRUTH (ALL) | | 4 | 0 |

*Table 9.* **Pendulum critical case**: results of estimated parameters from **one** single trajectory (all unidentifiable).

| INITIAL VALUE | TIME (S) | $\hat{\gamma}_0$ | $\hat{\gamma}_1$ |
|---|---|---|---|
| (50, 0) | 1 | $8.3073 \pm 0.3260$ | $0.5131 \pm 0.2225$ |
| | 2 | $2.5257 \pm 0.1637$ | $0.7948 \pm 0.4642$ |
| | 3 | $1.3892 \pm 0.0395$ | $0.8086 \pm 0.0670$ |
| (60, 0) | 1 | $6.6972 \pm 0.2655$ | $0.5538 \pm 0.1308$ |
| | 2 | $3.1256 \pm 0.3415$ | $1.7637 \pm 0.4217$ |
| | 3 | $1.3927 \pm 0.1343$ | $0.7721 \pm 0.4944$ |
| (70, 0) | 1 | $6.7580 \pm 0.5776$ | $0.8701 \pm 0.3439$ |
| | 2 | $3.0945 \pm 0.1550$ | $1.5450 \pm 0.2624$ |
| | 3 | $1.3124 \pm 0.1115$ | $0.5736 \pm 0.3260$ |
| GROUND TRUTH (ALL) | | 4 | 4 |

*Table 10.* **Pendulum overdamped case**: results of estimated parameters from **one** single trajectory (all unidentifiable).

| INITIAL VALUE | TIME (S) | $\hat{\gamma}_0$ | $\hat{\gamma}_1$ |
|---|---|---|---|
| (50,0) | 1 | $8.4446 \pm 0.5896$ | $1.0713 \pm 0.2950$ |
| | 2 | $2.3675 \pm 0.0591$ | $0.9689 \pm 0.2481$ |
| | 3 | $1.3218 \pm 0.1109$ | $0.9170 \pm 0.2339$ |
| (60,0) | 1 | $6.7109 \pm 0.4008$ | $0.8854 \pm 0.4445$ |
| | 2 | $3.3553 \pm 0.2659$ | $1.9423 \pm 0.2899$ |
| | 3 | $1.5812 \pm 0.1313$ | $1.2932 \pm 0.2085$ |
| (70,0) | 1 | $6.5106 \pm 0.6147$ | $0.8947 \pm 0.6298$ |
| | 2 | $2.8975 \pm 0.2008$ | $1.4462 \pm 0.2637$ |
| | 3 | $1.6166 \pm 0.1250$ | $1.4672 \pm 0.1912$ |
| GROUND TRUTH (ALL) | | 4 | 5 |

*Table 11.* **Critical case**: results of estimated parameters from **three** trajectories under different initial values.

| CASE | INITIAL VALUE | TIME (S) | $\hat{\gamma}_0$ | $\hat{\gamma}_1$ |
|---|---|---|---|---|
| IDENTIFIABLE | (50,-200) | 2 | | |
| | (50,-600) | 2 | $3.9946 \pm 0.0033$ | $3.9891 \pm 0.0044$ |
| | (50,-1000) | 2 | | |
| | (60,-200) | 2 | | |
| | (60,-600) | 2 | $3.9633 \pm 0.0032$ | $3.9908 \pm 0.0026$ |
| | (60,-1000) | 2 | | |
| | (70,-200) | 2 | | |
| | (70,-600) | 2 | $4.0218 \pm 0.0040$ | $4.0352 \pm 0.0017$ |
| | (70,-1000) | 2 | | |
| UNIDENTIFIABLE | (50,0) | 2 | | |
| | (50,-100) | 2 | $2.4572 \pm 0.0299$ | $2.9322 \pm 0.0366$ |
| | (-50,100) | 2 | | |
| | (60,0) | 2 | | |
| | (60,-100) | 2 | $2.4919 \pm 0.1090$ | $3.0487 \pm 0.0815$ |
| | (-60,100) | 2 | | |
| | (70,0) | 2 | | |
| | (70,-100) | 2 | $2.3462 \pm 0.0437$ | $2.7642 \pm 0.0787$ |
| | (-70,100) | 2 | | |
| GROUND TRUTH (ALL) | | | 4 | 4 |

*Table 12.* **Overdamped case**: results of estimated parameters from **three** trajectories under different initial values.

| CASE | INITIAL VALUE | TIME (S) | $\hat{\gamma}_0$ | $\hat{\gamma}_1$ |
|------|---------------|----------|------------------|------------------|
| IDENTIFIABLE | (50,-200)
(50,-600)
(50,-1000) | 2
2
2 | $3.9569 \pm 0.0024$ | $4.9413 \pm 0.0004$ |
| | (60,-200)
(60,-600)
(60,-1000) | 2
2
2 | $4.0311 \pm 0.0082$ | $4.9912 \pm 0.0057$ |
| | (70,-200)
(70,-600)
(70,-1000) | 2
2
2 | $3.9733 \pm 0.0033$ | $4.9723 \pm 0.0010$ |
| UNIDENTIFIABLE | (50,0)
(50,-200)
(-50,200) | 2
2
2 | $5.8776 \pm 0.1955$ | $6.0282 \pm 0.1127$ |
| | (60,0)
(60,-200)
(-60,200) | 2
2
2 | $2.8135 \pm 0.0166$ | $4.1708 \pm 0.0178$ |
| | (70,0)
(70,-200)
(-70,200) | 2
2
2 | $2.6314 \pm 0.0080$ | $4.0396 \pm 0.0095$ |
| GROUND TRUTH (ALL) | | | 4 | 5 |

*Table 13.* Additional results for the **intensity** system under different damping regimes.

| REGIME | INITIAL VALUE | TIME (S) | $\hat{\gamma}_0$ | $\hat{\gamma}_1$ |
|--------|---------------|----------|------------------|------------------|
| UNDERDAMPED | $(0.4, 0)$
$(0.6, 0)$
$(0.8, 0)$
$(1.0, 0)$
GROUND TRUTH | $3\pi$
$3\pi$
$3\pi$
$3\pi$
– | $4.006 \pm 0.004$
$4.007 \pm 0.007$
$4.010 \pm 0.003$
$4.008 \pm 0.004$
4.0016 | $0.090 \pm 0.008$
$0.088 \pm 0.004$
$0.084 \pm 0.005$
$0.083 \pm 0.003$
0.08 |
| UNDAMPED | $(0.4, 0)$
$(0.6, 0)$
$(0.8, 0)$
$(1.0, 0)$
GROUND TRUTH | $3\pi$
$3\pi$
$3\pi$
$3\pi$
– | $3.971 \pm 0.001$
$3.970 \pm 0.001$
$3.979 \pm 0.012$
$3.975 \pm 0.002$
4 | $0.005 \pm 0.003$
$0.002 \pm 0.004$
$-0.001 \pm 0.007$
$0.008 \pm 0.006$
0 |
| CRITICAL | $(0.4;\ 0, -200, 200)$
$(0.6;\ 0, -200, 200)$
$(0.8;\ 0, -200, 200)$
$(1.0;\ 0, -200, 200)$
GROUND TRUTH | 2
2
2
2
– | $3.934 \pm 0.030$
$4.031 \pm 0.022$
$4.088 \pm 0.030$
$4.059 \pm 0.014$
4 | $3.929 \pm 0.011$
$3.964 \pm 0.012$
$3.940 \pm 0.043$
$3.957 \pm 0.006$
4 |
| OVERDAMPED | $(0.4;\ 0, -200, 200)$
$(0.6;\ 0, -200, 200)$
$(0.8;\ 0, -200, 200)$
$(1.0;\ 0, -200, 200)$
GROUND TRUTH | 2.2
2.2
2.2
2.2
– | $3.929 \pm 0.020$
$3.962 \pm 0.030$
$4.055 \pm 0.129$
$3.976 \pm 0.030$
4 | $4.967 \pm 0.028$
$4.968 \pm 0.059$
$5.015 \pm 0.024$
$4.877 \pm 0.017$
5 |

*Table 14.* Additional results for the **scale** system under different damping regimes.

| REGIME | INITIAL VALUE | TIME (S) | $\hat{\gamma}_0$ | $\hat{\gamma}_1$ |
|---|---|---|---|---|
| UNDERDAMPED | $(3, 0)$ | $3\pi$ | $4.020 \pm 0.0005$ | $0.074 \pm 0.0003$ |
| | $(5, 0)$ | $3\pi$ | $4.019 \pm 0.0003$ | $0.074 \pm 0.0003$ |
| | $(7, 0)$ | $3\pi$ | $4.017 \pm 0.0056$ | $0.075 \pm 0.0016$ |
| | $(9, 0)$ | $3\pi$ | $4.020 \pm 0.0002$ | $0.074 \pm 0.0002$ |
| | GROUND TRUTH | $-$ | $4.0016$ | $0.08$ |
| UNDAMPED | $(3, 0)$ | $3\pi$ | $3.983 \pm 0.0003$ | $0.00004 \pm 0.0001$ |
| | $(5, 0)$ | $3\pi$ | $3.984 \pm 0.0021$ | $0.00006 \pm 0.0005$ |
| | $(7, 0)$ | $3\pi$ | $3.983 \pm 0.0005$ | $0.00002 \pm 0.0002$ |
| | $(9, 0)$ | $3\pi$ | $3.983 \pm 0.0003$ | $0.00012 \pm 0.0004$ |
| | GROUND TRUTH | $-$ | $4$ | $0$ |
| CRITICAL | $(3; \ 0, -200, 200)$ | $2.5$ | $4.025 \pm 0.0001$ | $4.074 \pm 0.0002$ |
| | $(5; \ 0, -200, 200)$ | $2.5$ | $4.124 \pm 0.0003$ | $3.908 \pm 0.0002$ |
| | $(7; \ 0, -200, 200)$ | $2.5$ | $4.122 \pm 0.0003$ | $3.917 \pm 0.0003$ |
| | $(9; \ 0, -200, 200)$ | $2.5$ | $4.113 \pm 0.0004$ | $4.086 \pm 0.0007$ |
| | GROUND TRUTH | $-$ | $4$ | $4$ |
| OVERDAMPED | $(3; \ 0, -200, 200)$ | $2.5$ | $4.033 \pm 0.0003$ | $4.855 \pm 0.0002$ |
| | $(5; \ 0, -200, 200)$ | $2.5$ | $4.179 \pm 0.0004$ | $4.976 \pm 0.0003$ |
| | $(7; \ 0, -200, 200)$ | $2.5$ | $4.128 \pm 0.0001$ | $4.918 \pm 0.0001$ |
| | $(9; \ 0, -200, 200)$ | $2.5$ | $4.130 \pm 0.0002$ | $4.918 \pm 0.0002$ |
| | GROUND TRUTH | $-$ | $4$ | $5$ |

*Table 15.* Effect of the variance-floor target $\tau$ on parameter estimation in the pendulum system across damping regimes.

| REGIME | $\tau$ | $\hat{\gamma}_0$ | $\hat{\gamma}_1$ |
|---|---|---|---|
| UNDER | 1 | $4.0037 \pm 0.0003$ | $0.0800 \pm 0.0003$ |
| | 10 | $4.0048 \pm 0.0004$ | $0.0797 \pm 0.0001$ |
| | 100 | $4.0048 \pm 0.0008$ | $0.0801 \pm 0.0006$ |
| UN | 1 | $4.0032 \pm 0.0010$ | $0.0002 \pm 0.0004$ |
| | 10 | $4.0029 \pm 0.0008$ | $0.0003 \pm 0.0003$ |
| | 100 | $4.0032 \pm 0.0005$ | $0.0005 \pm 0.0006$ |
| CRI | 1 | $4.0218 \pm 0.0040$ | $4.0352 \pm 0.0017$ |
| | 10 | $4.0212 \pm 0.0026$ | $4.0353 \pm 0.0015$ |
| | 100 | $4.0232 \pm 0.0025$ | $4.0406 \pm 0.0022$ |
| OVER | 1 | $3.9733 \pm 0.0033$ | $4.9723 \pm 0.0010$ |
| | 10 | $3.9735 \pm 0.0018$ | $4.9743 \pm 0.0020$ |
| | 100 | $3.9818 \pm 0.0058$ | $4.9799 \pm 0.0051$ |

