# OpenReview forum: "Physics from Video: Identifiability of Time-Invariant Second-Order ODEs under Minimal Trajectory Conditions"
_ICML.cc/2026/Conference — ICML 2026 regular_

### Official Review · Reviewer_fVSn · 2026-03-03

**Soundness:** 3
**Presentation:** 3
**Significance:** 2
**Originality:** 4
**Overall Recommendation:** 4
**Confidence:** 4

**Summary:**

This paper proposes an encoder-only pipeline that uniquely recovers the parameters of homogeneous second-order linear time-invariant ODEs from video data. They separate the ideal noiseless, continuous-time regime from the discrete, noisy regime, and provide identifiability guarantees for both regimes. For identifiability, the authors introduce level-set slope coverage, which requires that the trajectory passes through all states in some interval at least three times with distinct velocities. When this condition holds, the learned latent coordinate is shown to be locally affine to the true physical state, which leads to ODE parameter identifiability. They also introduce a variance floor regularizer that prevents the state collapse and is shown to be essential for a better conditioned regression problem. Experiments on synthetic pendulum, intensity, and scale video systems show accurate recovery when coverage holds and failure otherwise. The real-world experiments on a pendulum movie benchmark demonstrate better performance compared to competitive baselines.

**Compliance With Llm Reviewing Policy:**

Affirmed.

**Key Questions For Authors:**

1. Th. 4.5 claims that single-trajectory identification should fail in the undamped regime, yet the experiments in Sec. 5.1.1 show reliable recovery. The authors attribute this to discretization effects breaking the continuous-time symmetry. Can the authors provide a formal argument or a more careful characterization of when and why this happens?
2. Def. 3.1 requires the encoder to produce outputs that are a smooth function of the physical state alone. Isn't this a strong assumption for a neural network that takes as input raw pixels? How do the authors envision enforcing or verifying this in practice beyond the controlled capture conditions in Remark 3.2?
3. The framework is restricted to 1D latents and homogeneous second-order LTI ODEs, and all experiments are conducted on pendulum systems and synthetic scenarios. Do the authors see a path toward extending the analysis to multi-dimensional latents or nonlinear systems?

**Limitations:**

Yes, the limitations are sufficiently discussed.

**Strengths And Weaknesses:**

### Soundness
The claims in the paper are matched with the theoretical investigation as well as the experimental setup. I thoroughly read the main paper and appendix C.1, which proves single-trajectory identifiability, and did not find any error (I could not follow Sec 4.2. Finite-sample convergence with discrete-time observations and related sections in the appendix. I would appreciate a walk-through by the authors). The experiments are designed to analyze the theoretical aspects (single vs multi trajectory, under-/un-/over-/critically damped settings, coverage condition, exact parameter estimation).

However, I have a concern about the assumption structure underlying Theorem 4.2. The theorem assumes that the learned latent $\hat{z}(t)$ satisfies a homogeneous second-order linear ODE (Eq. 8). This is addressed in practice by the ODE loss. But then the state consistency (Def. 3.1) assumption is hard to verify. It requires the encoder to produce outputs that are a smooth function of **the physical state alone**. An encoder taking raw pixels as input would react to visual nuisances, such as lighting and background texture, that are unrelated to the physical state. No mechanism is provided to verify or enforce state consistency, nor the smoothness of the encoder, during training beyond Remark 3.2. I also note that the real-world experiments partially show that this is no big deal in practice but it is not formally interrogated.

The undamped regime creates a conflict between theory and practice. Theorem 4.5 establishes that single-trajectory identification should fail in this regime, yet the experiments in Sec. 5.1.1 show reliable recovery. The authors attribute this to discretization effects, which is plausible but I believe this finding/intuition should be more carefully expressed or supported by a formal argument. On a related note, the ground-truth values in Table 1 do not appear to match the dashed reference lines in Figure 3, which should be clarified.

Finally, regarding the real-world experiments, the clean pendulum benchmark follows the synthetic setting (static camera, clean background, controlled conditions) rather than stress-testing these. The more challenging wheel-mounted phone experiment is valuable, but the authors had to manually crop and resize the video to make it work, which itself suggests that state consistency is fragile under real visual conditions. The theory offers no guidance on how much visual variety is tolerable before identifiability guarantees break, which limits the practical applicability of the framework.

### Presentation
The paper is generally well-structured and written. The narrative flows nicely from motivation to problem setting, theory, and experiments. The regime-by-regime organization of both theoretical results and experiments makes it easy to track which claims correspond to which results. Figure 1 provides a helpful high-level overview of the pipeline, and the connection between the coverage condition and the experimental diagnostic is a nice bridge between theory and practice. I propose the following for improvement:
- Section 4.2 on finite-sample convergence is very dense and relies heavily on appendix references, making it difficult to follow and evaluate the contribution from the main paper alone.
- The undamped regime results deserve more discussion. The paper neither clearly claims success nor clearly claims failure there, the takeaway isn't clear to me.
- There is a disconnect between the high-level motivating question posed in the introduction (*"Can we identify the true underlying physical parameters from raw video, and if so, under what formal conditions?"*) and the actual scope of the work, which is restricted to 1D latents and homogeneous second-order LTI ODEs. This gap should be acknowledged explicitly early on.
- Important experimental details are deferred to the appendix, making the experiments difficult to assess from the main paper alone.
- Figure 1 could be improved. In particular, the RELIABLE and IDENTIFIABILITY CHECK boxes are not understandable at first glance a nt not explained in the caption.
- The definition of $\mathcal{C}^2(\mathcal{O})$ is never formally stated in the paper.
- The hat notation convention ($\hat{\cdot}$ is a placeholder for the latent produced by an encoder) is introduced in Sec. 4 but should be introduced already in Sec. 3 where hatted quantities first appear.
- It is unclear how to interpret the underdamped and undamped phase portraits in Figure 3. The caption does not provide enough guidance for a reader to understand what the axes, trajectories, and markers jointly convey.
- The critique of decoder-based methods (being prone to "appearance shortcuts") is somewhat overstated. Many decoder-based identifiability methods explicitly constrain or regularize the decoder, e.g., by restricting its functional form or imposing disentanglement objectives, to prevent such shortcuts. The paper should refer to this line of work as well and discuss their shortcomings.

### Significance
The problem this paper addresses is important. Recovering physical parameters from raw video without ground-truth labels has clear practical value, e.g., in biology and robotics. The theoretical contribution (the formal identifiability guarantees for an encoder-only pipeline that learns from videos) is meaningful. The level-set slope coverage condition is a clean and verifiable condition that replaces hard-to-check model constraints often encountered in the identifiability literature. Likewise, the variance-floor regularizer is a simple and practically useful contribution that outperforms the KL regularizer baseline.

However, I believe the significance is limited by the scope of the work. The theoretical framework is restricted to 1D latents, scalar physical states, and homogeneous second-order LTI ODEs. This setting is clean enough for analysis but far from the complexity of real physical systems. More concretely, all experiments are conducted on pendulum systems and highly controlled synthetic scenarios (intensity and scale). In my opinion, these are setups that nicely follow the theory. It is not clear to me how the main idesas (e.g., the coverage condition and state consistency) extend to multi-dimensional, nonlinear, or controlled systems. The real-world experiments partially expose this gap but do not bridge it. As a result, the work is best understood as a foundational theoretical contribution that opens a direction rather than one with immediate broad applicability.

### Originality
Overall, the mathematical tools underlying this work as well as the encoder-only pipeline are already established. Yet, the combination of ideas in this paper is novel. Applying structural identifiability analysis to an encoder-only video pipeline is a new framing. The level-set slope coverage condition is a creative geometric reformulation that, to my knowledge, is novel. The regime characterizations naturally follow the theory. The paper also builds a valuable bridge from the classic system identification literature, where the main focus is the identifiability of linear ODEs from state observations, to deep-learning based identifiability approaches. Overall, the work makes a focused and clear contribution to a well-identified gap.

---

> ### Author Rebuttal · Authors · 2026-03-30
>
> Thank you for the careful review and for the detailed comments on theory, presentation, and experiments. We found your feedback very helpful and respond point by point below.
>
> Updated figures/tables are in the anonymous link: https://conferenceml.github.io/icml2026-supplement/.
>
> **Q1.Undamped regime/Fig. 3.**
> Thank you for the careful reading and for pushing us on the undamped regime. This comment helped us improve both the figure and the explanation. The intended takeaway is now the following: in the ideal continuous-time problem, an undamped single trajectory is not structurally identifiable, exactly as Thm. 4.5 states; in the actual discrete pipeline, however, the centered-difference regression has its own discrete target, and once the window spans enough phase to make that regression well conditioned, the resulting estimates can become very close to the ground truth. To support this more formally, we added a **new appendix proposition**: for an ideal sampled undamped sinusoid, the centered-difference regression fit by our estimator has a unique zero-loss target $(\gamma_0^\Delta,\gamma_1^\Delta)=(\gamma_0+O(\Delta t^2),0)$ under a full-rank discrete design matrix. Thus, when $\Delta t$ is small, this discrete target is already very close to the continuous-time ground-truth coefficients $(\gamma_0,0)$, which explains why the estimator can become accurate on sufficiently long sampled clips. In short, the continuous-time structural result remains negative, while the discrete estimator used in practice can still succeed on long enough sampled clips. If helpful, we would be very happy to discuss the undamped case in more detail in the follow-up discussion.
>
> We made three concrete changes. (i) We replaced bottom row of Fig. 3 with a numerical continuous-time coverage check based directly on the analytic trajectory. With this update, the undamped panel is never coverage-positive, exactly as Thm. 4.5 states. (ii) We rechecked Fig. 3 against Tab. 1. The dashed reference values were correct; the main plotting issue was that the right-side $\gamma_1$ tick labels were missing in the first three top-row panels, and this is now fixed. (iii) We formalized the discrete explanation in a new appendix proposition with a detailed proof. The updated Fig. 3 is in the anonymous link, and the full derivation now is in the appendix.
>
> **Q2.State consistency.**
> We agree that Def. 3.1 is hard to verify from raw pixels, and we do not claim a generic verifier beyond controlled capture/preprocessing. Our intended claim is narrower: Def. 3.1 is the explicit structural assumption needed for the exact theory, not something arbitrary neural encoders satisfy automatically. To address the practical side more directly, we added a synthetic state-consistency stress test that keeps the underdamped ODE fixed and perturbs only the observation model. The result is encouraging but not unlimited: static background noise has almost no effect; camera jitter remains robust across all three severities; weak/moderate moving clutter is tolerated but strong moving clutter breaks recovery; dynamic lighting is mostly tolerable but less stable. Thus the practical picture is not all-or-nothing: exact state consistency is the theorem regime, but moderate violations are often survivable in practice, while stronger time-varying nuisance can indeed break the method. We also now state more explicitly that the crop/resize step in the wheel-mounted-phone example should be read as preprocessing that moves the clip closer to the modeling regime, not as evidence that the theory already handles arbitrary nuisance. The frames and result table are in the anonymous link.
>
> **Q3.Scope/follow-up work.**
> We agree that the current theory is deliberately restricted to a scalar latent state and a homogeneous second-order LTI ODE. The 1D geometry is exactly what makes the current coverage argument sharp. The natural next step is low-dimensional coupled/multi-mode linear systems. Forced dynamics and mild departures from LTI would come next. For weak nonlinearity/slow time variation, please see our response to reviewer gGhi; for broader roadmap and applications, please see our response to reviewer NWQz.
>
> **Presentation**
> 1. Sec. 4.2 density. Right after Eq. (9), we added a short guide explaining the four terms in the bound and clarifying  $\psi_{\min}$.
> 2. Undamped regime/Fig. 3. See Q1.
> 3. Intro scope. Immediately after the motivating question, we now state the scope explicitly.
> 4. Experimental details. At the start of Sec. 5, we added a paragraph summarizing main experimental settings.
> 5. Fig. 1. We updated the figure and caption, see the anonymous link.
> 6. Notation. We now define $C^2(\mathcal{O})$ and hat convention at first use.
> 7. Decoder-based critique. We softened this passage, added representative citations and discussed shortcomings.
>
> If helpful, we would be very happy to walk through Sec. 4.2 in the follow-up discussion.
>
> Thank you again for the very helpful review.

---

> > ### Author Rebuttal · Reviewer_fVSn · 2026-04-03
> >
> > Thanks for the detailed rebuttal. I have one question and one comment:
> > - Is the "new appendix proposition" written anywhere other than the rebuttal?
> > - I believe the additional experiments on state consistency and your response to my "going beyond 1D" comment mostly strengthen my points: The framework seems to work despite the violation of the state consistency, and a multi-dimensional extension seemingly requires extending the coverage-based theory, which is not straightforward to me.
> >
> > As before, I consider this work interesting for the community but its merits are rather limited; so I keep my score.

---

> > > ### Author Response · Authors · 2026-04-04
> > >
> > > Thank you for the thoughtful follow-up and for reading the rebuttal so carefully. We appreciate both the specific question and the broader comment. We respond to them separately below.
> > >
> > > **Q1.** The “new appendix proposition” is already included in our revised manuscript. It does not appear in the anonymous link, because that link is restricted to figures/tables. So, to make the point precise, we state the proposition and a proof sketch here.
> > >
> > > **Proposition (discrete pseudotrue coefficients in the ideal undamped case).**
> > >
> > > Assume the true continuous-time trajectory is the undamped sinusoid
> > > $z(t)=\tilde A\cos(\omega t+\psi), \omega^2=\gamma_0, \gamma_1=0,$ sampled at
> > > $t_k=t_0+k\Delta t, k=0,1,\dots,T.$
> > > Let $z_k:=z(t_k), d_{1,k}:=\frac{z_{k+1}-z_{k-1}}{2\Delta t}, d_{2,k}:=\frac{z_{k+1}-2z_k+z_{k-1}}{\Delta t^2},$
> > > for $k=1,\dots,T-1$, and consider the discrete regression loss actually used by our estimator:
> > > $L_{\mathrm{ODE}}^{\mathrm{disc}}(\eta_1,\eta_0) = \frac{1}{T-1}\sum_{k=1}^{T-1}\bigl(d_{2,k}+\eta_1 d_{1,k}+\eta_0 z_k\bigr)^2.$ Then:
> > >
> > > (1) The sampled sinusoid satisfies the exact centered-difference relation
> > > $d_{2,k}+\gamma_0^\Delta z_k=0\text{ for all }k=1,\dots,T-1,$ with $\gamma_0^\Delta:=\frac{4\sin^2(\omega\Delta t/2)}{\Delta t^2}, \gamma_1^\Delta:=0.$
> > > Hence $L_{\mathrm{ODE}}^{\mathrm{disc}}(\gamma_1^\Delta,\gamma_0^\Delta)=0.$
> > >
> > > (2) Let $d_1 := (d_{1,1},\dots,d_{1,T-1})^\top, d_2 := (d_{2,1},\dots,d_{2,T-1})^\top, z := (z_1,\dots,z_{T-1})^\top,$ and define $X := [d_1;z], y := -d_2.$ If $X$ has full column rank, then $(\eta_1^\star,\eta_0^\star)=(\gamma_1^\Delta,\gamma_0^\Delta)=(0,\gamma_0^\Delta)$ is the unique minimizer.
> > >
> > > (3) As $\Delta t\to 0$, $\gamma_1^\Delta=0, \gamma_0^\Delta=\frac{4\sin^2(\omega\Delta t/2)}{\Delta t^2}=\gamma_0+O(\Delta t^2).$
> > >
> > > **Proof.**
> > >
> > > Write $\theta_k:=\omega t_k+\psi, z_k=\tilde A\cos\theta_k.$
> > > Then
> > > $z_{k+1}=\tilde A\cos(\theta_k+\omega\Delta t), z_{k-1}=\tilde A\cos(\theta_k-\omega\Delta t).$
> > > Using the trigonometric identity $\cos(\theta+\alpha)-2\cos\theta+\cos(\theta-\alpha)=-4\sin^2(\alpha/2)\cos\theta,$ we obtain $z_{k+1}-2z_k+z_{k-1}=-4\sin^2(\omega\Delta t/2)z_k.$
> > > Dividing by $\Delta t^2$ yields $d_{2,k}=-\frac{4\sin^2(\omega\Delta t/2)}{\Delta t^2}z_k=-\gamma_0^\Delta z_k.$
> > > Therefore $d_{2,k}+\gamma_1^\Delta d_{1,k}+\gamma_0^\Delta z_k=d_{2,k}+0\cdot d_{1,k}+\gamma_0^\Delta z_k=0$ for every interior index $k$, which proves part (1).
> > >
> > > For part (2), define $\eta:=(\eta_1,\eta_0)^\top$. Then $L_{\mathrm{ODE}}^{\mathrm{disc}}(\eta_1,\eta_0)=\frac{1}{T-1}\|y-X\eta\|_2^2.$ By part (1), we have $y = X \eta^\star, \eta^\star := (0,\gamma_0^\Delta)^\top.$
> > > so $\eta^\star=(0,\gamma_0^\Delta)^\top$ achieves zero loss. Now suppose another parameter pair $\eta$ also achieves zero loss. Then $0=\|y-X\eta\|_2^2=\|X\eta^\star-X\eta\|_2^2=\|X(\eta^\star-\eta)\|_2^2.$ Hence $X(\eta^\star-\eta)=0.$
> > > If rank$(X)=2$, then $X$ has trivial null space, so necessarily $\eta=\eta^\star$. This proves uniqueness.
> > >
> > > For part (3), let $x=\omega\Delta t/2$. Using $\sin x=x-\frac{x^3}{6}+O(x^5),$ we obtain $\sin^2 x=x^2-\frac{x^4}{3}+O(x^6)$. Substituting $x=\omega\Delta t/2$ gives $\gamma_0^\Delta=\frac{4\sin^2(\omega\Delta t/2)}{\Delta t^2}=\omega^2-\frac{\omega^4}{12}\Delta t^2+O(\Delta t^4).$ Since $\omega^2=\gamma_0$, the stated expansion follows.
> > >
> > > **Interpretation.**
> > > This proposition does *not* overturn Thm. 4.5. In continuous time, a single undamped trajectory still fails the coverage condition and is therefore not structurally identifiable in our theory. The proposition only explains why the *discrete* centered-difference estimator can nevertheless behave well on long enough sampled clips: it is converging to a well-defined discrete pseudotrue target, and that target is already $O(\Delta t^2)$-close to the continuous-time ground truth. Longer clips help because they span more phase and make the discrete regression better conditioned.
> > >
> > > **Q2.**
> > > We think we largely agree on the substance. The new state-consistency ablation is not meant to show that Def. 3.1 is unnecessary; it makes a narrower empirical point, namely that the finite-sample estimator has some robustness to modest violations, while stronger time-varying nuisance does break recovery. Likewise, our response on “going beyond 1D” was not meant to suggest that the extension is straightforward.
> > >
> > > Our intended claim is that the present paper isolates the first nontrivial case cleanly, and that doing so already makes the next obstacle explicit: extending a scalar coverage argument to higher-dimensional latent geometry is genuinely nontrivial. So we think the remaining gap is mainly about scope and significance, rather than about the correctness of the technical core. We hope that even a narrow but sharp identifiability result can still be useful as a starting point for broader work.
> > >
> > > Thanks again for your thoughtful follow-up comments.

---

### Official Review · Reviewer_NWQz · 2026-03-10

**Soundness:** 4
**Presentation:** 4
**Significance:** 2
**Originality:** 3
**Overall Recommendation:** 5
**Confidence:** 4

**Summary:**

The paper is proposing an encoder-only system identification method for second-order ODEs. Its key contribution is the theoretical analysis of the circumstances under which their method can succeed. The paper demonstrates results on both real and synthetic benchmarks, outperforming prior work.

**Compliance With Llm Reviewing Policy:**

Affirmed.

**Final Justification:**

The authors addressed my comments and made a strong case with a real-world example application, which I really liked!

**Key Questions For Authors:**

I don't think that I will be able to improve my score much, as it is mainly driven by the scope that you address. I commend you for keeping your scope tight and nailing the problem within that scope.

I guess in order to significantly raise my score, the authors would need to argue how their method could at least inspire methods that have a shot at working in real-world, messy problems, and what follow-up work they imagine could significantly increase the generality of their method to physical systems of practical interest? What are the applications - AI for science, robotics, ...? Do you have specific problems that you could identify?

**Limitations:**

Yes.

**Strengths And Weaknesses:**

\textbf{Presentation.}
I am not an expert in this area, and nevertheless found the paper easy to follow. Theoretical analysis was contextualized well with its motivation and key insights were discussed both in text and in math, making the paper easy to follow. Figures are explanatory and pretty.

Two criticisms remain:
1.Presentation of Table 2. Instead of presenting string length estimates, please present the *error* in string length estimates, as is standard in the ML literature. As it is, readers will get snagged on this presentation.
2. The paper should highlight the methodological difference to LFPV more clearly in the related work section, that is, you should point out explicitly that you make a minor change (which is a different computation of the derivative in latent space). This will clarify the change in performance in Table 2, which is otherwise a bit confusing b/c LPFV is mostly identical. You mention this in the text, but it would be good to also mention in the related work.

\textbf{Originality}
I understand that the method introduced by Garcia et al is very similar to the method introduced in this paper - both method and loss are almost identical. The key contributions of this paper, then, are the theoretical analysis of identifiability. There is a minor methodological contribution in how derivatives are computed.

\textbf{Soundness.}
I found the paper to be technically sound as much as I can judge it; though I did not verify all proofs in the paper. The high-level idea is certainly sound: \textit{if} there assumption 3.1 is valid (which is a rather strong assumption as far as I can grasp, more on that next), it is certainly reasonable to expect that a latent spaces that is temporally regularized to not collapse in terms of variance ought to capture some information about the dynamics displayed in the video. It is further reasonable to expect that if we have a strong prior about which system we seek - in this case, a second-order ODE - fitting that model to the latents and jointly training encoder and ODE parameters might identify the system.

While I did not verify the proofs, I did rethink the assumptions and proof ideas, and find them technically sound. The general idea is that if assumption 3.1 holds, then for underdamped systems, the scalar latent will pass through the same value in the latent space three times for a reasonable observation period. Then, in turn, if that is true, the only encoders that map video to latents are such that the ODE in latent space is an affine reparameterization of the original ODE. This makes sense to me!

\textbf{Significance.}
This is perhaps the key weakness of this paper. The paper uses world models as the motivation for the whole system identification approach. Yet, the paper makes exceedingly strong assumptions that make a practical application in real-world contexts (particularly those that world models are interesting for, such as robot planning and control) questionable. First, we assume a second-order ODE - these are not exactly omnipresent in real-world applications. Next, assumption 3.1 is actually very strong, as the authors note themselves. For many practical applications of interest, cameras are moving, lighting *is* changing, and there is an outrageous number of nuisance factors.

Nevertheless, I wouldn't significantly ding the paper for this. It has a clear scope and the scope is IMO significant enough and "inspiring".

---

> ### Author Rebuttal · Authors · 2026-03-30
>
> Thank you for the thoughtful review and for the clear comments on scope, practical relevance, and presentation. We found your feedback very helpful and respond point by point below.
>
> **1.Scope, applications, and follow-up work.**
> We agree that the theorem covers a deliberately narrow setting: a scalar latent, a homogeneous second-order LTI ODE, and state consistency. We have revised the framing to make this boundary explicit earlier in the paper. Our claim is therefore not that we already solve general physics-from-video, but that we provide a first sharp identifiability result for encoder-only video models under explicit assumptions.
>
> Even within this narrower scope, we do not view the setting as only a toy example. A concrete application is video-based free-decay monitoring of a bridge, beam, or other slender structure after a short disturbance. Over a window where one dominant mode is active and external forcing is weak, the motion is commonly approximated by a single damped second-order ODE. In that regime, $\gamma_0$ is the stiffness-like coefficient and $\gamma_1$ is the effective damping coefficient; together they determine the dominant oscillation frequency and decay. A stationary camera can record this motion remotely, and our method can then estimate $\gamma_0$ and $\gamma_1$ from the video. Changes in these estimated quantities across inspections can serve as diagnostic signals; for example, a drop in dominant frequency or an abnormal change in damping can indicate a change in stiffness or energy dissipation.
>
> As for generalization, we see the path as staged rather than one-shot: first multi-dimensional latents and coupled/multi-mode linear systems, and then forced dynamics and mild departures from LTI such as weak nonlinearity. A natural intermediate goal in the multi-dimensional case is not immediately full parameter-level identifiability, but to constrain the learned latent to the physical latent up to a restricted transformation class---the multi-dimensional analogue of the affine relation proved here in 1D. That would already give a meaningful notion of identifiability for coupled modes and provide a foundation for subsequent parameter recovery. In that sense, we view the present paper as a starting point for a broader research program: first pin down when physically meaningful parameters are uniquely learnable from video in the simplest nontrivial encoder-only setting, and then generalize those identifiability tools to richer physical systems.
>
> **2.Practical boundary of Assumption 3.1.**
> We agree that Assumption 3.1 is strong, especially once camera motion, lighting variation, and other nuisance factors enter. We do not claim that arbitrary real-world video satisfies this regime. To better characterize the practical boundary, we added a synthetic state-consistency stress test spanning complex backgrounds, moving clutter, camera jitter, and lighting variation. The results show that the finite-sample pipeline often tolerates weak-to-moderate nuisance, while stronger time-varying nuisance eventually breaks recovery. Representative stress-test frames and full quantitative results are provided in the anonymous supplement: https://conferenceml.github.io/icml2026-supplement/. Our point is therefore not that the present paper already solves messy real-world physics-from-video, but that it cleanly isolates the identifiability question and that the resulting estimator exhibits nontrivial robustness beyond the exact theorem regime. We discuss this more fully in our response to reviewer gGHi.
>
> **3.Difference to LPFV.**
> We agree your point and revised the related-work. “Relative to LPFV (Garcia et al., 2025), our method is intentionally close at the architectural level: both are encoder-only and enforce physics directly in latent space. The main contribution of this paper is therefore the identifiability analysis. Methodologically, LPFV uses an Euler/one-sided derivative construction, whereas we use centered finite differences for both first and second derivatives, which reduces the discretization bias in the latent ODE residual from first order to second order in $\Delta t$. LPFV also uses a KL-style anti-collapse term, whereas we use a variance-floor regularizer. ”
>
> **4.Table presentation.**
> We agree that an error-based presentation is clearer. We therefore revised the real-video pendulum table to report, for each ground-truth length, both the original estimate (mean$\pm$std over 5 clips) and RMSE. This preserves the variance information while making error comparison direct.
>
> For PAIG/NIRPI/LPFV, only published mean$\pm$std estimates are available from Garcia et al. (2025), so exact MAE cannot be reconstructed without clip-level predictions. RMSE, however, is derivable via
> $\mathrm{RMSE}=\sqrt{(\bar L-L^\star)^2+\tfrac{n-1}{n}s^2}, \quad n=5,$
> and thus provides an apples-to-apples error metric for all methods. Updated table: see the anonymous supplement.
>
> Thank you again for the very helpful review.

---

> > ### Author Rebuttal · Reviewer_NWQz · 2026-04-03
> >
> > I thank the authors for the careful rebuttal, especially the real-life example for how this method could be useful, which I really enjoyed. I increase my score!

---

> > > ### Author Response · Authors · 2026-04-04
> > >
> > > Thank you very much for the kind follow-up and for raising your score. We are especially happy that the concrete real-life example helped clarify why we believe this line of work can be useful in practice.

---

### Official Review · Reviewer_gGhi · 2026-03-13

**Soundness:** 2
**Presentation:** 3
**Significance:** 3
**Originality:** 4
**Overall Recommendation:** 4
**Confidence:** 2

**Summary:**

The paper studies the identifiability of physical parameters from raw video in an encoder-only, decoder-free setting for scalar, second-order, homogeneous LTI ODEs. The core idea is a geometric “level-set slope coverage” condition: if the latent trajectory revisits each state level with three distinct velocities, then any smooth latent reparameterization compatible with the same ODE must be locally affine, which makes the physical parameters identifiable. The paper further examines this result across damping regimes and provides a finite-sample error bound that separates the contributions of encoder mismatch, observational noise, and finite-difference approximation error. Experiments on synthetic data and real pendulum-style videos are used to validate the theory.

**Compliance With Llm Reviewing Policy:**

Affirmed.

**Final Justification:**

I appreciate the author’s detailed response. I have raised my evaluation accordingly.

**Key Questions For Authors:**

Key Questions For Authors:
1) The analyses are all based on time-invariant linear second-order ODEs. How sensitive are the results to mild violations of linearity (e.g., nonlinear forces, slowly varying parameters)?
2) There are results on the finite sample error bounds that involve different sources of errors. Should these errors also impact the minimal trajectory requirements?
3) The paper introduces a variance-floor regularizer to prevent latent collapse: how does this regularizer compare to alternatives typically used in autoencoders: whitening, contrastive, and so on?

**Limitations:**

yes

**Strengths And Weaknesses:**

The main theoretical contribution is sound. The level-set slope coverage condition provides geometric intuition and yields a clean identifiability argument via the vanishing of a quadratic in the trajectory slope.  Empirical experiments with synthetic data are carefully designed to validate the theoretical hypothesis.

The real-data evidence in the empirical section is limited for a paper making a broad claim about physics from video, especially since the state consistency assumption (definition 3.1) could be a strong assumption for real-life data.
The paper is conceptually clear and well structured.

A notable weakness is that the presentation of method comparisons could be improved. It would be much easier to compare methods if the true parameter values were on the same plots rather than on a separate plot on a different page. Or provide some error plots that are more straightforward.

The paper addresses a meaningful and undertheorized problem in AI for science: Is the model really learning the physics or merely fitting the data? The framing is well motivated and distinct from prior work that assumes access to state trajectories or relies on reconstruction objectives.

The main concern with this paper is its narrow theoretical scope. It only covers 1D latents and homogeneous second-order LTI dynamics under a state-consistency assumption. That is a fair starting point, but it materially limits significance for broader physics-from-video settings, where coupled modes, inputs, nonlinearity, and partial observability are central. The paper acknowledges some of this, but the practical gap remains large.

The originality of this paper is strong. The authors approach the problem of determining whether AI are learning physics from video or merely fitting data from a unique and important perspective. It is important to first determine whether the physics are actually learnable from data before making further claims about how well AI models capture physics.

---

> ### Author Rebuttal · Authors · 2026-03-30
>
> Thank you for the thoughtful review and for the clear questions on scope, finite-sample issues, and practical relevance. We found your comments very helpful and respond point by point below.
>
> **1.State consistency/real-data gap.**
> We agree that Def. 3.1 is strong, especially for real video. Our claim is not that arbitrary raw video automatically satisfies it, but that it is the explicit structural assumption under which exact identifiability can be proved. This is also why Remark 3.2 recommends controlled capture conditions such as static camera, stable exposure/lighting, and simple backgrounds.
>
> To probe the practical gap more directly, we added a synthetic state-consistency stress test in the underdamped regime. We keep the ODE fixed and perturb only the observation model with exogenous nuisance. The result is encouraging but not unlimited: static background noise causes almost no degradation; camera jitter remains robust across all three severities; moving clutter is tolerated at weak/moderate levels but breaks at the strongest level. Dynamic lighting: Level 1 remains close to the clean baseline; at Level 2, one of the five seeds converges to a poor local optimum, while the remaining four stay reasonably close to the ground truth; at Level 3, recovery does not collapse, but $\hat{\gamma}_1$ is consistently biased downward. So the right message is not that state consistency does not matter, but that the finite-sample pipeline retains nontrivial robustness to modest violations and degrades under stronger time-varying nuisance.
>
> We do not claim a general way to verify or enforce Def. 3.1 from pixels alone. What we can currently offer is: (i) explicit capture/preprocessing guidance, and (ii) this new stress test, which makes the practical tolerance much more concrete. We believe this materially improves the paper’s discussion of the real-data gap. Sample frames and result table are included in the anonymous link: https://conferenceml.github.io/icml2026-supplement/.
>
> **2.Q1: mild nonlinearity/slow time variation.**
> The exact theorem is limited to homogeneous second-order LTI dynamics. For weak nonlinearity or slowly varying parameters, we do *not* claim exact structural identifiability; the right interpretation is recovery of *effective local linear parameters* over the observed window. This is already how the real examples should be read: the clean pendulum uses the standard small-angle linearization of $\sin(\theta)$, and the wheel-mounted-phone experiment uses an effective linear damping approximation over the clip. Likewise, if parameters drift slowly relative to the clip length, $(\hat{\gamma}_0,\hat{\gamma}_1)$ should be read as a local frozen-coefficient approximation. For broader applications and generalization paths, please see our response to Reviewer NWQz.
>
> **3.Q2: finite-sample errors vs. minimal trajectory requirements.**
> Not at the structural level, but yes at the practical finite-sample level. Thms. 4.2-4.6 answer an “in principle” question: under the ideal continuous-time model, when does the trajectory contain enough information for unique recovery? Finite-sample noise and discretization do not change those structural conditions. What they do change is the conditioning margin for stable estimation. This is the role of Eq. (9): the statistical, noise, and discretization terms are all scaled by $1/\psi_{\min}$, so a trajectory that barely satisfies coverage may still be numerically fragile. Thus the theoretical minimum is unchanged, but in practice one may need a longer clip or more diverse trajectories to obtain a comfortable buffer.
>
> **4.Q3: variance-floor regularizer.**
> The role of the variance floor is very specific. In an encoder-only ODE objective, a nearly constant latent is a degenerate solution, so we need a regularizer that both prevents per-clip collapse and keeps the regression well conditioned. In our theory, the variance floor lower-bounds within-clip latent variance, which in turn lower-bounds $\psi_{\min}$ via Lem. D.1; since the main finite-sample error terms scale like $1/\psi_{\min}$, this is what keeps the parameter error controlled.
>
> We therefore do not claim that the variance floor is universally better than whitening, contrastive, or covariance-based objectives. Our narrower point is that it is the most direct regularizer for the quantity appearing in our bound. Empirically, Tab. 4 supports this narrower claim: KL and variance-floor both work in under/undamped regimes, while variance-floor is much more stable in critical/overdamped regimes.
>
> **5.Presentation of method comparison.**
> We agree and made two changes to improve readability. First, we added regime-specific ground truth directly to the captions of Tables 2-4. Second, we revised the real-video pendulum table to report, for each ground-truth length, the original estimate and the corresponding RMSE. Please see our response to Reviewer NWQz for the RMSE conversion.
>
> Thank you again for the very helpful review.

---

> > ### Author Rebuttal · Reviewer_gGhi · 2026-04-03
> >
> > I appreciate the author’s detailed response. I have raised my evaluation accordingly.

---

> > > ### Author Response · Authors · 2026-04-04
> > >
> > > Thank you very much for the thoughtful follow-up and for raising your evaluation. We are glad the additional clarifications were helpful.

---

### Decision · Program_Chairs · 2026-04-30

**Decision:**

Accept (regular)

**Comment:**

## Summary

The paper studies the identifiability of physical parameters from raw video in an encoder-only, decoder-free setting for scalar, second-order, homogeneous LTI ODEs. The core idea is a geometric “level-set slope coverage” condition: if the latent trajectory revisits each state level with three distinct velocities, then any smooth latent reparameterization compatible with the same ODE must be locally affine, which makes the physical parameters identifiable. The paper further examines this result across damping regimes and provides a finite-sample error bound that separates the contributions of encoder mismatch, observational noise, and finite-difference approximation error. Experiments on synthetic data and real pendulum-style videos are used to validate the theory.



**The reviewers pointed out the following strengths:**

1. The claims in the paper are matched with the theoretical investigation as well as the experimental setup.

2. The level-set slope coverage condition provides geometric intuition and yields a clean identifiability argument via the vanishing of a quadratic in the trajectory slope.

3. The variance-floor regularizer is a simple and practically useful contribution that outperforms the KL regularizer baseline.

4. The framing is well motivated and distinct from prior work that assumes access to state trajectories or relies on reconstruction objectives.

5. The problem this paper addresses is important. Recovering physical parameters from raw video without ground-truth labels has clear practical value, e.g., in biology and robotics.

6. **Overall:** The main theoretical contribution of the paper is sound, it is well structured, and addresses a timely and important problem.  ​​

**The reviewers pointed out the following weaknesses:**

The main concern with this paper is its narrow theoretical scope. It only covers 1D latents and homogeneous second-order LTI dynamics under a state-consistency assumption.

## Final justification

All of the reviewers agreed that the paper is technically solid and well-motivated, with clear theoretical analysis and supporting experiments. The problem addressed by the paper is timely, and of significant interest to the representation learning community. The author-reviewer discussion addressed most of the weaknesses in the paper and two of the reviewers raised their scores from 3 to 4 and from 4 to 5. ​​The questions/concerns raised by the third reviewer have also been properly addressed during the rebuttal, and the reviewer recommended positively.  The reviewers pointed out that the authors made a strong case  with a real-world example application during the rebuttal. I would recommend adding this example to the final version.